



# Stable climate simulations using a realistic GCM with neural network parameterizations for atmospheric moist physics and radiation processes

Xin Wang[1], Yilun Han[2], Wei Xue[1], Guangwen Yang[1], Guang J. Zhang[3]

[1]Department of Computer Science and Technology, Tsinghua University, Beijing, 100084, China
[2]Department of Earth System Science, Tsinghua University, Beijing, 100084, China
[3]Scripps Institution of Oceanography, La Jolla, CA USA

*Correspondence to*: Wei Xue (xuewei@mail.tsinghua.edu.cn), Yilun Han (hanyl16@mails.tsinghua.edu.cn)

**Abstract.** In climate models, subgrid parameterizations of convection and cloud are one of the main reasons for the biases in
precipitation and atmospheric circulation simulations. In recent years, due to the rapid development of data science, Machine learning (ML) parameterizations for convection and clouds have been proven the potential to perform better than conventional parameterizations. At present, most of the existing studies are on aqua-planet and idealized models, and the problems of simulated instability and climate drift still exist. In realistic configured models, developing a machine learning parameterization scheme remains a challenging task. In this study, a group of deep residual multilayer perceptrons with strong
nonlinear fitting ability is designed to learn a parameterization scheme from cloud-resolving model outputs. Multi-target training is achieved to best balance the fits across diverse neural network outputs. The optimal machine learning parameterization, named NN-Parameterization, is further chosen among feasible candidates for both high performance and long-term simulation. The results show that NN-Parameterization performs well in multi-year climate simulations and reproduces reasonable climatology and climate variability in a general circulation model (GCM), with a running speed of about
30 times faster than the cloud-resolving model embedded Superparameterized GCM. Under real geographical boundary conditions, the hybrid ML-physical GCM well simulates the spatial distribution of precipitation and significantly improves the frequency of precipitation extremes, which is largely underestimated in the Community Atmospheric Model version 5 (CAM5) with the horizontal resolution of 1.9°×2.5°. Furthermore, the hybrid ML-physical GCM simulates a stronger signal of the Madden-Julian oscillation with a more reasonable propagation speed, which is too weak and propagates too fast in CAM5.
This study is a pioneer to achieve multi-year stable climate simulations using a hybrid ML-physical GCM in actual land-ocean boundary conditions. It demonstrates the emerging potential for using machine learning parameterizations in climate simulations.





## 1 Introduction

The general circulation models (GCMs) have been widely used for studying climate variability, prediction and projections.
Despite decades of GCM development, most GCMs still suffer from many systematic biases, especially at low latitudes. A prominent tropical bias in most current GCMs is the double intertropical convergence zone (ITCZ) syndrome, which is characterized by two parallel zonal bands of annual precipitation straddling the equator over the central and eastern Pacific (Mechoso et al., 1995; Lin, 2007; Zuidema et al., 2016; Zhang et al., 2019; Lu et al., 2021). Convectively coupled equatorial waves and the Madden-Julian Oscillation (MJO), featured by eastward propagating convective cloud clusters, are also not well
simulated in GCMs (Ling et al., 2017). The simulated MJOs in GCMs are often too weak and propagate too fast (Lin et al., 2006).

Many studies have attributed most of these biases to the imperfection of the parameterization schemes for atmospheric moist convection and cloud processes in current GCMs (Song and Zhang, 2009; Zhang and Song, 2010; Oueslati and Bellon, 2013; Crueger and Stevens, 2015; Deng et al., 2016; Cao and Zhang, 2017; Peters et al., 2017; Song and Zhang, 2018). Cloud-
related processes span a large range of spatial scales, from micron-scale cloud nucleation, meter-scale turbulence, to individual convective cells and organized convective systems, which are a few kilometers to hundreds of kilometers in size, and to tropical disturbances, which have a spatial scale of thousands of kilometers. They directly influence the radiation balance and hydrological cycle of the earth system. Their interaction with the atmospheric circulation affects the transport and distribution of energy and is the largest source of precipitation biases. Therefore, it is very important to simulate the cloud and convection
process in GCMs correctly. However, the current GCMs used for climate simulation have a horizontal resolution of ~100km and a vertical hydrostatic coordinate. Therefore, in most GCMs, besides parameterized cloud microphysics, convection and its influence on the atmospheric circulation are represented by convective parameterization schemes, which are usually based on simplified theories, limited observations, and empirical relationships (Tiedtke, 1989; Zhang and McFarlane, 1995; Zhang, 2002; Wu, 2012; Storer et al., 2015; Zhao et al., 2018; Seo et al., 2019; Xie et al., 2019; Lopez-Gomez et al., 2020; Hourdin
et al., 2020). Those schemes regard convective heat and moisture transport as the collective effects of idealized individual kilometer-scale convective cells. They cannot represent the effects of many complicated convective structures, including organized convective systems, leading to large uncertainties and biases in climate simulations. (Bony et al., 2015).

Cloud Resolving Models (CRMs), on the other hand, have long been used to simulate convection. Because CRMs have higher horizontal and vertical resolutions and can explicitly resolve the thermodynamic processes in convection, they simulate
convection more accurately, including convective organization (Feng et al., 2018). In recent years, CRMs have been applied to low-resolution GCMs to replace conventional cumulus convection and cloud microphysical parameterization schemes, such as the superparameterized version of the Community Atmosphere Model (SPCAM) developed by the National Center for Atmospheric Research (Grabowski and Smolarkiewicz, 1999; Grabowski, 2001, 2004; Khairoutdinov and Randall, 2001; Randall et al., 2003; Khairoutdinov et al., 2005). Compared with conventional cumulus convection and cloud microphysical
parameterization schemes, SPCAM performs better in simulating mesoscale convective systems, diurnal cycles of





precipitation, monsoons, and MJOs (Bretherton et al., 2014; Jin et al., 2016; Jones et al., 2019; Hannah et al., 2020). However, the computing resource required for SPCAM is an order of magnitude larger compared with that for CAM. The use of SPCAM in long-term climate simulations and ensemble prediction is restricted by the current computing resource. Developing novel and computationally efficient schemes for high performance convection and cloud processes is still an open problem in GCM
development.

The conventional theory-driven parameterization schemes are based on the limited mathematical theory and observations, guided by physical laws of atmospheric motion, while the data-driven parameterization scheme can identify and extract complex nonlinear relationships from high resolution and high-fidelity data sets. More recently, the rapid development of machine learning (ML) technologies, especially deep learning technologies such as neural networks (NNs), has provided novel
approaches to constructing parameterization schemes. Machine learning can identify and discover complex nonlinear relationships that exist in large data sets and model them. Several studies have used machine learning methods to develop convection and cloud parameterization schemes (e.g., Schneider et al., 2017; Dueben and Bauer, 2018; Gentine et al., 2018; Rasp et al., 2018). These studies followed a similar approach. The first step is to derive a target dataset from a reference simulation, which is later used for machine learning algorithm training. Then, the trained ones are often evaluated offline
against other independent reference simulations and finally implemented in a GCM to replace the conventional parameterization schemes (Rasp, 2020).

Krasnopolsky et al. (2013) first proposed a proof-of-concept for developing convection parameterization based on the NN technique. Specifically, an ensemble of shallow NNs was applied to learn convective temperature and moisture tendencies, with training data from CRM simulations forced by observations in the tropical western Pacific. The resulting convective
parameterization scheme was able to simulate the main features of cloud and precipitation in the NCAR CAM4 diagnostically. However, the key issue of prognostic validation in 3-D GCMs was not addressed. Recent studies have investigated ML parameterizations in prognostic mode in simplified aqua-planet GCMs. For example, Rasp et al. (2018) developed a deep NN algorithm to predict convection and clouds, which was trained with the data from an aqua-planet SPCAM. The NN parameterization was then implemented in the corresponding aqua-planet CAM and produced multi-year prognostic results
close to SPCAM. They found that minor changes, either to the training dataset or in the input/output vectors, can lead to model integration instabilities. Brenowitz and Bretherton (2019) fitted a DNN for convection and clouds to the coarse-grained data from a near-global aqua-planet cloud-resolving simulation using the System for Atmospheric Modeling (SAM). The NN scheme was then tested prognostically in a coarse-grid SAM. Their results showed that there were unphysical correlations learned by the network, and information in the upper levels from the input vector had to be removed to produce stable long-
term simulations. Rather than using NNs, Yuval and O'Gorman (2020) used random forest to develop an ML parameterization based on the training data from a high-resolution idealized 3-D model with a setup of equatorial beta plane. Stable simulations that replicated the climatology of the high-resolution model were achieved after they implemented this parameterization in a coarse resolution GCM.



Although machine learning is a promising approach for developing new parameterizations, issues of instability and
climate drift still prevent it from the application of machine learning parameterization in models. Rasp (2020) proposed coupled
online learning to tackle instabilities and biases in NN parameterizations, which is a concept illustration using the idealized
Lorenz 96 model. In real-world climate models with varied underlying surfaces, convection and clouds are more diverse under
different climate backgrounds, which makes the task of developing ML-based parameterizations more complicated. A few
early works have shown the feasibility of using neural networks fitting cloud processes in real-world models. Han et al. (2020)
used a 1-D deep residual convolutional neural network (ResNet) to emulate moist physics in SPCAM. This ResNet based
parameterization fitted the targets with high accuracy and is successfully implemented in a single column model. Mooers et
al. (2021) got a high-skill DNN via auto-learning technique and forced an offline land model with DNN emulated atmospheric
fields. However, neither of these studies have tested their NNs prognostically for long-term simulations with ML parameterized
GCMs. This study uses a group of NNs to emulate convection and cloud processes in SPCAM with an actual global land-
ocean distribution. We apply two innovative methods in neural network models: multi-target training to achieve balanced
results across diverse neural network outputs and multilayer perceptron with residual blocks (ResMLP) to enhance nonlinear
fitting ability. Furthermore, an optimal DNN emulator is chosen among a set of well trained neural networks by using multi-
target training and ResMLP, to achieve both high-performance and long-term simulations. The NN parameterization scheme
is then implemented in the realistically configured CAM to obtain long-term stable simulations. Technically, NNs are
commonly implemented via high-level programming languages such as Python and deep learning libraries. However, GCMs
are mainly written in Fortran, making it difficult to integrate with deep learning algorithms. Therefore, we introduce a DNN-
GCM coupling platform in which the DNN model and the GCM interact through data transmission. This coupling strategy can
facilitate the development of ML-physical hybrid models with high flexibility. Under real-geography boundary conditions, our
work achieves 10-year stable climate simulations in Atmospheric Model Intercomparison Project (AMIP)-style experiments
by using a hybrid ML-physical GCM. To our knowledge, this is the first time a decade-long stable real-world climate
simulation is achieved with a NN-based parameterization.

The remainder of this paper is organized as follows. Section 2 briefly describes the model, the experiments, the DNN
algorithm, and the DNN-GCM coupling platform. Section 3 presents the offline validation of the DNN scheme, focusing on
the output temperature and moisture tendencies. Results of multi-year simulations, employing the DNN parameterization
scheme, are shown in section 4. A summary and conclusions are presented in section 5.

## 2 Methods and data

In this study, we choose SPCAM as the reference model to generate target simulations. A group of DNNs is trained with the
target simulation data using optimized hyperparameters. Then, they are organized as a subgrid physics emulator and
implemented into the superparameterized version of Community Atmospheric Model (SPCAM), replacing both the CRM and
CRM radiation. This DNN-enabled GCM is referred to as NNCAM hereafter.



## 2.1 SPCAM setup and data generation

The GCMs used in this study are the CAM5.2 developed by the National Center for Atmospheric Research and its superparameterized version SPCAM (Khairoutdinov and Randall, 2001; Khairoutdinov et al., 2005). A complete description of CAM5 is given by Neale et al. (2012). The dynamic core of CAM5 has a horizontal resolution of 1.9°×2.5° and 30 vertical
levels with a model top at about 2 hPa. To represent moist processes, CAM5 adopts a plume-based treatment of shallow convection (Park and Bretherton, 2009), a mass-flux parameterization scheme for deep convection (Zhang and McFarlane, 1995), and an advanced two-moment representation of cloud microphysical processes (Morrison and Gettelman, 2008; Gettelman et al., 2010). In the AMIP experiments we conducted, CAM5 is coupled to a land surface model Community Land Model version 4.0 (Oleson et al., 2010) and uses prescribed sea surface temperatures and sea ice concentrations.

In this study, SPCAM is used to generate the training data. In SPCAM, a two-dimensional (2-D) CRM is embedded in each grid column of the host CAM. The 2-D CRM has 32 grid points in the zonal direction and 30 vertical levels that are shared with the host CAM. The CRM handles convection and cloud microphysics to replace the conventional parameterization schemes, and the radiation is calculated on the CRM subgrids to include the cloud-radiation interaction at cloud scale (Khairoutdinov et al., 2005), referring to CRM radiation hereafter. The other physical processes and the dynamic core are
computed on the CAM grid as usual. One conceptual advantage of using SPCAM as the reference simulation is that the subgrid and grid-scale processes are clearly separated, making it easy to define the parameterization task for an ML algorithm (Rasp, 2020).

## 2.2 NN-Parameterization

In the NNCAM, a DNN emulator is trained with the target data from the 2-D CRM and CRM radiation to achieve better results
than the CAM5 with conventional parameterizations. In the following section, we refer to this DNN emulator as the NN-Parameterization. As shown in Figure 1, the NN-Parameterization replaces the CRM moist physics and radiation calculations in SPCAM. It is coupled with the dynamic core and other parameterization schemes in each time integration loop in NNCAM.

      NN-Parameterization is a challenging deep learning application. It integrates the NNs into a scientific computing program for continuous time integration. In the numerical model system, the prediction errors of the NNs are at the risk of being
amplified by the continuous iterations with the dynamic core and other physical processes, causing model state drift and even model crashes in NNCAM. This is expected to be more difficult in the GCMs with real land-ocean distributions. Because of the energy exchange between the land and the atmosphere, the nonlinearity of the simulation system is more complicated than the idealized aqua-planet models, bringing more uncertainties and more numerical sensitivities. In this section, we propose a DNN based Parameterization that can fit the grid average of the 2-D CRM data in SPCAM, and on this basis, achieve high
computational performance and stable time integration.





### 2.2.1 Data sets

The NN-Parameterization, as a deep learning emulator of the CRM and the CRM radiation in SPCAM, is designed to replace physics processes in the host CAM5, including deep and shallow convection, cloud microphysics and macrophysics, and radiation. Therefore, the inputs of this emulator are borrowed from CRM inputs such as the grid-scale state variables and

forcings composed of the dynamic core and the planetary boundary diffusion. They are the specific humidity $q_v$, temperature T, largescale water vapor forcing $\left(\frac{\partial q_v}{\partial t}\right)_{ls}$ and temperature forcing $\left(\frac{\partial T}{\partial t}\right)_{ls}$. Additionally, we select surface pressure $P_s$ and solar insolation (SOLIN) at the top of the model from the radiation module. The outputs of NN-Parameterization are subgrid-scale tendencies of moisture $\left(\frac{\partial q_v}{\partial t}\right)$ and of temperature $\left(\frac{\partial T}{\partial t}\right)$ at each model level as well as net shortwave and longwave radiative fluxes at both the surface and the TOA. This heating is composed of moist heating in the CRM and the GCM-grid-averaged

radiative heating from the CRM radiation module. Since we use the real geography, we also include direct and diffuse downwelling solar radiation fluxes as output the NN-Parameterization to force the coupled land surface model, which is critical to improve the performance of the NNCAM. Specifically, they are solar downward visible direct to surface (SOLS), solar downward near infrared direct to surface (SOLL), solar downward visible diffuse to surface (SOLSD), and solar downward near infrared diffuse to surface (SOLLD). Those downwelling solar radiation fluxes with separation of direct versus diffusion

are introduced for different land cover types and processes in the land surface model (Mooers et al., 2021). The precipitation is derived from column integration of predicted moisture tendency to keep basic water conservation.

Table 1 lists the input and output variables and their normalization factors. There are 30 model levels for each profile variables. Therefore, the input vector consists of 122 elements for 4 profile variables and 2 scalars, while the 68-element output vector is made of 2 profiles and 8 scalars. All input and output variables are normalized to ensure that they are in the same

magnitude before they are put into the NN-parameterization for training, testing, model prognostic validation. The normalization factor for each variable shown in the supplemented codebase is determined by the maximum of its absolute values.

In this study, the target model SPCAM is run for 3 years from January 1, 1997 to December 31, 1999 with a time step of 30 min. The first year of SPCAM simulations is for spinup, the second and third years are for training, with a total of 484

million samples. In the training process, 64% of the data are randomly selected for training and 16% is for validation in every training epoch to monitor the performance and to prevent overfitting, and the remaining 20% of the data are used for testing.

### 2.2.2 Multi-Target ResMLP

In this study, the NN-Parameterization is an isolated column subgrid parameterization. This means that the inputs and outputs of NN-Parameterization are both 1-D vectors. Compared with machine learning tasks such as computer vision and natural

language processing, the data input and output of NN-Parameterization are relatively simpler. Therefore, we choose the commonly used multilayer perceptron (MLP) to achieve a better generalization.





In the development of the NN parameterization scheme, it is found that the NN has a different fitting capability for different types of output variables. It usually has higher accuracy in predicting radiative fluxes and temperature tendency, while the accuracy is lower in predicting moisture tendency. This is also found in Gentine et al. (2018), in which the coefficient of determination ($R^2$) of radiative heating tendency is higher than that of moisture tendency at most model levels. The physical processes behind the data representing convection and radiation are different, introduce high nonlinearity in the NN-Parameterization.

The distribution of different physical variables also varies greatly. Using a single NN with one target to train the relatively independent physical variables, i.e., moisture tendency, temperature tendency, and radiation fluxes, inevitably causes mutual interference. Since gradient descending is applied to optimize the network in training, mutual interference between different targets is expected to cause the cancel out of gradient directions used for descending (Crawshaw et al., 2020; Zhang et al., 2021) and ultimately affect the convergence of the network.

As shown in Figure 2, we divide the single network prediction target, a single stacked 1-D output, into multiple targets, and use multiple NNs to train these targets separately. By doing so, we avoid the gradient cancellation between multiple targets and improve the convergence speed and fitting accuracy when training the network.

After dividing multiple targets, the learning target of NN-Parameterization still contains a lot of nonlinear information. This research attempts to construct a deeper neural network so that NNs can have stronger nonlinear fitting abilities. As the depth of a fully connected neural network increases, its performance becomes saturated and begins to decline when the depth increases further (He et al., 2016). For this reason, we use residual blocks to solve the problem of network degradation. By testing different depths and widths, we finally determined the network architecture for the optimal performance. As shown in Figure 3, a total of 7 residual blocks are used to form a deep neural network, and each residual block includes two 512 node-wide dense layers connected with a layer jump. Therefore, the deep neural network that has 14 layers with 3.5 million parameters is named ResMLP hereafter. As shown in Figure 4, under multi-target training with the same hyperparameters, ResMLP's fitting accuracy ($R^2$) for different output variables is better than MLP, achieving a balanced prediction result, that is, the $R^2$ of each variable is as close to 1 as possible, especially tendencies of moisture and temperature.

Ott et al. (2020) found that on aqua-planet, the fitting accuracy of an NN-Parameterization is positively correlated with the stability of prognostic validation. In the real-world prognostic validation, we also find that the fitting accuracy of NN-Parameterization has an important impact on stability. When NN-Parameterization is under-fitting, NNCAM can only run for a few days before it crashes. To obtain a feasible NN-Parameterization, a well-fit is necessary. As shown in Table 2, after multiple trials, we determine the hyperparameter configuration of ResMLP for high fitting accuracy. It can support NNCAM to run stably for at least several months.

However, we find that a high $R^2$ may not guarantee the stability of real-world prognostic validation. We have trained several groups of ResMLPs with doubled samples and epochs, having their $R^2$ increased by 1% to 2%. Surprisingly, NNCAM using these ResMLPs crashed even earlier than before. As a result, high fitting accuracy is necessary but not sufficient for both high performance and long-term simulations. In this work, we proposed a trial-and-error method to effectively find the optimal





neural network that can guarantee both multi-year and high performance simulations. Firstly, we used the hyperparameter configuration of Table 2 as a baseline and prepared 50 groups of ResMLPs with similar $R^2$ as candidate models using different train samples and epochs. Secondly, we conducted comprehensive prognostic tests on these candidate neural networks and obtained the feasible NN-Parameterization schemes that can support NNCAM's stable simulation for multiple years. To our

knowledge, running the relatively shorter NNCAM simulation is enough to screen out the feasible networks for stable simulations since we found that the ResMLP groups that cannot support long-term integration made NNCAM to collapse within half a year of simulation; and the feasible groups, on the other hand, coupled stably in NNCAM for the first half a year and showed no sign of crashes in the 10-year prognostic simulation. Finally, the optimal NN-Parameterization was selected for the best performance ResMLP group among the feasible candidats.

### 230 2.2.3 Implementation of NN-Parameterization

After being fully trained with CRM data, the NN-Parameterization is implemented into SPCAM to replace both the CRM and CRM radiation on the basis of coarse grid average. In the beginning of each timestep, NNCAM calls the NN-Parameterization and predict the moisture tendency $\left(\frac{\partial q_v}{\partial t}\right)$, the temperature tendency $\left(\frac{\partial T}{\partial t}\right)$ and radiation fluxes. Then the DNN predictions are returned to NNCAM, updating the model states and fluxes. Additionally, the surface total precipitation is derived from column

integration of the predicted moisture tendency. The near surface conditions of the atmosphere and downwelling radiation fluxes are transferred to the land surface model. After the coupling of the land surface model, the host CAM5 performs the planetary boundary layer diffusion and let its dynamic core complete a timestep integration (Figure 1). In the next timestep, the dynamic core returns the new model states to the NN-Parameterization as inputs again. During the whole process, NN-Parameterization and GCM will constantly update each other's status. How to couple the NN Parameterization (DNN) with

GCM and run efficiently and effectively is the key of the implementation of NNCAM. To solve these problems, we develop the DNN-GCM coupler that integrates DNN into NNCAM, which will be introduced in the following section.

### 2.3 The DNN-GCM Coupler

Deep learning research mainly uses machine learning frameworks based on Python interfaces to train neural network models and deploy them through C++ or Python programs. While GCM is mainly developed in Fortran, it is a very challenging work

to call a neural network model based on Python/C++ interface in GCM codes written in Fortran. Solving the problem of code compatibility between DNN and GCM can significantly help develop DNN based Parameterizations for climate models.

To implement a DNN based Parameterization in current climate model which is mostly developed in Fortran, many researchers try to get the network parameters (e.g., weight, bias) from the machine learning models and implement the DNN models with hard coding in Fortran. At runtime, NNCAM will call DNN as a function (Rasp et al., 2018; Brenowitz and

Bretherton, 2019). Recently, some researchers have developed a Fortran-neural network interface that can be used to deploy DNNs into GCMs (Ott et al., 2020). This interface can import neural network parameters from outside, and the Fortran-based



implementation ensures that it can be flexibly deployed in GCMs. However, embedding DNN in NNCAM is still a troublesome task with no existing coupling framework to support many of the latest network structures. This problem will restrict developers from building more powerful DNNs and deploying them in NNCAM.

In this research, we regard NN-Parameterization as a component model coupled to NNCAM. We develop the coupler to bridge NN-Parameterization with the host CAM5. Through this coupler, the neural network can communicate with the dynamic core and other physical schemes in NNCAM in each time step. When NNCAM is running, as shown in ① in Figure 5, the coupler receives the state and forcing output from dynamic core in Fortran based CAM5. For each input variable, we use the native MPI interface in CAM5 to gather the data of all processes to the master process into a tensor. Then, as shown in ② of

Figure 5, the coupler will transmit the gathered tensor through the data buffer to the NN-Parameterization running on the same node as the master process. The NN-Parameterization gets the input, infers the outputs, and transmits them back to the coupler. As shown in ③ of Figure 5, the coupler will first write these tendencies and radiation fluxes back to the master process and then broadcast the data to CAM5 processes running on the computing nodes through the MPI transmission interface. Therefore, other parameterizations get the predictions from NN-Parameterization to complete the follow-up procedures (④ in Figure 5).

In practice, the DNN-GCM Coupler introduces a data buffer that supports system-level interface, which is accessible by both Fortran based GCM and Python based DNN without supplementary foreign codes. This can avoid code compatibility issues when building Machine Learning coupled numerical models. It supports all mainstream machine learning frameworks, including native PyTorch and Tensorflow. Based on the coupler, one can efficiently and flexibly deploy the Deep Learning Model in NNCAM, and can even take advantage of the latest developed neural networks.

All neural network models deployed through DNN-GCM Coupler can support GPU accelerated inference to achieve excellent computing performance. In this study, we ran SPCAM and NNCAM on 196 CPU cores. NNCAM also used 2 GPUs for acceleration. During the NNCAM runtime, each time step of NNCAM requires NN-Parameterization to complete an inference and conduct data communication with NNCAM. This is a typical high-frequency communication scenario. We evaluated the amount of data (about 20MB for CAM5 with the horizontal resolution of $1.9°\times2.5°$) that needs to be transmitted

for each communication, and determined to establish a data buffer on a high-speed solid-state drive to ensure a balance of performance and compatibility. It takes about $1\times10^{-2}$ seconds to access the data buffer in each time step, which is enough to support the efficient simulation of NNCAM. The Simulation Years per Day (SYPD) of NNCAM based on DNN-GCM Coupler has a performance improvement of nearly 30x compared to SPCAM, and reaches half the speed of CAM5.

## 3 Offline Validation of NN-Parameterization

To assess how well the NN-Parameterization learns the subgrid tendencies from the CRM and its effects on radiation in SPCAM, we performed offline testing with a realistically configured SPCAM from January 1997 to December 1998, where NN-Parameterization is diagnostically run paralleled to the CRM in SPCAM. The results over the entire second year of 1998 are chosen for evaluation. As suggested in Han et al. (2020) and Mooers et al. (2021), faced with the diverse realistic boundary





condition, it is necessary to conduct such evaluation before prognostic experiments. We choose mean fields and coefficient of determination ($R^2$) as the two metrics in the offline testing.

The mean diabatic heating and drying rates produced by convection and large-scale condensation in SPCAM and NN-Parameterization are in close agreement. Figure 6 shows the latitude-height cross-sections of the annual mean heating and moistening rates in SPCAM and the corresponding NN-Parameterization. At 5 °N, SPCAM shows maximum latent heating in the deep troposphere, corresponding to deep convection at the ITCZ. In the subtropics, there is heating and moistening in the 290 lower troposphere, corresponding to stratocumulus and shallow convection in the subtropics. In the midlatitudes, there is a secondary heating maximum below 400 hPa due to midlatitude storm tracks. All these features are well reproduced by NN-Parameterization. Note that in the midtroposphre, the ITCZ peak in the drying rates is slightly weaker in NN-Parameterization compared with that of SPCAM (Figure 6c and 6d).

In addition to the mean fields, the high prediction skill of NN-Parameterization is also shown in the spatial distribution 295 of $R^2$. We choose pressure-latitude cross-sections to better demonstrate $R^2$ for the 3D variables such as diabatic heating and moistening. Therefore, zonal averages are calculated in advance. For diabatic heating, $R^2$ is above 0.7 over the entire mid to low troposphere, and the high skill regions with $R^2$ greater than 0.9 concentrates in low levels but are extended to mid-troposphere in storm tracks (Figure 7a). As for the moistening rate, the high skill zones concentrate in the mid to upper troposphere (Figure 7b), leaving low skill areas below. Those regions with low accuracy generally locate in the mid to low 300 troposphere in tropics and subtropics, corresponding to deep convection at ITCZ and shallow convection in subtropics.

The global distribution of $R^2$ for the derived precipitation is shown in Figure 8. Our NN-Parameterization shows a great prediction skill globally, especially in the midlatitude storm tracks. The prediction skill is relatively low in many areas between 30°S to 30°N and some midlatitude continents. In particular, the prediction skill of precipitation is not ideal in the ITCZ deep convection regions. Moreover, for shallow convection in Subtropical Eastern Pacific and Subtropical Eastern Atlantic, the 305 precipitation prediction skill hits bottom, corresponding to the subtropical low skill zones for moistening rate (Figure 7b).

Generally, NN-Parameterization shows high performance in the offline testing regarding mean fields and fitting $R^2$. As suggested in previous studies, manually tunned fully-connected neural networks often fail the mission of fitting variables in realistically configured simulations (Mooers et al., 2021). NN-Parameterization succeeds in fitting diabatic heating and moistening rate, which suggests that, with a multi-target framework and the implementation of residual blocks, a well-designed 310 DNN has a great potential of serving as a replacement of convection and cloud parameterization. Low accuracy of DNN prediction in subtropical shallow convection areas is a great challenge for machine learning emulation of moistening rate and precipitation. Similar results are noted in previous studies (Gentine et al., 2018 & Mooers et al., 2021). However, even with this shortcoming, NN-Parameterization still manages to carry in multi-year prognostic simulations with reasonable results shown in the coming Section 4.





## 4 Long-term Prognostic Validation

As described in Section 2.2, NN-Parameterization is coupled in NNCAM to replace the conventional deep and shallow convection, microphysics and macrophysics, and radiation parameterizations. At the same time, the planetary boundary layer diffusion, as well as the dynamic core are still kept in the host GCM. Starting with the SPCAM checkpoint on July 1, 1998, NNCAM is run for 9 years and a half to December 31, 2007. After a spinup for half a year, we choose the next four years from January 1, 1999 to December 31, 2002 for evaluation. Although long-term stable run of NNCAM is achieved, a very slow climate drift is still inevitable. Thus, we do not put the last 5 years into analysis. In addition, as the referencing target model, SPCAM is only run from January 1, 1997 to December 31, 1999 due to excessive computing resources consumption. We gather the results of the last two years to compare with NNCAM simulations. CAM5, as the coarse-grided conventional parameterized control model, is run from January 1, 1998 to December 31, 2002. Similarly, the first year is for spinup and the last four years for analysis. In analysis of prognostic results, the following are selected for demonstration of climatology and variability: multi-year mean fields of temperature and humidity, precipitation, and the Madden Julian Oscillation.

### 4.1 Vertical profiles of temperature and humidity

In this section, we first evaluate the vertical structure of the mean temperature and humidity. Figure 9 presents the zonally averaged vertical profiles of air temperature and specific humidity as simulated by the NNCAM and the CAM5, in contrast to the SPCAM simulations. Overall, the NNCAM does an excellent job in reproducing the thermal structure. The resulting mean state in temperature and specific humidity of NNCAM closely resembles SPCAM throughout the troposphere. The larger deviations are temperature biases in the tropopause, where the cold-point region is thinner and warmer in NNCAM than in SPCAM and CAM5. In addition, there are cold biases above 200 hPa over polar regions and slight dry biases over the equator in NNCAM.

### 4.2 Precipitation

Figure 10 shows the spatial distributions of winter (December-January-February) and summer (June-July-August) mean precipitation simulated by SPCAM, NNCAM, and CAM5, with SPCAM simulation results as reference precipitation. In SPCAM (Figure 10a and 10b), massive precipitation can be found in regions of Asian monsoon and midlatitude storm tracks over the northwest Pacific and Atlantic oceans. In the tropics, the primary peaks of rainfall are in the eastern Indian Ocean and Maritime Continent regions. Furthermore, two zonal precipitation bands are located at 0°–10°N in the equatorial Pacific and Atlantic oceans, constituting the northern ITCZ. The southern South Pacific Convergence Zone (SPCZ) is mainly located around 5°S–10°S near the western Pacific warm pool region and experiences a southeast tilt as it extends eastward into the central Pacific. The main spatial patterns of SPCAM precipitation climatology are properly reproduced by both NNCAM and CAM5. In NNCAM, strong rainfall centers are well simulated over the tropical land regions over Maritime Continent, the Asian monsoon region, and South America and Africa (Figure 10c and 10d). In addition, the heavy summertime precipitation



over the northwestern Pacific is well represented in both SPCAM and NNCAM (Figure 10b and 10d). In CAM5, there is too little precipitation over that area (Figure 10f), which is a common bias in many GCMs. However, the equatorial region is too dry in NNCAM, especially over sea surface area in boreal winter (Figure 10c).

Figure 11 shows the annually averaged zonal mean precipitation from SPCAM, NNCAM, and CAM5. NNCAM generally
reproduces very similar latitudinal variations to that of SPCAM, but with weaker precipitation intensity near the equator. Precipitation of both NNCAM and CAM5 agrees with SPCAM in global annual averages (Figure 11a). NNCAM results even come closer with SPCAM than CAM5 in boreal summer (Figure 11f). In ANN and DJF average, precipitation of NNCAM is underestimated over tropical continents but overestimated in subtropical land regions compared with SPCAM targets (Figure 11d, e). When the ocean areas are included, the SPCZ in NNCAM is excessively separated from the ITCZ in boreal winter. Its
precipitation center south shifted, resulting in a minimum zone of equatorial precipitation (Figure 11b). NNCAM shows moderate precipitation prediction biases in some latitudes in the tropics which are corresponding to the low skill regions of tropical and subtropical moistening rate and rainfall in Section 3. In our speculation, the weaker drying tendencies of the ITCZ midtroposphre from the NN parameterization causes separated convergence zones, leading to underestimation of equatorial rainfall rate.

Moreover, NNCAM shows better performance in simulating precipitation extremes. Figure 12 shows the probability densities function of simulated daily precipitation in the tropics (30°S−30°N) with a precipitation intensity interval of 1 mm day$^{-1}$. In CAM5, the frequency of light rainfall events with values smaller than 1 mm day$^{-1}$ is lower than that in SPCAM, and heavy precipitation events exceeding 20 mm day$^{-1}$ are insufficient. Compared with CAM5, the spectral distribution of precipitation in NNCAM is much closer to SPCAM. Both light and heavy rainfall events are substantially enhanced, and the
overestimated precipitation occurrence between 2−20 mm day$^{-1}$ is reduced. In addition, NNCAM avoids the unreal probability peak around 10 mm day$^{-1}$ appeared in CAM5, which is a common simulation bias found in simulations with parameterized convection but not in explicitly resolved convections (Holloway et al., 2012).

### 4.3 The MJO

The MJO is a crucial tropical intraseasonal variability at the time scale of 20–100 days (Wheeler and Kiladis, 1999). Figure
13 presents the wavenumber and frequency spectra for equatorial precipitation anomalies from SPCAM, NNCAM, and CAM5 in boreal winter. SPCAM shows a concentration of power at zonal numbers of 1−3 and periods of 30−40-day for eastward propagation (Figure 13a). In NNCAM, there is a spectral peak at the wavenumbers of 1−2 and longer periods of greater than 80 days (Figure 13b). There is also a second spectral peak with comparable wavenumbers at 50−80-day periods for eastward propagation, exhibiting intense intraseasonal signals. For CAM5 (Figure 13c), the spectral power is concentrated around 30-
day and more extended periods (greater than 80 days) at wavenumber 1 for eastward propagation. In addition, CAM5 also shows signals of westward propagation around 30-day period. Compared with CAM5, NNCAM shows stronger intraseasonal power and resembles SPCAM better.





The MJO is characterized by the eastward propagation of deep convective structures with an average phase speed of around 5 m s$^{-1}$ along the equator. Generally, it generally forms over the Indian Ocean, strengthens over the Pacific, and weakens

in the eastern Pacific due to interaction with cooler SSTs (Madden and Julian, 1971). Figure 14 presents the longitude-time lag evolution of 10°S−10°N averaged intraseasonal precipitation anomalies. The results show that both NNCAM and CAM5 reasonably reproduce the eastward propagating convection from the Indian Ocean across the Maritime Continent to the Pacific (Figure 14b and 14c). NNCAM captures the key MJO propagation simulated in SPCAM. The average phase speed of eastward propagation of deep convection in NNCAM is much closer to 5 m s$^{-1}$ than the overly fast propagation speed in CAM5, denoted

by the dashed line in Figure 14b and 14c.

## 5 Conclusions and Discussion

This study investigates the potential of DNN based parameterizations embedded into SPCAM in reproducing long-term climatology and climate variability. We present NN-Parameterization, a group of organized ResMLPs, to emulate convection, cloud processes and radiation in a SPCAM with a realistic global land-ocean distribution. The input variables to the NN-

Parameterization include specific humidity, temperature, largescale water vapor and temperature forcings, surface pressure and solar insolation. The output variables of the NN-Parameterization consist of the subgrid tendencies of moisture and temperature as well as radiation fluxes. The output variables are divided into multiple groups, and each group is trained as one target through independent neural network following the same architecture. Both long-term stable and high performance climate simulations are finally obtained in this work. To effectively bridge the host CAM and NN-Parameterization, we have

expanded the coupler idea in a earth system model to ensure the flexibility of embedding DNN into GCM. As a result, we can efficiently use DNNs to construct NN-Parameterization that can support NNCAM stable simulation for multi-years. This study is the first attempt to achieve stable climate simulations using a hybrid ML-physical GCM, which is configured with real land-ocean distributions.

The offline test shows the great skills of the NN-Parameterization in emulating the CRM and CRM radiation in SPCAM.

The overall diabatic heating and drying rates in the NN-Parameterization and SPCAM are in close agreement. When implemented in the host coarse-grided CAM5 to replace the conventional schemes of moist physics and radiation, the NN-Parameterization successes in an extensive long-term stable prognostic simulation and performs well in reproducing the mean vertical structures in temperature and humidity, and the precipitation distributions. The prominent MJO signal and its phase speed are also well captured using our NN parameterization.

Machine learning parameterizations implemented in aqua-planet configured 3D GCM have been well studied in many previous research. Some faced instability in coupled simulations (Brenowitz & Bretherton, 2019), and some tried to solve such instability through online learning (Rasp, 2020). Some others achieved in long-term stable prognostic simulations with deep fully-connected neural networks (Rasp et al., 2018; Yuval et al., 2021) as well as random forest (Yuval & O'Gorman, 2020). In contrast to aqua-planet simulations, the spatial heterogeneity is prominent over land in GCMs which are configured with





real-geography boundary conditions. In this case, the conventional MLPs have been unable to fit CRM outputs (Mooers et al., 2021). The convection, clouds, and the interacted radiation of the CRM together with real-geography boundary conditions are no doubtfully far more complicated and nonlinear than in idealized models. We first propose a multi-target DNN method to control nonlinearity, reduce gradient directions cancellation during training, and make each output variable reach the best fit state. Moreover, we design ResMLP with sufficient depth to further improve the nonlinear fitting ability of NN-

Parameterization. A trial-and-error method is further presented to effectively find the optimal neural network for NN-Parameterization. By doing so, the NN-Parameterization we constructed achieved good fitting accuracy in offline testing and successfully carried out stable online simulations for multiple years.

Embedding deep neural networks into Fortran based atmospheric models is still a handicap. Before this study, researchers mainly used hard coding to build neural networks (Rasp et al., 2018; Brenowitz and Bretherton, 2019). An easier way is to use

Fortran based neural network libraries that can flexibly import network parameters (Ott et al., 2020). These methods have successfully implemented DNN in GCM, but they can only support dense layer based DNN. As a result, developers cannot take advantage of the most advanced neural network structures such as convolution, shortcut, self-attention, variational autoencoder, etc., to build powerful DNN based Parameterizations. In this research, through DNN-GCM Coupler, NN-Parameterization can support the mainstream GPU-enabled machine learning frameworks. Thanks to the simple and effective

implementation of the DNN-GCM Coupler, our NNCAM achieves 30 times SYPD compared to SPCAM by using four 14-layer deep ResMLPs in NN-Parameterization, although these DNNs are much deeper than the previous state-of-the-art fully-connected NNs in this field.

Different from the existing works, we find that high fitting accuracy of NN-Parameterization is necessary but not sufficient for both high performance and long-term simulations since neural networks with high fitting accuracy may crash

before achieving years of stable simulation. The mechanism of the observed early-crashed simulations needs to be further investigated. We obtained the optimal NN-Parameterization that can perform both multi-year and high performance simulations by trial-and-error as an initial attempt. In addition, we observe that NN-Parameterization has spatially non-uniform accuracy, such as relatively low fitting accuracy in tropical deep convective regions and subtropical shallow convection and stratiform cloud regions. Such problems have also been reported in previous studies (Gentine et al., 2018; Mooers et al., 2021).

We believe that a NN parameterization with heterogenous characteristics across different regions, rather than a globally uniform scheme, can further improve the fitting accuracy in this tropical and subtropical region.

*Code and data availability*. The source codes of SPCAM version 2 can be accessed at https://svn-ccsm-release.cgd.ucar.edu/ model_development_releases/spcam2_0-cesm1_1_1. The associated codes of the NN parameterization scheme have been arc

hived and made publicly available for downloading from https://doi.org/10.5281/zenodo.5335929.

*Author contributions*. XW trained the deep learning model, constructed the DNN-GCM Coupler, performed the NNCAM and CAM5 experiments, and wrote the main part of the paper. YH conducted the SPCAM simulations, offered valuable suggestions



on the development of the NN parameterization, and participated in the writing of the paper. WX supervised this work,
provided critical comments on this work and participated in the writing of the paper. GJZ provided key points for this research
and participated in the revision of the paper. GWY supported this research and gave important opinions. All the authors
discussed the model development and the results.

*Competing interests.* The authors declare no conflict of interest.

*Acknowledgements.* This work is partially supported by the National Key R&D Program of China (grant no. 2017YFA0604500)
and the National Natural Science Foundation of China (grant no. 42130603). We thank Prof. Yong Wang for his guidance on
SPCAM simulations and valuable discussions on this work.

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

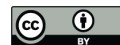



**Table 1**. Input and output variables. For inputs, $q_v(z)$ denotes the vertical profile of water vapor. $T(z)$ is the profile of temperature, and $dq_{v\,l.s.}(z)$ and $dT_{l.s.}$ are the large scale forcing of water vapor and temperature, respectively. $P_s$ is the surface pressure and *Solin* is the TOA solar insolation. For outputs, $dq_v(z)$ and $dT(z)$ are the tendencies of water vapor and temperature due to moist physics and radiative processes calculated by the NN-Parameterization. The net longwave and shortwave fluxes at the surface and the TOA are surface net longwave flux (FLNS), surface net shortwave flux (FLNT), TOA net longwave flux (FLNT), and TOA net shortwave fluxes (FSNT). The 4

downwelling solar radiation including solar downward visible direct to surface (SOLS), solar downward near infrared direct to surface (SOLL), solar downward visible diffuse to surface (SOLSD), and solar downward near infrared diffuse to surface (SOLLD) are shortwave radiation fluxes reaching the surface.

| Inputs | Outputs |
|---|---|
| $q_v(z)$, $T(z)$, $dq_{vls}(z)$, $dT_{ls}(z)$, $P_s$, *Solin* | $dq_v(z)$, $dT(z)$, *FLNS, FSNS, FLNT, FSNT, SOLS, SOLL, SOLSD, SOLLD* |





**Table 2.** Training configuration for the baseline ResMLP

| Description | Value |
| --- | --- |
| Number of samples trained per iteration | 1024 |
| Strategies for Declining Learning Rate | coslr |
| Initial learning rate | 0.001 |
| Number of rounds to traverse the data set | 50 |
| Probability of randomly setting neurons to 0 | 0 |
| L2 regularization | 0 |





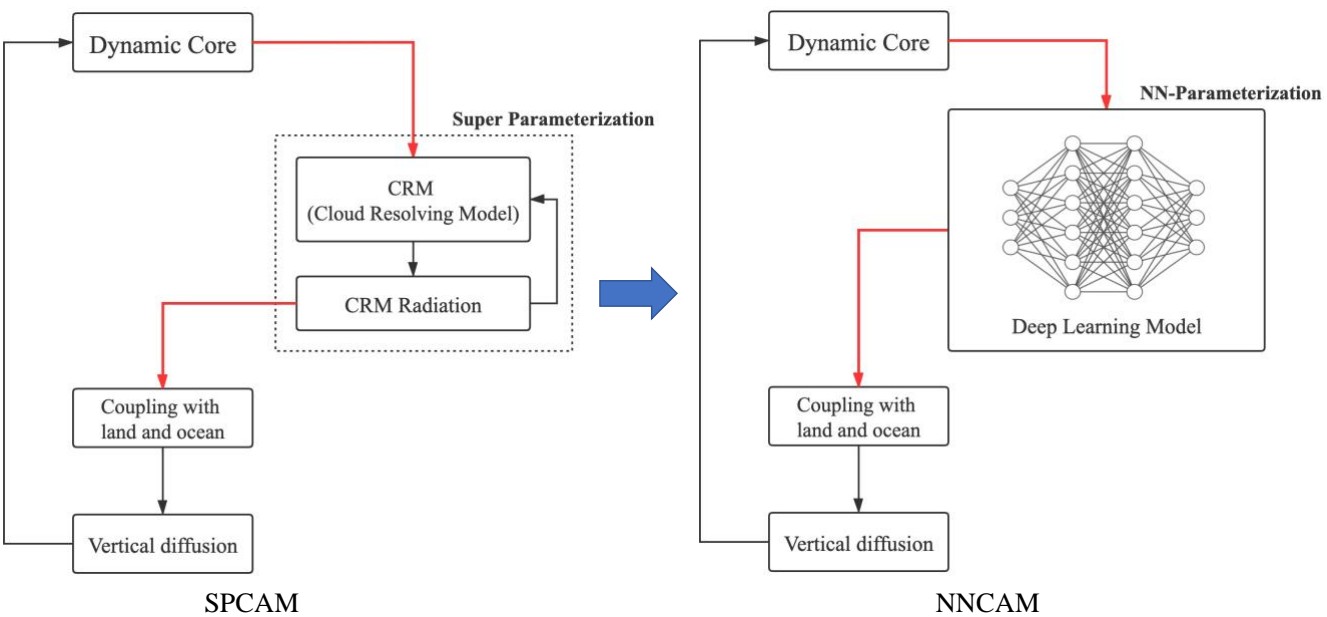

SPCAM

NNCAM

**Figure 1**. Workflow diagram of the NNCAM. In the NNCAM, both the CRM and CRM radiation in SPCAM are replaced
with the NN-Parameterization. All other model components, including the dynamic core, coupled land surface model and
prescribed ocean, and planetary boundary layer, remain the same as in SPCAM.





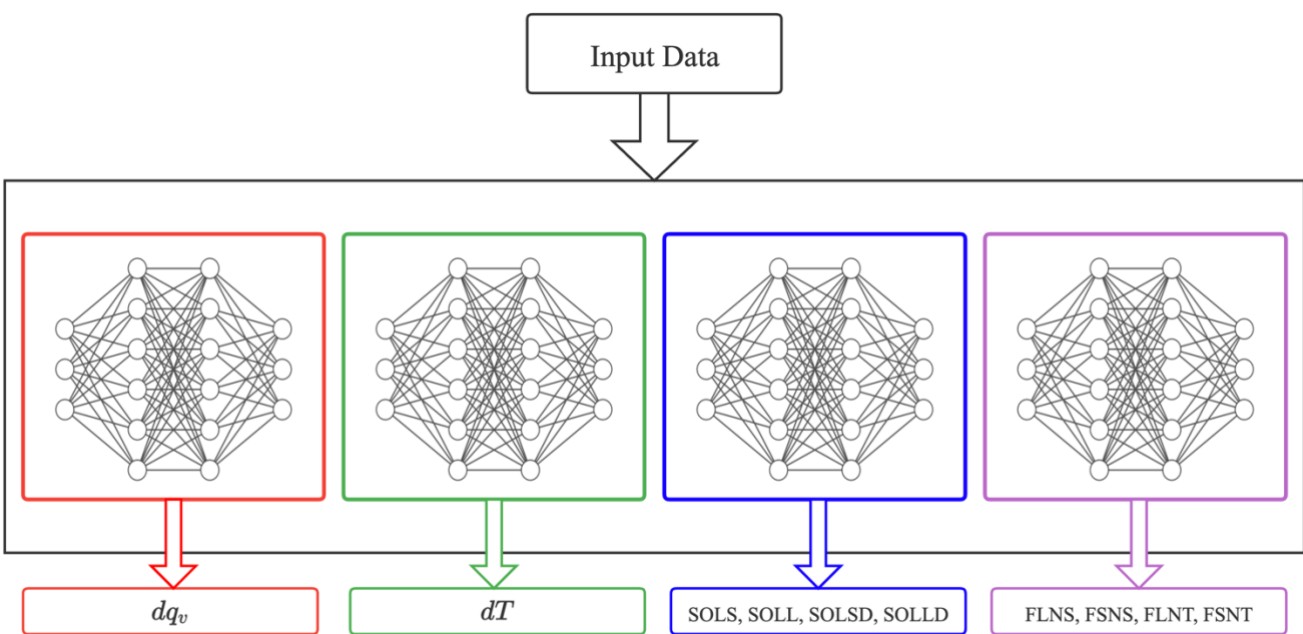

**Figure 2**. Schematic diagram of multi-target NN-Parameterization. It is composed of three neural networks, sharing unified inputs, to predict $dq_v(z)$, $dT(z)$, the downwelling solar radiation fluxes at the surface, and net radiation fluxes at the surface and the TOA respectively.



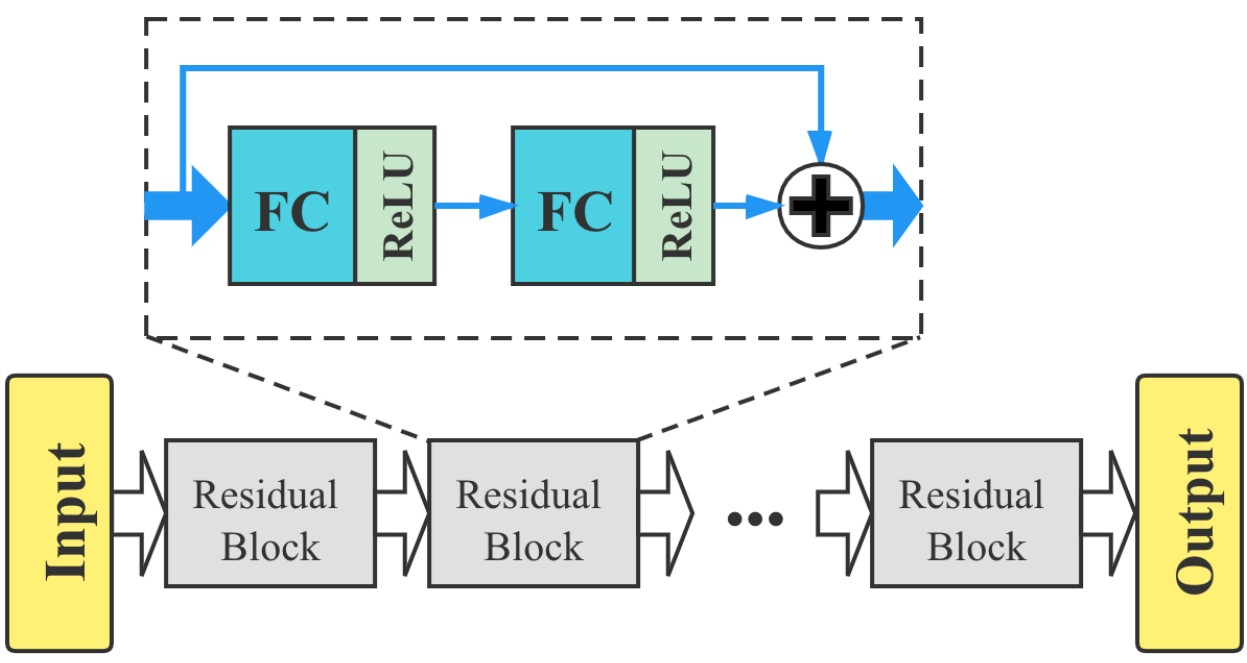

**Figure 3.** Schematic showing the structure of ResMLP. It consists of 7 residual blocks, each of which (shown in dashed box) contains two 512 node-wide dense (fully-connected) layers with a ReLU as activation, and a layer jump. The input and output are discussed in section 2.2.2.



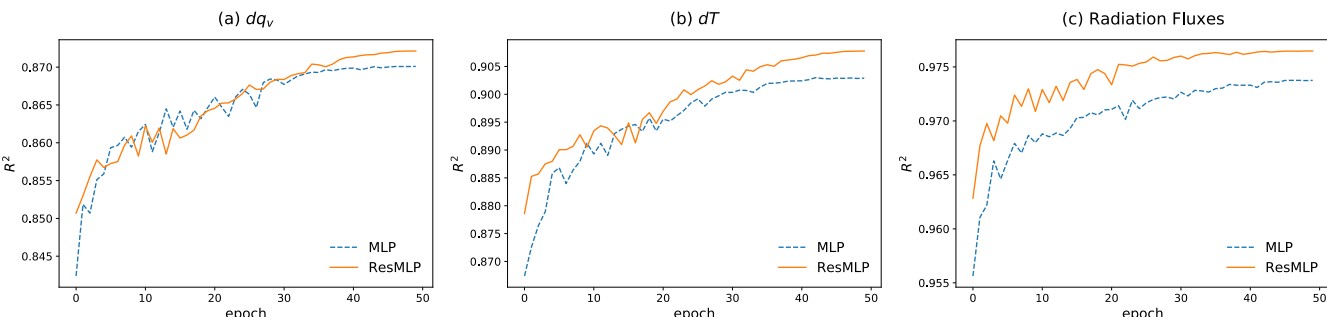

**Figure 4.** Fitting accuracies ($R^2$) of both the proposed ResMLP and MLP for different targets. (a) represents the fitting accuracy of moistening, (b) is the fitting accuracy of heating, and (c) shows the fitting accuracy of the average $R^2$ over the 8 radiation fluxes. Note: Spatial averaging of MSE is performed before calculating $R^2$.







**Figure 5**. A flow chart of NNCAM including DNN-GCM Coupler. NNCAM runs in the direction of the arrow, and each box represents a module. Among them, DNN-GCM Coupler is indicated by light red. NN-Parameterization is shown in the sub-figure on the right. Note: ① represents the dynamic core transmits data to DNN-GCM Coupler; ② and ③ represent the data communication between DNN-GCM Coupler and NN-Parameterization; ④ represents the host GCM accepts the result from NN-Parameterization.





**Figure 6.** Latitude-pressure cross sections of annual and zonal mean heating (top) and moistening (bottom) from moist physics during 1997−1998 for (a, c) SPCAM simulations, and (b, d) offline test by the DNN.





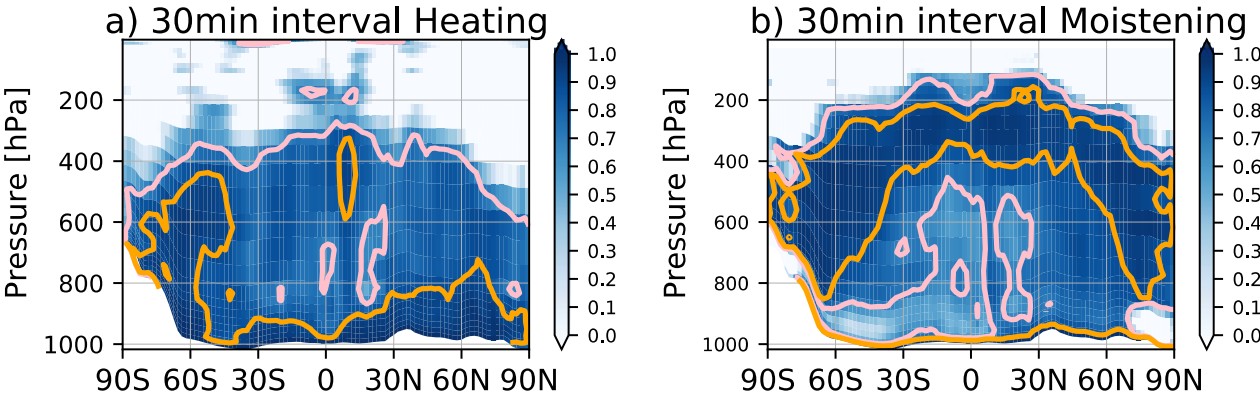

**Figure 7.** Latitude-pressure cross sections of coefficient of determination ($R^2$) for zonal averaged heating (a) and moistening
(b). They are predicted by NN-parameterization in the offline one-year SPCAM run, and are evaluated at 30-min timestep
interval. Note: areas where $R^2$ is greater than 0.7 are contoured in pink and those greater than 0.9 are contoured in orange.



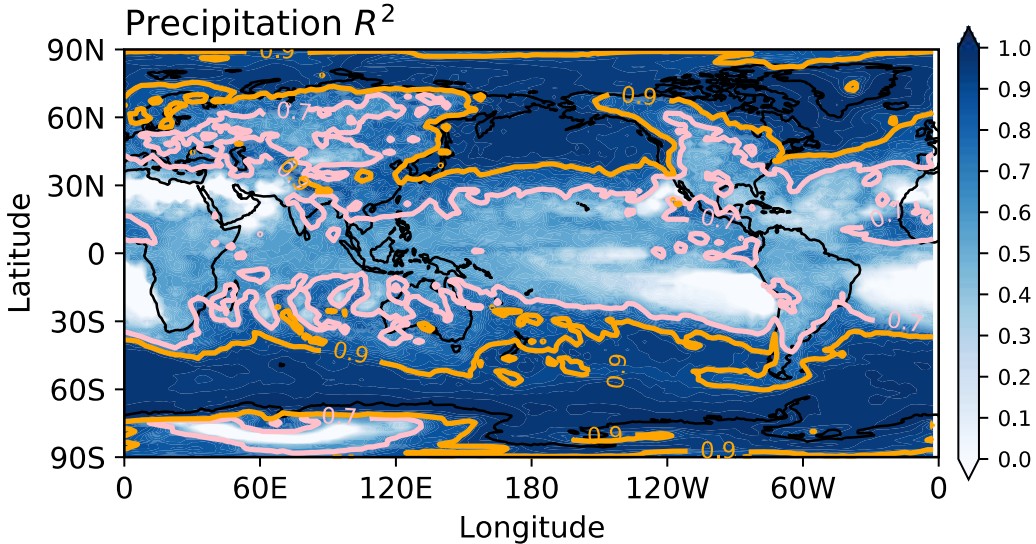

**Figure 8.** Latitude-pressure cross sections of coefficient of determination ($R^2$) for the derived precipitation predicted by NN-parameterization in the offline one-year SPCAM run. The predictions and SPCAM targets are in 30min timestep interval. Note: areas where $R^2$ is greater than 0.7 are contoured in pink and those greater than 0.9 are contoured in orange.





**Figure 9**. Latitude-pressure cross sections of annual and zonal mean temperature (left panels) and specific humidity (right panels) from (a, b) SPCAM (1998−2001), (c, d) NNCAM (1999−2003), and (e, f) CAM5 (1999−2003).

655





**Figure 10.** The mean precipitation rate (mm day$^{-1}$) of December-January-February (left panels) and June-July-August (right panels) for (a, b) SPCAM (1998−2001), (c, d) NNCAM (1999−2003), and (e, f) CAM5 (1999−2003).



**Figure 11**. The zonal mean precipitation rate (mm day$^{-1}$) averaged for (a, d) the annual mean, (b, e) December-January-February, and (c, f) June-July-August. Black, blue and red solid lines denotes SPCAM, NNCAM and CAM5, respectively.





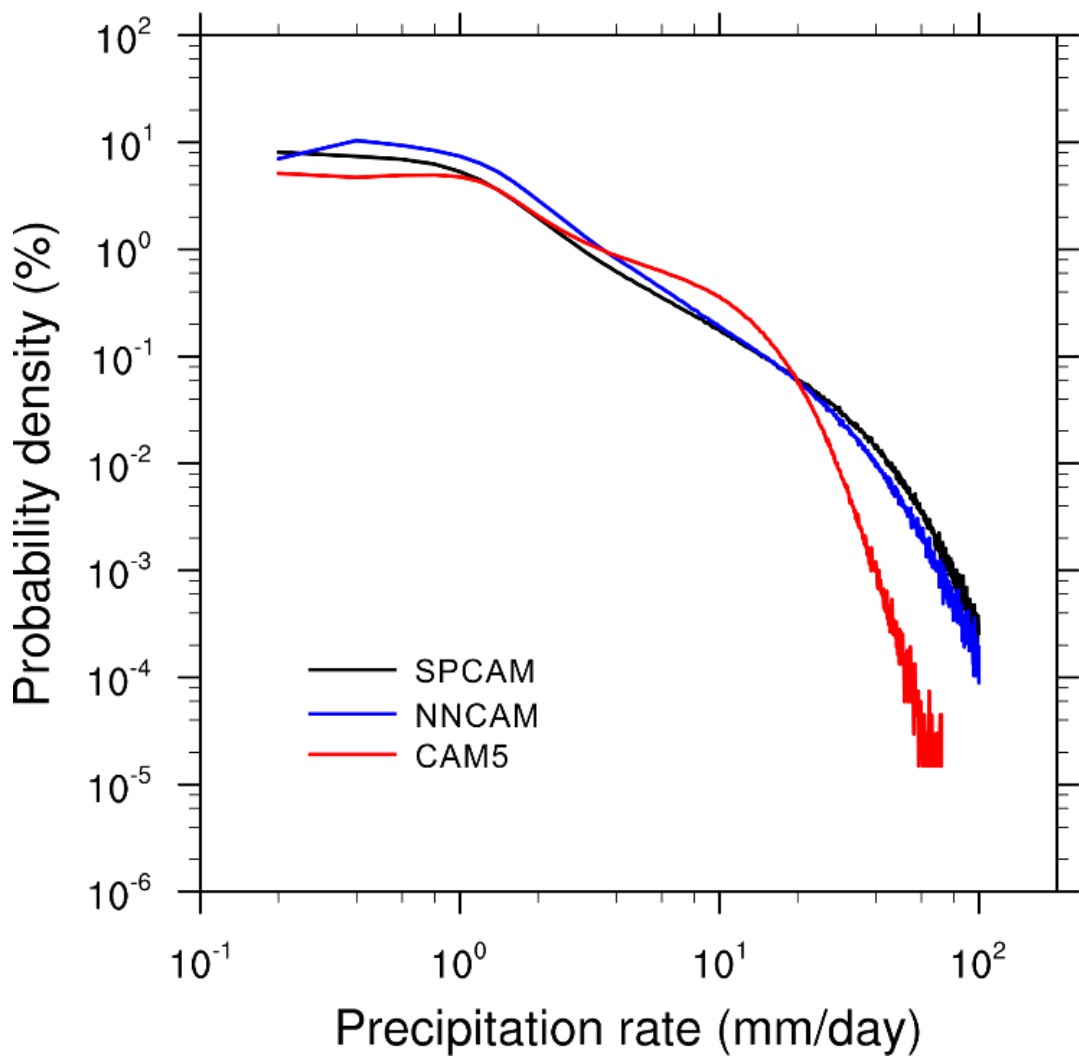

665    **Figure 12**. Probability densities of daily mean precipitation in the tropics (30°S−30°N) from the three model simulations.

Black, blue and red solid lines denotes SPCAM, NNCAM and CAM5, respectively.





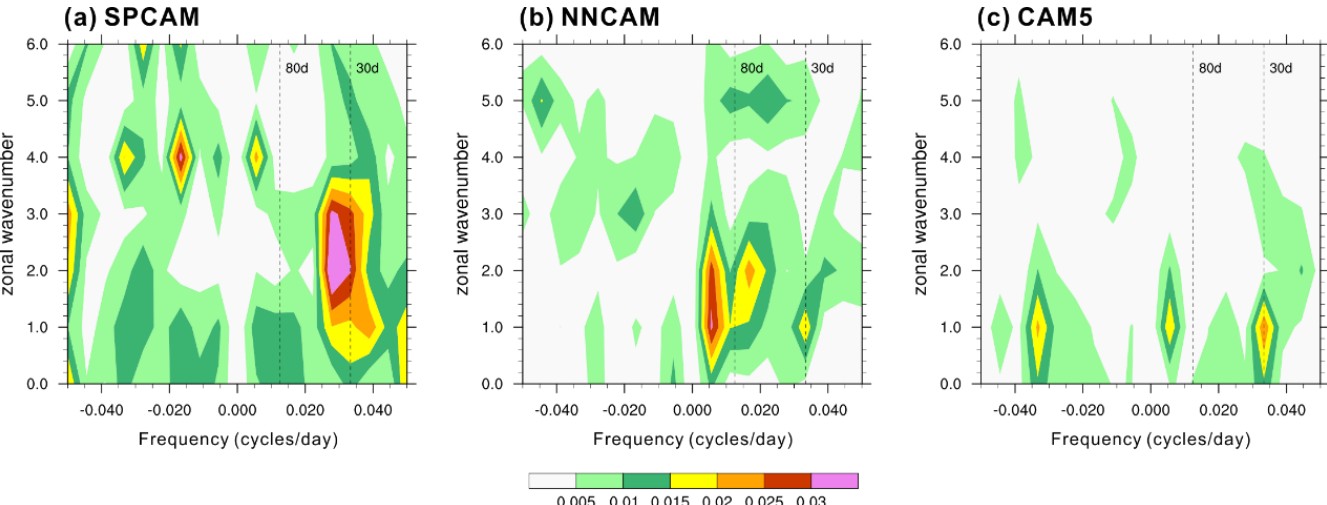

**Figure 13**. The wavenumber-frequency spectra of 10°S−10°N daily precipitation anomalies for (a, b) SPCAM, (c, d) NNCAM, and (e, f) CAM5 simulations for boreal winter.



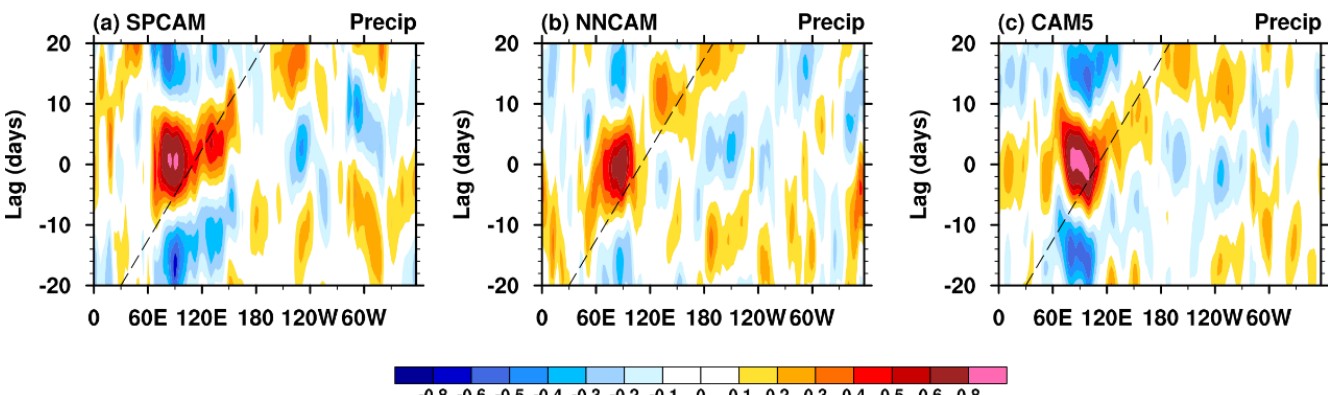

**Figure 14**. Longitude-time evolution of lagged correlation coefficient for the 20-100 d band-pass-filtered precipitation anomaly (averaged over 10°S−10°N) against regionally averaged precipitation over the equatorial eastern Indian Ocean (80E−100°E, 10°S−10°N). Dashed lines in each panel denote the 5 m s⁻¹ eastward propagation speed.