# Peer review of "Stable climate simulations using a realistic GCM with neural network parameterizations for atmospheric moist physics and radiation processes"

_Geoscientific Model Development, 2021_

## Referee Comment (RC1)

**1 Summary**

This paper describes the use of a neural network (NN) to emulate a super-parameterization, and the implementation of the NN parameterization in CAM with realistic boundary conditions. The authors develop a coupler between python and Fortran which allows them to easily use python-trained NNs during online CAM runs. The authors show that the architecture they use (fully connected with skip connection) combined with the idea of separate the prediction of different outputs to different networks improve the performance of the ML parameterization. The author test their trained NNs in an online CAM setting and find that some of their NN-parameterizations lead to unstable simulations, but some lead to stable simulation (although the one simulation that the results are show for has a climate drift).

I generally think that it is impressive that the authors have succeeded to run a global GCM with realistic boundary conditions with NN-parameteriztions, however I have several general comments on the manuscript that need to be addressed (and more detailed comments below):

- (a) It seems to me very disappointing that the authors do not provide any idea/hypothesis why some of their networks are stable and others are unstable. From what I understood, the strategy of the authors is to conduct an exhaustive random search for an accurate and stable parameterization. Personally, I think that from a scientific point of view this is not satisfactory. Furthermore, a previous study by Brenowitz et al. (2020) suggested a method to understand why certain parameterizations are unstable (they suggested that when the ML parameterizations are coupled to dynamics it may lead to unstable gravity waves) and certain actions that can be tested in order to remove such instability (e.g., input ablation), so I think it would be crucial to understand if there is an underlying physical reason why some parameterization go unstable (for example by testing if similar framework like Brenowitz et al. (2020) could really help in this case). Furthermore, unlike Mooers et al. (2020) that showed that stability of simulations is correlated with accuracy of the parameterization, other papers (e.g., Rasp et al. (2018), Yuval and O'Gorman (2020)) showed that even inaccurate parameterization can run stably for long periods. So I think it is still undetermined whether it is really necessary to have a very accurate parameterization scheme in order for it to run stably in a more realistic case. Therefore, I think it would be great if the authors could add their input on this since it seems that they did train also less accurate networks than the resMLP that they use. Namely, did all the non-resMLP networks were unstable? Overall, The achievement of running CAM with orography and a with a neural network parameterization is an important achievement that should be documented in the literature. However, from a scientific point of view, the contribution of this paper is a bit limited, and also it is not clear to me whether the NN parameterization performs better than traditional CAM parameterizations (in a few aspects it does, but in many other it does not).

- (b) I think that more work on the text is needed to bring it to a level that is appropriate for the journal (see comments below). There are multiple repetitions, and sometimes

sentences with little context.

- (c) There are multiple times that the authors cite works that are not relevant to what they claim. For me especially alarming since I am not familiar with many of the references in the paper, and I found such mistakes on a small subset of references that I am familiar with. Therefore, I am worried that also other references are not relevant to what they claim.

- (d) I think that some of the claims that are made in the manuscript are not supported by the results that are shown.

Overall, I think that the manuscript deserves publication but needs major revisions before it can be published.

**2   Comments**

- Abstract - I think that the authors overstate some of their outcomes (or at least do not show explicitly in the text these results; see more comments below). Some examples: I am not sure that in their parameterization outperforms CAM in terms of spatial distribution of precipitation and MJO propagation. Of the author want to state this, I think it is necessary to show the RMSE for the spatial distribution of precipitation and to show a quantitative comparison between the MJO spectrum/propagation in CAM, SPCAM and NNCAM.

- line 15 (and also through the abstract): the authors write that they learn from a cloud resolving model. This is confusing. I think that it is important to explain that they use SPCAM, and the task at hand is an emulation of a super-parameterization rather learning from a cloud-resolving output.

- line 38: I think that one important reference regarding the biasses that are caused by different parameterization is a paper by Wilcox and Donner (2007). I might be mistaken, but this is the only case that I know where they show how two different parameterization schemes in the same model lead to a very different frequency precipitation distribution.

- line 44: Is there a citation the authors can provide for supporting the sentence "Their interaction with the atmospheric circulation affects the transport and distribution of energy and is the largest source of precipitation biases". If not please remove.

- line 56: "The CRMs have been applied to low-resolution GCMs to replace conventional cumulus convection and cloud microphysical parameterization schemes" - the sentence is unclear

- line 57: Is there any other cases where CRMs were nested in GCM except super-parameteriztation (SP)? if not, please don't use "such as" as it implies there are other examples

- line 61: "an order of magnitude larger compared with that for CAM." -this really depends on the CRM configuration, right? Even in the authors simulation it seems that SPCAM uses much more than factor 10 compared to CAM.

- line 66: Strange sentence, and I think it has some grammatical errors (e.g., which "the data-driven parameterization scheme", from the sentence it seems you refer to a specific scheme, and I do not think it is the case)

- line 68: "More recently," -more recently than what?

- line 72: I cannot see why the citations of Schneider et al. (2017) or of Duben and Bauer are related here. There is no convection scheme learned in these papers.

- line 74: "trained ones" - unclear

- line 76: I do not see how Rasp (2020) citation is relevant here. This citation describes a method for online learning. Nothing related to an implementation of NN in GCM.

- line 85: "They found that minor changes, either to the training dataset or in the input/output vectors, can lead to model integration instabilities." - can you give a citation. I could not find such statement in the manuscript.

- line 89: There is a very relevant paper that the authors do not refer to. Brenowitz et al. (2020) have investigated both for SPCAM and for SAM why they can lead to numerical instabilities. Please add a discussion about it - and more importantly, I think that if the community could benefit from the this work it would be if the authors could identify why some of their networks are unstable (as was done by Brenowitz et al. (2020)).

- line 94-95: maybe instability and drift prevents the application some models, but other studies (that you cite) and also (Yuval et al., 2020) did not have any problem of unstable simulations. It is not fully determined why this is the case but the two possiblities that are raised in these papers are (a) because subgrid terms were calculated more accurately in these works compared to Brenowitz and Bretherton (2019), or (b) because these works succeeded to implement physical-constraints in the ML parameterizations).

- line 102: What is "auto-learning technique".

- line 104: Unclear what is "group of NNs" - do you mean different NNs for predicting different outputs? if yes, you should mention that this was already done by Yuval et al. (2020) (although in a different model).

- line 105: The authors write: "We apply two innovative methods in neural network models: multi-target training to achieve balanced results across diverse neural network outputs and multilayer perceptron with residual blocks (ResMLP) to enhance nonlinear fitting ability." To me it is unclear if what the authors write here is really

innovative: (a) multi-target training was already done by Yuval et al. (2020), although in a different model, so I could understand that it is a slightly different context. (b) If I understand correctly, Han et al. (2020) also used a DNN with shortcuts (called ResDNN in Han et al. (2020)) and showed it performs better than a standard DNN, which if I understand correctly is exactly what the authors call ResMLP

As a side comment, I think that the term multi-target training is unclear and should be modified.

- line 111: I understand that it would be difficult to integrate complicated NNs into Fortran, but DNN (or DNN with shortcuts) should be a very simple procedure in Fortran.

- line 115: I think it is very problematic not to mention here that the simulations are stable without mentioning that there is a climate drift (although relatively very slow one).

- line 145: Why is there a distinction between 2D CRM and CRM radiation? What does it mean CRM radiation?

- line 148-151: this is a repetition of things that were already written in the intro.

- figure 1: I might be missing something but for me these figures and the text that describe this figure is confusing. Aren't you just replacing SP with NN (like what was done in several other papers?). If yes I think that a very short and concise description will be much clearer.

- line 152-154: I am not sure on what the statement is based on. I can agree that the prediction is more difficult, but why it has more numerical sensitivities (I am not even sure what it means)

- line 161: the authors use the large-scale forcing as inputs. These were not used in previous papers trying to emulate SPCAM. It would be good to mention why they introduce these inputs, and if whether the inclusion of these inputs substantially improves the offline (or online) performance.

- line 164: The authors predict also different outputs compared to previous studies that used ML to emulate a SP. e.g., they do not predict the vertical structure of radiative heating, but do predict other quantities. It would be helpful to understand what is the reason that they use different outputs (and inputs), and explain to the reader the motivation for these choices (and highlight differences from previous attempts to learn ML parameterization from SPCAM).

- line 170: How does the authors deal with a negative precipitation (both offline and online)? Can you give information what is the percentage of (both online and offline samples) with negative precip?

- line 180: Random split for the train,validation and test set is not ideal (and not the common practice) due to the time correlation between samples. In order get a reliable results for the test set what usually is done is to take the samples for each of the data sets from different time intervals. Please do that and report offline performances when the test set is taken from a time interval that was used during training. Alternatively, make a justification for this choice.

- line 185: Is there some citation that can backup the statement that MLP can generalize better than other types of networks? If not please remove statement

- General comment: I might not understand what is the NN that you used. But if your multilayer perception is just a fully connected NN please say that, and preferably change the terminology to the common one (fully connected NN)

- line 186-192: Please add graphs supporting the statements (these graphs can also go in the supplementary). Also the last sentence is unclear to me and I am not sure how it is related to the rest of the text.

- line 193: The first sentence in the paragraph is unclear to me.

- line 193-194: I disagree with this sentence about the independence - see Yuval et al. (2020) where they use the dependencies between the tendencies of moisture and thermodynamic variable to predict only one of them, and diagnose the other (because of a 1 to 1 mapping between the two for parameterizations like microphysics and sedimentation).

- line 201: I am not sure why this statement is necessary, and how it fits in the manuscript.

- figure 3: To me it seems that the fully connected NN performs almost identically to the resMLP (roughly a difference of 0.002 in $R^2$). Did you test several fully connected NNs in an online setup and verified that using skip connections is really what makes the difference in terms of stable simulations? If not, please confirm that fully connected NNs do not lead to similar results (since if I understand correctly, the authors argue that this is one of the important aspects of their work). Furthermore, I think that the $R^2$ results that presented are confusing and not the relevant ones for the reader. $R^2$ should be calculated over different samples without any average before since such calculation gives the idea of the real performance of the network. So please change the way $R^2$ is evaluated

- line 213-214: On what are you basing your statement that if the NN is underfitting NNCAM crashes quickly? Did you test 100s of NNs? 1000s NNs? 1 NN? Please show a graph supporting this evidence (similar to the one that Ott did) if you want to keep this part of the text

- line 218-219: To me, this result (more accurate NNs that crushed earlier) indicates that the accuracy is not the important part of the stability. Previous work (e.g., Rasp et al. (2018) showed that also very shallow and less accurate NNs lead to stable NNCAM simulations in an aquaplanet setting). This raises a question why the resMLP is necessary and whether fully connected network could work as well.

- table 2: Please give the full details of the training (e.g., which optimized). Furthermore, I do not think that the data has to be shown in a table.

- lines 223-229: The division between the stable and unstable group is interesting and I think here would be necessary to use the tools developed by Brenowitz et al. (2020) to check stability. If these tools cannot explain the instabilities in the simulations you are conducting, it is important for the community to know this.

- line 249: The authors claim several times during the manuscript that it is difficult to code in Fortran an NN. However, a fully connected layer is very easy to implement as it involves only matrix multiplications so it would be good to clarify that it might be difficult to use some fancy architecture, but the basic one that the authors use is very simple to code in Fortran.

- line 249: "At runtime..." This is a confusing sentence and it is unclear to me why you cite the papers (as they use a different code infrastructure so I do not understand why these citations are included here)

- line 251: "outside", you need to explain outside of what.

- line 255: "In this..." this is repeating similar statements so I do not think it is necessary

- line 265-269: Sounds like a great achievement! (I haven't tried it myself though)

- line 278: Sounds great!

- line 296 and figure 7: Please show $R^2$ before zonal averaging! It does not make sense to me first zonal average (and also this was not done in previous work)

- figure 8: SP is especially important in the tropics, and it seems That the skill in the tropics is very low (many regions have $R^2$ close to 0). Can you give some insight?

- line 301: I think that it is difficult to determine whether this is a good job or not because there is no baseline to compare to. If you could provide a baseline from CAM (for offline prediction), then it would make sense to give this statement.

- line 306: The word "fitting" should not be there as far as I understand

- line 306-308: "As suggested..." This sentence is unclear to me. How do you manually tune an NN?

- line 309-314: I think that these sentences are not well related to each other.

- line 317-318: "At the same time...." If I remember correctly this is the default choice of SPCAM so it is very confusing you are writing this as if this is something special

- line 321: The fact simulation have a climate drift should be mentioned already in the abstract.

- line 321-325: Please compare NNCAM and CAM to the relevant simulated period of SPCAM. I do not understand why you compare different periods (and different length of time intervals).

- 324: Why here (CAM) 1 year spinup and for NNCAM - half a year? I guess that this is how it is initialized?

- 324: Is it critical to use SPCAM for initialization for NNCAM? if yes please explain why? Previous studies used a coarse run (in this case CAM ) to initialize the model which makes more sense to me.

- figure 9: Please show RMSE in the figure for each of the sublots - so we could compare NN to CAM. Also for other figures and claims made by the the authors - if the authors want to state that NNCAM performs better than CAM, please provide a quantitative metric for the comparison.

- figure 10: Need to quantify the accuracy - please show RMSE

- figure 11: NNCAM has less skill than CAM so it would be fair to clearly mention it. Furthermore, please use some metric for the comparison if such a comparison is made. The fact that CAM is closer to SPCAM than NNCAM in many of the fields should be also mentioned in the abstract (to me it is a bit misleading to only mention in the abstract that NNCAM performs better than CAM without stating that in many aspects CAM is better than NNCAM)

- figure 12: Would be good To avoid the noise in this plot by using bins that have similar distances in a log scale.

- figure 13: Can the authors quantify the distance between distributions? It is difficult for me to determine if NNCAM or CAM is closer to SPCAM, since NNCAM has a very different MJO structure compared to SPCAM. Furthermore, please add more contours and make sure that the colorbar isn't saturated (at the moment it seems saturated).

- line 383/figure 14: The comparison should be between SPCAM propagation and NNCAM propagation because SPCAM is the baseline. I do not see why using the value of 5 m/s is relevant. To my eyes it seems that SPCAM propagates faster than 5m/s and has similar propagation as CAM. Namely, I disagree that NNCAM propagation is closer to SPCAM propagation compared to CAM.

- figure 14: Can you describe on which data was that calculated for.

- line 395: instead of "GCM" use or "a GCM" or "GCMs"

- line 399-404: You cannot not mention the climate drift.

- line 425-427: Also I think that the relevant comparison would be how fast it would run without the coupler. As far as I understand it should work pretty fast also without the coupler (which uses also additional resources)- as the matrix multiplication operation in Fortran is optimized pretty well.

- General comment: The authors mention that they have a group of NNs that lead to stable simulations. Are all these NNs lead to similar online results? If yes, please mention this. I suggest showing the STD for precipitation and a couple of other fields among the different stable online simulations you achieved since it will show the reader that there is no "cherry picking" with the choice of the simulation you end up showing.

- references not ordered alphabetically in some cases which makes it more difficult to find them.

**References**

Brenowitz, N. D., T. Beucler, M. Pritchard, and C. S. Bretherton, 2020: Interpreting and stabilizing machine-learning parametrizations of convection. *arXiv preprint arXiv:2003.06549.*

Brenowitz, N. D., and C. S. Bretherton, 2019: Spatially extended tests of a neural network parametrization trained by coarse-graining. *Journal of Advances in Modeling Earth Systems*, **11**, 2727–2744.

Han, Y., G. J. Zhang, X. Huang, and Y. Wang, 2020: A moist physics parameterization based on deep learning. *Journal of Advances in Modeling Earth Systems*, e2020MS002076.

Mooers, G., M. Pritchard, T. Beucler, J. Ott, G. Yacalis, P. Baldi, and P. Gentine, 2020: Assessing the potential of deep learning for emulating cloud superparameterization in climate models with real-geography boundary conditions. *arXiv preprint arXiv:2010.12996.*

Rasp, S., 2020: Coupled online learning as a way to tackle instabilities and biases in neural network parameterizations: general algorithms and lorenz 96 case study (v1. 0). *Geoscientific Model Development*, **13 (5)**, 2185–2196.

Rasp, S., M. S. Pritchard, and P. Gentine, 2018: Deep learning to represent subgrid processes in climate models. *Proceedings of the National Academy of Sciences of the United States of America*, **115**, 9684–9689.

Schneider, T., S. Lan, A. Stuart, and J. Teixeira, 2017: Earth system modeling 2.0: A blueprint for models that learn from observations and targeted high-resolution simulations. *arXiv preprint arXiv:1709.00037.*

Wilcox, E. M., and L. J. Donner, 2007: The frequency of extreme rain events in satellite rain-rate estimates and an atmospheric general circulation model. *Journal of Climate*, **20 (1)**, 53–69.

Yuval, J., C. N. Hill, and P. A. O'Gorman, 2020: Use of neural networks for stable, accurate and physically consistent parameterization of subgrid atmospheric processes with good performance at reduced precision. *arXiv preprint arXiv:2010.09947.*

Yuval, J., and P. A. O'Gorman, 2020: Stable machine-learning parameterization of subgrid processes for climate modeling at a range of resolutions. *Nature Communications*, **11 (1)**, 1–10.

---

## Author Comment (AC4)

**Response to reviewer 1**

*This paper describes the use of a neural network (NN) to emulate a super-parameterization, and the implementation of the NN parameterization in CAM with realistic boundary conditions. The authors develop a coupler between python and Fortran which allows them to easily use python-trained NNs during online CAM runs. The authors show that the architecture they use (fully connected with skip connection) combined with the idea of separate the prediction of different outputs to different networks improve the performance of the ML parameterization. The author test their trained NNs in an online CAM setting and find that some of their NN-parameterizations lead to unstable simulations, but some lead to stable simulation (although the one simulation that the results are show for has a climate drift).*

We thank the reviewer for his/her throught review of the manuscript. We appreciate many of the constructive comments and suggestions. Below is our point-by-point response. The reviewer's comments are in italic and our response is in normal and blue font. We have considered them carefully in the coming revised manuscript. Please note that the Figures 1-14 are in the revised manuscript, the Figures S1-S4 are in the supplementary, the Figures R1-R5 in the last part of this response, and the Movies S1 and S2 are in the attached files. For your convenience, we also put all related figures in the last part of the response.

*I generally think that it is impressive that the authors have succeeded to run a global GCM with realistic boundary conditions with NN-parameteriztions, however I have several general comments on the manuscript that need to be addressed (and more detailed comments below):*

- *It seems to me very disappointing that the authors do not provide any idea/hypothesis why some of their networks are stable and others are unstable. From what I understood, the strategy of the authors is to conduct an exhaustive random search for an accurate and stable parameterization. Personally, I think that from a scientific point of view this is not satisfactory. Furthermore, a previous study by Brenowitz et al. (2020) suggested a method to understand why certain parameterizations are unstable (they suggested that when the ML parameterizations are coupled to dynamics it may lead to unstable gravity waves) and certain actions that can be tested in order to remove such instability (e.g., input ablation), so I think it would be crucial to understand if there is an underlying physical reason why some parameterization go unstable (for example by testing if similar framework like Brenowitz et al. (2020) could really help in this case). Furthermore, unlike Mooers et al. (2020) that showed that stability of simulations is correlated with accuracy of the parameterization, other papers (e.g., Rasp et al. (2018). Yuval and O'Gorman (2020)) showed that even inaccurate parameterization can run stably for long periods. So I think it is still undetermined whether it is really necessary to have a very accurate parameterization scheme in*

*order for it to run stably in a more realistic case. Therefore, I think it would be great if the authors could add their input on this since it seems that they did train also less accurate networks than the resMLP that they use. Namely, did all the non-resMLP networks were unstable? Overall, The achievement of running CAM with orography and a with a neural network parameterization is an important achievement that should be documented in the literature. However, from a scientific point of view, the contribution of this paper is a bit limited, and also it is not clear to me whether the NN parameterization performs better than traditional CAM parameterizations (in a few aspects it does, but in many other it does not).*

**Reply:**

Thank you for your insightful comments. We agree with the reviewer's point on providing possible ideas/hypotheses on stabilizing the NN parameterization, which would greatly benefit the scientific community. Since there are several questions in Subsection (a), we respond to them in the following order: neural network introduction, prognostic stability versus offline validation loss, gravity waves related to Brenowitz et al. (2020), and climate evaluation.

To determine the necessity of very accurate NN parameterizations and the architecture of ResMLP/ResDNN, we plan to show the prognostic performance versus offline validation accuracy of both ResMLP/ResDNN and fully connected neural networks which represent the non-ResMLP networks. All neural networks are trained for epochs with the same dataset. Small training epoch numbers are for less accurate NNs and large epochs for well-trained and accurate NNs.

First, we briefly list the clarifications of NN design, terminology and training. The ResMLP proposed by our study is a deep residual fully connected neural network. It is 512-node wide and 14-layer deep with an identity shortcut between every 2 layers. For the non-ResMLP networks, we introduce a 7-layer deep fully connected neural network with the same width. To avoid confusion, we apply the terms in Han et al. (2020) and respectfully change ResMLP and MLP in the manuscript to ResDNN and DNN. We think 7-layer is a proper depth for a plain DNN since the degradation problem will occur when stacking more layers (He et al., 2016). However, with the help of identity shortcuts, ResDNN can go much deeper and gain higher accuracy (Han et al., 2020). For other types of ML architectures, random forest is less likely to perform as accurately as neural networks and has not yet been well implemented in GPUs (Yuval et al., 2021). 1D convolution neural network (CNN) seemingly performs accurately offline (Han et al., 2020), but their prognostic performance is still unknown. A recent article by Chantry et al. (2021) found the local connection of CNN layers may not be suitable for emulating deep vertical connections. Therefore, we only compare the detail performance of our ResDNN with that of the DNN with dense and global connections between adjacent layers.

The training dataset used by all considered NNs is 40% temporally random sampled from the 2-year SPCAM simulation from 1997-01-01 to 1998-12-31. Notably, the

random sampling is only done in the time dimension but not in latitude and longitude. As a result, the training dataset includes 13,824 samples across global grid points for each selected time step. Finally we have 97 million samples in total. To avoid any mix or temporal connection between the training and offline validation set, we random sample 40% timesteps from the SPCAM simulation in 2000 as the offline validation set. All networks begin with randomly initialized weights and biases and are trained from 5 epoch to 50 epochs with a learning rate of 0.001 and an optimizer Adam. The training loss function is the mean squared error of the selected variables (moisntening rate, heating rate, and 8 radiation fluxes) as $MSE = \|y_{NN} - y_{target}\|_2$, where $y_{NN}$ is the NN prediction and $y_{target}$ is the target SPCAM simulation.

The prognostic tests of NN Parameterization begin at 1998-01-01 as startup. The half-a-year spin of SPCAM is not needed but calling the CRM in SPCAM at the first step is required to generate the correct large scale forcings as the NN Parameterization input.

Our work proposes a ResDNN set, where each neural network is responsible for the prediction of a class of variables as described in section 2.2.2 of the revised manuscript. The validation $R^2$ of all 8 radiation fluxes is above 0.98 for ResDNN. We believe the neural network of radiation prediction is highly accurate and well trained. To simplify the neural network choices, we do not change it in the following experiments. Different from the easily and accurately trained radiation fluxes, the tendencies of both temperature and moisture are rather difficult to train and, if not trained well or not with the right NN architecture, can seriously affect the prognostic performance. Multiple ResDNN pairs ($dqv$ and $dT$) and 3 DNN pairs are brought in this discussion with different offline validation accuracy. To evaluate dqv and dT in one metric, we introduce the MSE of moist static energy changing rate ($dh = C_p dT + L_v dq_v$) as:

$$MSE_h = \left\| \frac{1}{g}(dh_{NN} - dh_{SPCAM})\Delta p \right\|_2,$$

Where $g$ is the gravity constant, $C_p$ refers to the heat capacity of air, $L_v$ is the latent heat of water vapor, and $\Delta p$ is the layer thickness.

Figure 4 shows the offline validation $MSE_h$ versus the number of prognostic steps that our NNCAM can run. First, the DNN parameterizations are less accurate than the ResDNN ones in terms of offline validation accuracy. As a results, all the DNN parameterizations cannot run stably longer than half a year in prognostic tests. For the ResDNNs (blue dots and black inverted triangles), the less well-trained ones with high MSE crash within half a year simulation. However, when the offline MSE of ResDNN decreases to a certain level (e.g., $290 W^2/m^4$), the ResDNN parameterization may run stably for long periods. In Figure 4, we observed 4 ResDNNs can run stably.

Further analysis is conducted following the reviewer's suggestion. We included the framework of coupling a NN scheme into large scale dynamics introduced by Brenowitz et al. (2020). The stability is examined by the time evolution of the average energy of several stable and unstable runs (Figure 5). The system energy grows

exponentially and then blows up with unstable ResDNN parameterizations (the red and orange lines), while the stable ones can maintain the total energy at a certain level and reproduce the annual cycle fluctuations like SPCAM. Among the stable ResDNN schemes, some can get nearly a perfect reproduction of the total energy evolution of SPCAM (the blue line), while some inaccurately simulate the climate state with a deviation (the green line).

The fast-growing energy of the unstable runs indicates that certain unrealistic perturbations get amplified by NN coupled dynamic system, thereby can be directly identified by the wave spectra analysis. Following Brenowitz et al. (2020), we choose the large scale profiles of temperature and moisture of the earliest breaking point as the background profiles for the 2D gravity wave. We get such a point by picking the first unstable wave initial position (Movie S1), which is around step 2270 step in the maritime continent in an unstable simulation with an offline accurate ResDNN (the red line in Figure 5). After an unreal perturbation is triggered, it rapidly propagates to nearby regions with an intense mid-level moistening/drying. Due to complex background winds and the topography below, the wave is not perfectly round. Unlike the mid-latitude breaking points in the aquaplanet experiment in Brenowitz et al. (2020), we find the tropics are of importance where most unstable waves get trigged and propagate in a realistic configured GCM. The barotropic atmosphere in the tropics, rather than the baroclinic mid-latitudes, is more similar to the ideal 2D gravity wave model. Based on the background profiles of the breaking point, the spectra analysis for the unstable but offline accurate ResDNN indicates an unstable mode with the growth rate of vast waves between 1/day and 10/day (Figure S1b). In our study, we find ablation of the top input levels cannot stabilize the SPCAM trained NN parameterizations, which is the same as Brenowitz et al. (2020).

On the contrary, the stable ResDNNs are free from the obvious breaking points. Therefore, we have to choose the background condition of a typical convective point in the tropics. The spectra analysis of a stable ResDNN shows a stable mode with the growth rate of nearly all wave numbers with phase speed above 5m/s below zero, indicating the stable ResDNNs attenuate/damp unreal initial perturbation energy (Figure S1a). Related analysis has been added in the revised manuscript (Section 3).

As for whether the NN parameterization performs better than CAM parameterization, we show from aspects. First, in the one-year offline validation (section 4), ResDNN is very closer to both the CRM predicted moistening rate and heating rate with a high coefficient of determination $R^2$ in most regions in the latitude-pressure cross-section plot (Figure 7), while the CAM parameterization hardly predicts any similarity. This is also true in the global distribution of precipitation $R^2$ (Figure 8). Second, for the prognostic 5-year simulation (from 1999 to 2003), NNCAM simulates reasonable climate mean states and especially get both better extreme precipitation and better MJO than CAM5, because the NN parameterization inherits the high variability from the resolved cloud and convection in the SP. However, we do observed NNCAM has larger humidity and temperature biases than CAM5 in mean sates. All these results have been added in Section 5 of the revised manuscript.

In conclusion, adequate training with enough accuracy is necessary for a stable NN-parameterization in realistic configured GCM. To get "enough" accuracy, the architecture of ResDNN rather than DNN is essential. A DNN in Rasp et al. (2018) can accurately emulate convection in aqua-planet SPCAM and brought stable muti-year prognostic simulations. Even an inaccurate DNN is stable in aqua-planet configuration but simulates biased climate. However, the real-geography data can significantly decrease the emulation skill of a DNN model (Mooers et al., 2021). The convection backgrounds are much more complex with meridional and zonal asymmetric and seasonal variated circulations, not to mention the orograph and various types of the underlying land surface. We believe that the implementation of a stable NN parameterization in a realistic configured climate model should be much more difficult than that in an aqua-planet model. Errors caused by inaccurate emulations will broadcast to the entire model through more complex waves and circulations. Therefore, it is understandable that plain DNNs and other inaccurate NNs would not work well in the realistic configured models. Speaking of Yuval and O'Gorman (2020) and Yuval et al. (2021), their machine learning parameterizations are trained from aquaplanet GCRM and are designed to separate vertical fluxes from convection and moistening and heating rate in microphysics and sedimentation, which inherits physical conservations and probably not that sensitive to offline accuracy. The NN-parameterization in this study is tendency-based, so it should not be directly compared with the above two works. After getting enough offline accuracy, we use the 2D gravity wave analysis framework to test stability of the NN parameterizations. The unstable ones will simulate a breaking point and with an unstable wave amplified mode, while the stable ones can attenuate the perturbation energy all the time and show a stable mode in the wave spectra. This framework explains why unstable NN parameterizations crash the model but is not an a priori way to design NN models. We still have to use the trial-and-error to filter out unstable ones and then select the best NN parameterization that can reduplicate the total energy time evolution of SPCAM with the least deviation. This selected NN-parameterization is undoubtedly closer to the SPCAM's CRM than the CAM parameterization in the offline test. For the prognostic simulations, the current NNCAM still suffers from some deviated climatology but can inherit the CRM's convective variability and do better in extreme precipitation and MJO than CAM5.

- *I think that more work on the text is needed to bring it to a level that is appropriate for the journal (see comments below). There are multiple repetitions, and sometimes sentences with little context*

**Reply:** Thanks for the reminder. We have tried our best to fix the problems in the text.

- *There are multiple times that the authors cite works that are not relevant to what they claim. For me especially alarming since I am not familiar with many of the references in the paper, and I found such mistakes on a small subset of references that I am familiar with. Therefore, I am worried that also other references are not relevant to what they claim.*

**Reply:** Thank you for the comment. We have checked and corrected all citations in the manuscript.

- *I think that some of the claims that are made in the manuscript are not supported by the results that are shown.*

**Reply:** We have dropped all the unsupported claims. We think this answered in the last part of comment (a) of *"and also it is not clear to me whether the NN parameterization performs better than traditional CAM parameterizations (in a few aspects it does, but in many other it does not)"* and the comments related to SPCAM, NNCAM and CAM5 comparisons below.

Comments:

*Abstract - I think that the authors overstate some of their outcomes (or at least do not show explicitly in the text these results; see more comments below). Some examples: I am not sure that in their parameterization outperforms CAM in terms of spatial distribution of precipitation and MJO propagation. Of the author want to state this, I think it is necessary to show the RMSE for the spatial distribution of precipitation and to show a quantitative comparison between the MJO spectrum/propagation in CAM, SPCAM and NNCAM.*

**Reply:** Thank you for the comment. We agreed that there is some over statements. Following your suggestion, we have rephrased them. NN-Parameterization can emulate the CRM in SPCAM much better than the CAM5 parameterizations in the offline tests. In prognostic tests, the NNCAM is still an experimental climate model. It can simulate the basic climatology with realistic configurations. However, the simulated climatology by NNCAM still cannot match the well-tuned CAM5, except better boreal summer precipitation and less biased winter distribution. Importantly, NNCAM can inherit the convection variability predicted by SPCAM, improving climate variabilities such as rainfall intensity distribution and MJO spectrum and propagation with higher pattern correlations (Figure 12, 13& 14).

*• Line 15 (and also through the abstract): the authors write that they learn from a cloud resolving model. This is confusing. I think that it is important to explain that they use SPCAM, and the task at hand is an emulation of a super-parameterization rather learning from a cloud-resolving output.*

**Reply:** Thank you for the comment. What we do mean here is to learn from the cloud resolving model in SPCAM. We agree with the comment and have changed that part to

*"to emulate a learn a parameterization scheme from different types of outputs from superparameterization (SP) with different types of outputs"* in our revised version.

• **Line 38**: *I think that one important reference regarding the biasses that are caused by different parameterization is a paper by Wilcox and Donner (2007). I might be mistaken, but this is the only case that I know where they show how two different parameterization schemes in the same model lead to a very different frequency precipitation distribution.*

**Reply:** Thank you for the comment. Wilcox and Donner (2007) talked about their improvements on a RAS to simulate better extreme rainfall events in GCMs. What we tried to show here is how convection parameterizations affect the simulations of ITCZ and MJO in GCMs. We have checked the citations and changed the first sentence into *"Many studies have attributed most of these biases to the imperfection of the parameterization schemes for atmospheric moist convection and cloud processes in current GCMs (Zhang and Song, 2010; Cao and Zhang, 2017; Song and Zhang, 2018; Zhang and Song, 2019)"*.

• **Line 44**: *Is there a citation the authors can provide for supporting the sentence "Their interaction with the atmospheric circulation affects the transport and distribution of energy and is the largest source of precipitation biases". If not please remove.*

**Reply:** Thanks. Actually, this is from Emanual et al. (1994) which discussed the interaction between convection and large-scale circulation. "*largest source of precipitation biases*" is not stated in that paper. So we have deleted that part of the sentence.

• **Line 56**: *"The CRMs have been applied to low-resolution GCMs to replace conventional cumulus convection and cloud microphysical parameterization schemes" - the sentence is unclear*

Reply: Thanks for pointing out this. We have changed it to *"In recent years, CRMs have been used as superparameterization (SP) in low-resolution GCMs to replace conventional cumulus convection and cloud parameterization schemes"*.

• **Line 57**: *Is there any other cases where CRMs were nested in GCM except super-parameteriztation (SP)? if not, please don't use "such as" as it implies there are other examples*

**Reply:** As far as we know, there is no other cases than SP. We thank the reviewer's comment and remove *"such as"* in the revised manuscript.

• **Line 61:** *"an order of magnitude larger compared with that for CAM." -this really depends on the CRM configuration, right? Even in the authors simulation it seems that SPCAM uses much more than factor 10 compared to CAM.*

**Reply:** When using 192 Intel CPU Cores of our commondity cluseter computer for calculations, CAM runs 70 times faster than SPCAM. However, the computation speed for different for subdomain resolution. We have added *"SPCAM requires far more computing resources than the same resolution CAM in 1 to 2 order of magnitude according to the resolution of the subdomain"* at lines 60-62 in the revised manuscript.

• *Line 66: Strange sentence, and I think it has some grammatical errors (e.g., which "the data-driven parameterization scheme", from the sentence it seems you refer to a specific scheme, and I do not think it is the case)*

**Reply:** Thank you for the comment. We have removed that sentence in the revised manuscript.

• *Line 68: "More recently," - more recently than what?*

***Reply:*** Thank you for this comment. We have changed *"More recently"* to *"In the last five years"* in line 65.

• *Line 72: I cannot see why the citations of Schneider et al. (2017) or of Duben and Bauer are related here. There is no convection scheme learned in these papers.*

**Reply:** Thank you for this comment. After checking the two papers, we acknowledged the mistake and have removed the two citation in the revised manuscript.

• *Line 74: "trained ones" - unclear*

**Reply:** Thank you for this comment. The description here is indeed unclear. We changed the *"machine learning algorithm"* to *"machine learning models"*, and also changed *"trained ones"* to *"trained machine learning models"*. Please see line 70 in revised manuscript.

• *Line 76: I do not see how Rasp (2020) citation is relevant here. This citation describes a method for online learning. Nothing related to an implementation of NN in GCM.*

**Reply:** Thanks for the comment. We have deleted such citation in the revised manuscript.

• *Line 85: "They found that minor changes, either to the training dataset or in the input/output vectors, can lead to model integration instabilities." - can you give a citation. I could not find such statement in the manuscript.*

**Reply:** Thanks for the comment. We cited the wrong paper. This statement is from Rasp (2020) in a section where it describes the work of Rasp et al. (2018). We have corrected this citation in line 81.

• *Line 89: There is a very relevant paper that the authors do not refer to. Brenowitz et al. (2020) have investigated both for SPCAM and for SAM why they can lead to numerical instabilities. Please add a discussion about it - and more importantly, I think that if the community could benefit from the this work it would be if the authors could identify why some of their networks are unstable (as was done by Brenowitz et al. (2020)).*

**Reply:** Brenowitz et al. (2020) is indeed important work. We thank the reviewer for the advice. We have added the discussion about numerical instabilities in new section 3.2 in the revised version with the 2D gravity analysis used that work and have confirmed the positive growing rate in unstable NN parameterizations.

• *Line 94-95: maybe instability and drift prevents the application some models, but other studies (that you cite) and also (Yuval et al., 2020) did not have any problem of unstable simulations. It is not fully determined why this is the case but the two possiblities that are raised in these papers are (a) because subgrid terms were calculated more accurately in these works compared to Brenowitz and Bretherton (2019), or (b) because these works succeeded to implement physical-constraints in the ML parameterizations).*

**Reply:** Thanks for the comment. The machine learning parameterizations in Yuval and O'Gorman (2020) and Yuval et al. (2021) are trained with inherent physical-constraints. Specifically, they are designed to emulate separated convection, microphysics, sedimentation, and vertical diffusion. The first two rely on eddy fluxes that are inherently conserved in mass and energy, and heating rate can be diagnosed from moistening with a factor Lv/Cp in the third and fourth. With this "smart" NN design, the NN prognostic simulations become stable. However, such design is possible in GCRM related studies but not in SPCAM. In our study, the NN parameterizations are tendency-based trained with realistic configured SPCAM simulation without any physical-constrain, where stability is indeed a problem to face.

• *Line 102: What is "auto-learning technique".*

**Reply:** Thanks for the question. *"auto-learning technique"* is Automated Machine Learning. This is a program-driven technology that automatically searches for hyperparameters in ML algorithms (He at al. 2021). For better understanding, we change *"auto-learning technique"* to *"automated machine learning technique"*

• *Line 104: Unclear what is "group of NNs" - do you mean different NNs for predicting different outputs? if yes, you should mention that this was already done by Yuval et al. (2020) (although in a different model).*

**Reply:** Yes, we now refer them as a set of neural network models. In this set of neural networks, each NN is in charge of different class of outputs, including heating moistening and radiation fluxes. The work of Yuval et al. (2020) have been added as reference related statements in the revised manuscript.

• *Line 105: The authors write: "We apply two innovative methods in neural network models: multi-target training to achieve balanced results across diverse neural network outputs and multilayer perceptron with residual blocks (ResMLP) to enhance nonlinear fitting ability." To me it is unclear if what the authors write here is really innovative: (a) multi-target training was already done by Yuval et al. (2020), although in a different model, so I could understand that it is a slightly different context. (b) If I understand correctly, Han et al. (2020) also used a DNN with shortcuts (called ResDNN in Han et al. (2020)) and showed it performs better than a standard DNN, which if I understand correctly is exactly what the authors call ResMLP.*

*As a side comment, I think that the term multi-target training is unclear and should be modified.*

Reply: We thank you for mentioning the two papers. We have acknowledged the limited similarity between our work and theirs. We have revised that part of the manuscript and have changed the term of *"muti-target training"* to *"train a set of neural networks"*. Since the *"ResDNN"* in Han et al. (2020) is the same as our *"ResMLP"*, we have changed the term *"ResMLP"* to *"ResDNN"* in the manuscript. Importantly, the *"ResDNN"* in Han et al. (2020) was only shown in their sensivity test to prove that the shortcut can be implemented in fully connected neural network to gain better accuracy. It was not evaluated in details, not to mention be formly validiated in prognostic simulations. It is our work to firstly use such network to achieve long-term stable simulations.

• *Line 111: I understand that it would be difficult to integrate complicated NNs into Fortran, but DNN (or DNN with shortcuts) should be a very simple procedure in Fortran.*

**Reply:** Thank you for the comment. As far as we know, python is the base in most famous NN frameworks and is widely-used in AI community. As a result, Python-based NN design has the best flexibility and productivity. Obviously, Fortran is not friendly to NN design but it is the de facto standard of scientific community, especialy for climate modeling . In our work, we want to utilize the two systems (Fortran-based climate modeling and Python-based NN design) as much as possible, and try to use the coupler technique to bridge them. We believe the idea is right and important and we will continuously improve our implementation.

• *Line 115: I think it is very problematic not to mention here that the simulations are stable without mentioning that there is a climate drift (although relatively very slow one).*

**Reply:** Thank you for the comment. We observed that the climatology simulated by NNCAM is biased but not constantly drifting away. The global distribution of the mean (Figure S4) precipitation is reasonable in global average with patterns close to those in the first 5 years (1999-01-01 to 2003-12-31) and pressure-latitude cross-section of temperature and humidity (Figure S3) for the last 5 years shows similar structure.

• *Line 145: Why is there a distinction between 2D CRM and CRM radiation? What does it mean CRM radiation?*

**Reply:** Thank you for the comment. The 2D CRM refers to the 2-dimensional cloud reserving model implemented in SPCAM to replace conventional subgrid parameterization, while CRM radiation refers to the revised RRTMG radiation module to let cloud-adiation interaction take place in the CRM domain. We have revised this part and added proper explanation in the revised version.

• *Line 148-151: this is a repetition of things that were already written in the intro.*

**Reply:** Thanks. We have deleted that part in the revised manuscript.

• *Figure 1: I might be missing something but for me these figures and the text that describe this figure is confusing. Aren't you just replacing SP with NN (like what was done in several other papers?). If yes I think that a very short and concise description will be much clearer.*

**Reply:** Thanks. Here we want to use NN parameterization to replace the CRM as well as the following cloud-radiation interaction in the CRM domain. Your suggestion is helpful. We have replaced the old Figure 1 in the revised manuscript with a short description in lines 197-203.

• *Line 152-154: I am not sure on what the statement is based on. I can agree that the prediction is more difficult, but why it has more numerical sensitivities (I am not even sure what it means)*

**Reply:** We appreciate this comment. We replace it with *"The convection background is much more complicated added with asymmetrical zonal circulation and various underlying surface."* for better clarification.

• *Line 161: the authors use the large-scale forcing as inputs. These were not used in previous papers trying to emulate SPCAM. It would be good to mention why they introduce these inputs, and if whether the inclusion of these inputs substantially improves the offline (or online) performance.*

**Reply:** Thanks for pointing out this. The large-scale forcings are firstly used in Gentine et al., (2018) and later replaced with meridional wind V in Rasp et al (2018). In SPCAM, the SP is continuously forced by large scale tendencies which includes all

large scale processes after the SP call at previous CAM timestep (Khairoutdinov et al., 2005). Most importantly, the large scale forcings in our realistic configured SPCAM contain PBL process and orographic gravity wave drag beside dynamics, which help our NN parameterizations identify the complicated surface conditions. We have added such description in lines 157-161.

• *Line 164: The authors predict also different outputs compared to previous studies that used ML to emulate a SP. e.g., they do not predict the vertical structure of radiative heating, but do predict other quantities. It would be helpful to understand what is the reason that they use different outputs (and inputs), and explain to the reader the motivation for these choices (and highlight differences from previous attempts to learn ML parameterization from SPCAM).*

**Reply:** Thank you for the suggestion. Firstly, the vertical profiles of shortwave and longwave radiation heating are included in the emulated total SP heating rate, which is the similar to those in Rasp et al. (2018). The net radiation fluxes (FSNS, FSNT, FLNS, and FSNT) are commonly used in previous studies but we find them just diagnostic terms which do not affect model states in prognostic runs. By reading Mooers et al. (2021) and the source codes of SPCAM, we later find out the solar radiation fluxes down to surface, including solar downward visible direct to surface (SOLS), solar downward near infrared direct to surface (SOLL), solar downward visible diffuse to surface (SOLSD), and solar downward near infrared diffuse to surface (SOLLD), are critical for the land surface model. If not included, the land surface is not heated up by the sun, therefore, seriously weaking the sea and land breeze. By adding those fluxes, we succeeded in getting reasonable land precipitation in the prognostic runs.We have added this description in lines 161-163 of the revised version.

• *Line 170: How does the authors deal with a negative precipitation (both offline and online)? Can you give information what is the percentage of (both online and offline samples) with negative precip?*

**Reply:** Thanks for the question. In fact, we set all negative precipitation to zero. It is indeed a rough way to derive precipitation from column integration of drying rate. Even for the SPCAM target data, derived negative precipitation accounts for 29% of all samples but only 14% of total precipitation intensity since most of the negative rainfalls stay above -1mm/day. For offline validation, there are 27% negative precipitation predictions in number and 17% in intensity with 93% staying above -1mm/day. For online validation, there are 22% negative precipitation in number and 17% in intensity with 93% above -1mm/day. Generally, the negative precipitation ratio in online tests is close to SPCAM target data and the offline test.

• *Line 180: Random split for the train,validation and test set is not ideal (and not the common practice) due to the time correlation between samples. In order get a reliable results for the test set what usually is done is to take the samples for each of the data*

*sets from different time intervals. Please do that and report offline performances when the test set is taken from a time interval that was used during training. Alternatively, make a justification for this choice.*

**Reply:** Thank you for the comment. The following is how data set is made. We use SPCAM running data from January 1, 1997 to December 31, 1998 to make the training sets; we use SPCAM running data from 2000 to make test sets for all offline validation. First, divide all the original data according to GCM time steps, that is, each GCM step contains 13,824 training samples composed of global grid points. Second, we randomly sample the time dimension to ensure that their time intervals are random and uniform. This ensures that the training and validation data are not continuous in time and come from different time intervals.

*• Line 185: Is there some citation that can backup the statement that MLP can generalize better than other types of networks? If not please remove statement.*

**Reply:** Thank you for the suggestion. Compared with some generative models, fully connected DNN does not have an advantage in generalization performance. But it is worthwhile and necessary to try DNN first in NNCAM. First of all, the input and output of NN-Parametrization are 1-dimensional vectors composed of multiple variables. It is different from the existing mainstream machine learning problems, such as image recognition and text-speech recognition, so it is impossible to directly apply most of the existing neural networks. Taking the convolutional neural network CNN as an example, the study of Albawi et al. (2017) shows that CNN has more advantages than DNN in the learning of large-scale images. The problem we face is that the input is a 122-dimensional vector stitched by multiple different physical quantities. The dimension of a single physical quantity is only 30 dimensions, which cannot meet the requirements of large-scale (generally at least 32×32 two-dimensional images). So there is no need to use CNN. Hornik et al. (1989) proved that a single-layer neural network can approximate any function. Although the physical process problem that NN-Parametrization needs to deal with is complex, from the point of view of machine learning, it is essentially a mapping problem from a 1D-vector with a length of 122 to a 1D-vector with a length of 65. According to the universal approximation theorem, DNN is feasible. Therefore, when constructing NN-Parametrization, we first tried to use DNN for fitting, and further introduced Residual blocks to extend DNN to ResDNN.

*• General comment: I might not understand what is the NN that you used. But if your multilayer perception is just a fully connected NN please say that, and preferably change the terminology to the common one (fully connected NN)*

**Reply:** Thank you for the comment. A multilayer perceptron (MLP) is a class of feedforward artificial neural network (ANN). An MLP consists of at least three layers of nodes: an input layer, a hidden layer and an output layer. Except for the input nodes, each node is a neuron that uses a nonlinear activation function. MLP utilizes a

supervised learning technique called backpropagation for training (Rosenblatt et al., 1961; Rumelhart et al., 1986). In this paper, MLP does refer to the Fully Connected Neural Network . Since the definition of MLP may be ambiguous, we will replace all *"MLP"* in the text with *"DNN"* and *"ResMLP"* with *"ResDNN"* in the revised manuscript.

• *Line 186-192: Please add graphs supporting the statements (these graphs can also go in the supplementary). Also the last sentence is unclear to me and I am not sure how it is related to the rest of the text.*

Reply: Thanks. In the new Figure 2, the $R^2$ of moist static energy changing rate is significantly lower than those of radiation fluxes under the same NN and training data, indicating different training difficulties for different variables. Also, we have removed the last sentence.

• *Line 193: The first sentence in the paragraph is unclear to me.*

**Reply:** Thanks. We have also removed that sentence and rewritten that paragraph (lines 171-184) in revised manuscript.

• *Lines 193-194: I disagree with this sentence about the independence - see Yuval et al. (2020) where they use the dependencies between the tendencies of moisture and thermodynamic variable to predict only one of them, and diagnose the other (because of a 1 to 1 mapping between the two for parameterizations like microphysics and sedimentation).*

**Reply:** Thank you for the comment. Both microphysics and sedimentation are only parts of the entire SP process (convection, microphysics, sedimentation, and diffusion). Therefore, our study's SP heating and moistening are related to each other but not the same relationship in Yuval et al (2020). We have revised this paragraph and no longer claim they are muti-target NNs. What we claim is a set of neural networks with each NN taking care of one type of output variables. So the networks can be best trained for their responding targets. The most important is to separate the NN used to predict heating and moistening with that used to predict radiation fluxes. The description has been added in lines 187-189 of the revised manuscript.

• *Line 201: I am not sure why this statement is necessary, and how it fits in the manuscript.*

**Reply:** Thanks for the question. In ther revised manuscript, this paragraph has been rewritten and this statement is changed to *"Since gradient descending is applied to optimize the network in training, mutual interference between different targets is expected to cause the cancel out of gradient directions used for descending (Crawshaw et al., 2020; Zhang & Yang, 2021) and ultimately affect the convergence of the network".*

• *Figure 3: To me it seems that the fully connected NN performs almost identically to the ResMLP (roughly a difference of 0.002 in R2). Did you test several fully connected NNs in an online setup and verified that using skip connections is really what makes the difference in terms of stable simulations? If not, please confirm that fully connected NNs do not lead to similar results (since if I understand correctly, the authors argue that this is one of the important aspects of their work). Furthermore, I think that the R2 results that presented are confusing and not the relevant ones for the reader. R2 should be calculated over different samples without any average before since such calculation gives the idea of the real performance of the network. So please change the way R2 is evaluated.*

**Reply:** We guess you may be referring to Figure 4. We replotted this figure as the new Figure2 with the validation dataset from the SPCAM simulation in 2000 as described in Section 2.2.1. In the revised figure, the $R^2$ difference is siginicant between ResDNN and DNN.

• *Line 213-214: On what are you basing your statement that if the NN is underfitting NNCAM crashes quickly? Did you test 100s of NNs? 1000s NNs? 1 NN? Please show a graph supporting this evidence (similar to the one that Ott did) if you want to keep this part of the text*

**Reply:** Thank you for your suggestion. We have plotted the Figure 4 showing offline validation accuracy versus online prognostic steps. We have implemented about 20 ResDNNs and several DNNs in NNCAM and found out most inadequately trained with low offline accuracy NNs cannot achieve long term stable online runs. We have addressed your concerns in Section 3 of the revised version.

• *Line 218-219: To me, this result (more accurate NNs that crushed earlier) indicates that the accuracy is not the important part of the stability. Previous work (e.g., Rasp et al. (2018) showed that also very shallow and less accurate NNs lead to stable NNCAM simulations in an aquaplanet setting). This raises a question why the resMLP is necessary and whether fully connected network could work as well.*

**Reply:** Thank you for the question. As our answer in the major comment (a) and the analysis in the section, we believe that high offline accuracy is necessary. The poorly trained and inaccurate NNs are unstable for long term simulations and so do the fully connected DNNs. Among the accurate ResDNNs, there is still no a prior way to determine their stability. However, with the gravity wave analysis introduced by Brenowitz et al. (2020), we can identify differences in the spectrum for stable and unstable NNs.

• *Table 2: Please give the full details of the training (e.g., which optimized). Furthermore, I do not think that the data has to be shown in a table.*

**Reply:** Thank you for the suggestion. We accordingly deleted the table 2 and added the the full details of the training stage in lines 205-216 in the revised manuscript.

*• Lines 223-229: The division between the stable and unstable group is interesting and I think here would be necessary to use the tools developed by Brenowitz et al. (2020) to check stability. If these tools cannot explain the instabilities in the simulations you are conducting, it is important for the community to know this.*

**Reply:** Thank you for your suggestion. The tools developed by Brenowitz et al. (2020) can help filter out the unstable NN parameterizations and explain why they blow up the model. The unstable ones will simulate a breaking point and have an unstable wave amplified mode, while the stable ones are able to attenuate the perturbation energy all the time and show a stable mode in the wave growth rate plot. This framework explains why unstable NN parameterizations crash the model but is not an a priori way to select stable NN parameterization. The analysis has been added in the new Section 3.2 of the revised manuscript.

*• Line 249: The authors claim several times during the manuscript that it is difficult to code in Fortran an NN. However, a fully connected layer is very easy to implement as it involves only matrix multiplications so it would be good to clarify that it might be difficult to use some fancy architecture, but the basic one that the authors use is very simple to code in Fortran.*

**Reply:** Thank you for the comment. Our new coupler excels the hard coding Fortran NNs in the following aspects: 1) It directly use the original python codes, which is more convenient to switch between different NN models since we have to conduct numerous online tests. 2) It supports all advanced NN architectures. 3) It can compute much faster by using GPUs without hindering the computational efficiency. For your concerns, we implemented a ResDNN in Fortran and tested the performance, please see the lines 295-298 in the revised manuscript.

*• Line 249: "At runtime..." This is a confusing sentence and it is unclear to me why you cite the papers (as they use a different code infrastructure so I do not understand why these citations are included here)*

**Reply:** Thank you for the comment. We accordingly deleted *"At runtime"* in the revised manuscript.

*• Line 251: "outside", you need to explain outside of what.*

**Reply:** Thank you for the comment. It means "outside of fortran program". Deploying NN in a Fortran program requires importing bias and weights of NN from outside of Fortran program. We have clarifed it in the line 236 in the revised manuscript.

*• Line 255: "In this..." this is repeating similar statements so I do not think it is necessary.*

**Reply:** Thank you for the suggestion.We have deleted *"In this research, we regard NN-Parameterization as a component model coupled to NNCAM."* in the revised manuscript.

*• Line 265-269: Sounds like a great achievement! (I haven't tried it myself though)*

**Reply:** Thank you for your encouragement.

*• Line 278: Sounds great!*

**Reply:** Thank you for your encouragement.

*• Line 296 and Figure 7: Please show R2 before zonal averaging! It does not make sense to me first zonal average (and also this was not done in previous work)*

**Reply:** Thank you for the comment. Mooers et al. (2021) made clear that their $R^2$ is for the zonal averaged heating and moistening. Han et al. (2020) also calculated temporal standard deviation for zonal averaged fields. So we just follow their examples here.

*• Figure 8: SP is especially important in the tropics, and it seems That the skill in the tropics is very low (many regions have R2 close to 0). Can you give some insight?*

**Reply:** Thank you for the comment. Our NN parameterization is trained with the loss function of mean squared error, which is not sensitive to incorrect predictions of small values. In Figure R1b, the local variance/std is close to zero for those low skill regions. The MSE in those regions is also low but is still high compared with its variance. Therefore, when calculating $R^2$ as 1-mse/var, many of those low std regions will have $R^2$ close to zero.

*• Line 301: I think that it is difficult to determine whether this is a good job or not because there is no baseline to compare to. If you could provide a baseline from CAM (for offline prediction), then it would make sense to give this statement.*

**Reply:** We appreciate your suggestion here. We have added the $R^2$ of offline predictions form CAM parameterization in SPCAM. Our NN-Parameterization shows great emulation skill by bring far better offline accuracy than the CAM parameterizations in heating, moistening and precipitation in Figure 7 and Figure 8 of the revised version.

*• Line 306: The word "fitting" should not be there as far as I understand*

**Reply:** Thanks. We have deleted this word in revised manuscript.

• *Line 306-308: "As suggested..." This sentence is unclear to me. How do you manually tune an NN?*

**Reply:** Thanks. *"Manually tune an NN"* means that in the process of training the NN, manually adjust the hyperparameters in order to improve the performance of the NN. The study of Mooers et al. (2021) pointed out that manual tuning is difficult to obtain an NN with excellent performance. In practice, we did not use *"manual tuning"* training, but used some automated tools, which use the automated machine learning technology.

• *Line 309-314: I think that these sentences are not well related to each other.*

**Reply:** Thanks for the comment. We have revised these sentences in Section 4 of the revised manuscript as *"The real-geography data can significantly decrease the emulation skill of a deep learning model (Mooers et al., 2021), where the convection backgrounds are much more complex with meridional and zonal asymmetric and seasonal variated circulations, not to mention the orograph and various types of underlying land surface. This result further strengthens the necessity of the ResDNN architecture."*

• *Line 317-318: "At the same time...." If I remember correctly this is the default choice*

*of SPCAM so it is very confusing you are writing this as if this is something special*

**Reply:** Thanks for the comment. We have deleted that statement.

• *Line 321: The fact simulation have a climate drift should be mentioned already in the abstract.*

**Reply:** Thank you for the comment. We have added *"some biases in climate simulation"* in the abstract.

• *Line 321-325: Please compare NNCAM and CAM to the relevant simulated period of SPCAM. I do not understand why you compare different periods (and different length of time intervals).*

**Reply:** Thanks for the comment. We have changed the simulation period from 1999-01-01 to 2003-12-31 for NNCAM, CAM5 and SPCAM.

• *Line 324: Why here (CAM) 1 year spinup and for NNCAM - half a year? I guess that this is how it is initialized?*

**Reply:** Thanks for the question. We were afraid of the incompetence of our NN-Parameterization in model initialization. Therefore, we ran SPCAM with the original SP for half a year to spin up the model. We later find the half a year SP run is not needed. NNCAM can be run as start up. SPCAM, CAM5 and NNCAM all start up at 1998-01-01. We have emphasized this in the revised manuscript.

*• Line 324: Is it critical to use SPCAM for initialization for NNCAM? if yes please explain why? Previous studies used a coarse run (in this case CAM ) to initialize the model which makes more sense to me.*

**Reply:** The half-a-year spin of SPCAM is not needed but calling the SP in SPCAM at the first step is required to generate the correct large scale forcings for NN Parameterization. We have emphasized this in lines 364-371 in the revised manuscript.

*• Figure 9: Please show RMSE in the figure for each of the sublots - so we could compare NN to CAM. Also for other figures and claims made by the the authors - if the authors want to state that NNCAM performs better than CAM, please provide a quantitative metric for the comparison.*

**Reply:** We thanks for the suggestion and have added RMSE for the NNCAM and CAM predictions. Please take a look at the new Figure 9 in the revised manuscript. The mean climate states simulated by NNCAM is more biased than CAM5. We have added this statement the abstract.

*• Figure 10: Need to quantify the accuracy - please show RMSE*

**Reply:** Thank you for the suggestion. We have added RMSE in the new Figure 10 in the revised manuscript.

*• Figure 11: NNCAM has less skill than CAM so it would be fair to clearly mention it. Furthermore, please use some metric for the comparison if such a comparison is made. The fact that CAM is closer to SPCAM than NNCAM in many of the fields should be also mentioned in the abstract (to me it is a bit misleading to only mention in the abstract that NNCAM performs better than CAM without stating that in many aspects CAM is better than NNCAM)*

**Reply:** Thanks for the comment. We have corrected all the statements about the performance of climate mean states. We have made clear that NNCAM is still an experimental model and simulates reasonable but has some biases in climate mean states. We improved Figure 11 of zonal mean precipitation with spatial distribution differences. We believe that the improved Figure 11 will add more information than the previous one.

*• Figure 12: Would be good to avoid the noise in this plot by using bins that have similar distances in a log scale.*

**Reply:** Thank you for the suggestion. We have revised this figure by changing the log scale x axis to a linearized one and used a constant bin as 2mm/day.

*• Figure 13: Can the authors quantify the distance between distributions? It is difficult for me to determine if NNCAM or CAM is closer to SPCAM, since NNCAM has a very different MJO structure compared to SPCAM. Furthermore, please add*

*more contours and make sure that the colorbar isn't saturated (at the moment it seems saturated).*

**Reply:** Yes, we have reploted the Figure 13 and added coefficient of determination $R^2$ for the distrance between distributions. In the revised Figure 13, we don't think saturation is a problem anymore.

*• Line 383/Figure 14: The comparison should be between SPCAM propagation and NNCAM propagation because SPCAM is the baseline. I do not see why using the value of 5 m/s is relevant. To my eyes it seems that SPCAM propagates faster than 5m/s and has similar propagation as CAM. Namely, I disagree that NNCAM propagation is closer to SPCAM propagation compared to CAM.*

**Reply:** Thank you for the comment. We no longer use 5 m/s as a reference MJO propagation speed but directly compare the 3 time-lag plots of SPCAM, NNCAM and CAM5. When plotting precipitation and U200 at the same time, SPCAM and NNCAM show eastern propagation and but CAM5 contains a western propagation speed over Indian Ocean and the maritime continent.What's more, NNCAM has a higher $R^2$ of U200 than that of CAM5, showing more resemblance.

*• Figure 14: Can you describe on which data was that calculated for.*

**Reply:** We use the total precipitation and zonal wind U at 200mb predicted by SPCAM, NNCAM, and CAM5 respectfully.

*• Line 395: instead of "GCM" use or "a GCM" or "GCMs"*

**Reply:** Thanks. We applied this change in the revised manuscript.

*• Line 399-404: You cannot not mention the climate drift.*

**Reply:** Thank you for the comment. We think *"some biases in climate mean states"* is a more proper term than *"climate drift"*. We have discussed the stabilities and climate mean states of NNCAM in Section 3 of the revised manuscript.

*• Line 425-427: Also I think that the relevant comparison would be how fast it would run without the coupler. As far as I understand it should work pretty fast also without the coupler (which uses also additional resources) - as the matrix multiplication operation in Fortran is optimized pretty well.*

**Reply:** When using 192 CPU Cores of our commondity cluseter computer, the SYPD of Coupler-based NNCAM (with the support of 1 GPU) is 10.0, and the SYPD of Fortran-based NNCAM using Intel Math Kernel Library but witout GPU is only 1.5.

*• General comment: The authors mention that they have a group of NNs that lead to stable simulations. Are all these NNs lead to similar online results? If yes, please*

*mention this. I suggest showing the STD for precipitation and a couple of other fields among the different stable online simulations you achieved since it will show the reader that there is no "cherry picking" with the choice of the simulation you end up showing.*

**Reply:** Thank you for the comment. We so far have 4 stable NN parameterizations during our experiments. They all have reasonable global distribution of precipitation but are different from each other in the heavy precipitation regions. We plot the STD for precipitation across these models as in Figure R2.

*• references not ordered alphabetically in some cases which makes it more difficult to find them.*

**Reply:** Thank you for the suggestion. We have reorganized all the references.

**Reference:**

Albawi, S., Mohammed, T. A., and Al-Zawi, S.: Understanding of a convolutional neural network, 2017 International Conference on Engineering and Technology (ICET), 21-23 Aug. 1-6, 10.1109/ICEngTechnol.2017.8308186, 2017.

Chantry, M., Hatfield, S., Dueben, P., Polichtchouk, I., and Palmer, T.: Machine Learning Emulation of Gravity Wave Drag in Numerical Weather Forecasting, Journal of Advances in Modeling Earth Systems, 13, e2021MS002477, https://doi.org/10.1029/2021MS002477, 2021.

Emanuel, K. A., David Neelin, J., and Bretherton, C. S.: On large-scale circulations in convecting atmospheres, Quarterly Journal of the Royal Meteorological Society, 120, 1111-1143, 10.1002/qj.49712051902, 1994.

Gentine, P., Pritchard, M., Rasp, S., Reinaudi, G., and Yacalis, G.: Could machine learning break the convection parameterization deadlock?, Geophysical Research Letters, 45, 5742-5751, 2018.

Han, Y., Zhang, G. J., Huang, X., and Wang, Y.: A Moist Physics Parameterization Based on Deep Learning, Journal of Advances in Modeling Earth Systems, 12, e2020MS002076, 10.1029/2020ms002076, 2020.

He, K., Zhang, X., Ren, S., and Sun, J.: Deep residual learning for image recognition, The IEEE Conference on Computer Vision and Pattern Recognition (CVPR), Las Vegas, Nevada, June 26 - July 1, 770-778, 2016.

He, Xin, Kaiyong Zhao, and Xiaowen Chu. "AutoML: A Survey of the State-of-the-Art." Knowledge-Based Systems 212 (2021): 106622.

Hornik, K., Stinchcombe, M., and White, H.: Multilayer feedforward networks are universal approximators, Neural Networks, 2, 359-366, https://doi.org/10.1016/0893-6080(89)90020-8, 1989.

Khairoutdinov, M., Randall, D., and DeMott, C.: Simulations of the Atmospheric General Circulation Using a Cloud-Resolving Model as a Superparameterization of Physical Processes, Journal of the Atmospheric Sciences, 62, 2136-2154, 10.1175/jas3453.1, 2005.

Mooers, G., Pritchard, M., Beucler, T., Ott, J., Yacalis, G., Baldi, P., and Gentine, P.: Assessing the Potential of Deep Learning for Emulating Cloud Superparameterization in Climate Models With Real-Geography Boundary Conditions, Journal of Advances in Modeling Earth Systems, 13, e2020MS002385, https://doi.org/10.1029/2020MS002385, 2021.

Rasp, S.: Coupled online learning as a way to tackle instabilities and biases in neural network parameterizations: general algorithms and Lorenz 96 case study (v1.0), Geosci. Model Dev., 13, 2185-2196, 10.5194/gmd-13-2185-2020, 2020.

Rasp, S., Pritchard, M. S., and Gentine, P.: Deep learning to represent subgrid processes in climate models, Proceedings of the National Academy of Sciences, 115, 9684-9689, 10.1073/pnas.1810286115, 2018.

Rosenblatt, F.: Perceptions and the theory of brain mechanisms, Spartan books, 1962.

Rumelhart, D. E., Hinton, G. E., and Williams, R. J.: Learning internal representations by error propagation, California Univ San Diego La Jolla Inst for Cognitive Science, 1985.

Yuval, J. and O'Gorman, P. A.: Stable machine-learning parameterization of subgrid processes for climate modeling at a range of resolutions, Nature Communications, 11, 3295, 10.1038/s41467-020-17142-3, 2020.

Yuval, J., O'Gorman, P. A., and Hill, C. N.: Use of Neural Networks for Stable, Accurate and Physically Consistent Parameterization of Subgrid Atmospheric Processes With Good Performance at Reduced Precision, Geophysical Research Letters, 48, e2020GL091363, https://doi.org/10.1029/2020GL091363, 2021.

**Figure R1 - R5**

[Figure]

**Figure R1.** Spatial distribution of a) root mean square error (RMSE) and b) standard deviation (STD) of precipitation prediction.

[Figure]

**Figure R2.** Spatial distribution of precipitation STD accros all 4 stable NN parameterizations for the prognotistic simulation from 1999 to 2003

**Figure 1 – 14**

[Figure]

**Figure 1.** Schematic showing the structure of ResDNN. It consists of 7 residual blocks, each of which (shown in dashed box) contains two 512 node-wide dense (fully-connected) layers with a ReLU as activation, and a layer jump. The input and output are discussed in section 2.2.2.

[Figure]

**Figure 2.** Fitting accuracies ($R^2$) of both the proposed ResDNN (orange solid lines) and DNN (blue dashed lines) for different targets. (a) shows the R2 of moist static energy changing rate (dh) versus training epochs and (b) shows the fitting accuracy of the average $R^2$ over the 8 radiation fluxes. Note: Spatial averaging of MSE is performed before calculating $R^2$.

[Figure]

**Figure 3**. A flow chart of NNCAM including NN-GCM Coupler. NNCAM runs in the direction of the arrow, and each box represents a module. Among them, NN-GCM Coupler is indicated by light red. NN-Parameterization is shown in the sub-figure on the right. Note: ① represents the dynamic core transmits data to NN-GCM Coupler; ② and ③ represent the data communication between DNN-GCM Coupler and NN-Parameterization; ④ represents the host GCM accepts the result from NN-Parameterization.

[Figure]

**Figure 4.** The offline moist static energy mean square error vs. prognostic steps. The black reversed triangles are stable NN coupled prognostic simulations lasting more than 10 years, blue ones are unstable simulations, and the blue triangles are for DNNs. The marked dots with colored outline are later exhibited in Figure 5 for time evolution of global averaged energy.

[Figure]

**Figure 5.** Time evolution of global averaged column integral total energy of NNCAM with different ResDNN parameterizations (marked with the same colors in Figure 4) and SPCAM target (the black line): Blue for stable and accurate ResDNN, green for a stable but deviated ResDNN, orange and red lines for unstable ResDNN.

[Figure]

**Figure 6.** Latitude-pressure cross sections of annual and zonal mean heating (top) and moistening (bottom) from moist physics during the year 2000 for (a, c) SPCAM simulations, and (b, d) offline test by the NN-Parameterizations.

[Figure]

**Figure 7.** Latitude-pressure cross sections of coefficient of determination ($R^2$) for zonal averaged heating (left panels) and moistening (right panels). They are predicted by (a & b) NN-Parameterization in the offline one-year SPCAM run, and (c & d) by offline CAM5 parameterizations. Both are evaluated at 30-min timestep interval. Note: areas where $R^2$ is greater than 0.7 are contoured in pink and those greater than 0.9 are contoured in orange.

[Figure]

**Figure 8.** Latitude-pressure cross sections of coefficient of determination ($R^2$) for the derived precipitation predicted by NN-parameterization (a) and total precipitation from CAM5 parameterization (b) in the offline one-year SPCAM run. The predictions and SPCAM targets are in 30min timestep interval. Note: areas where $R^2$ is greater than 0.7 are contoured in pink and those greater than 0.9 are contoured in orange.

[Figure]

**Figure 9**. Latitude-pressure cross sections of annual and zonal mean temperature (left panels) and specific humidity (right panels) from (a, b) SPCAM (1999−2003), (c, d) NNCAM (1999−2003), and (e, f) CAM5 (1999−2003).

[Figure]

**Figure 10**. The mean precipitation rate (mm day$^{-1}$) of June-July-August (left panels) and December-January-February (right panels) for (a, b) SPCAM (1999−2003), (c, d) NNCAM (1999−2003), and (e, f) CAM5 (1999−2003).

[Figure]

**Figure 11.** Global distribution of precipitation difference averged over boreal summer (left panels) and winter (right panels) between NNCAM and SPCAM (a & b) and between CAM5 and SPCAM (c & d).

[Figure]

**Figure 12**. Probability densities of daily mean precipitation in the tropics (30°S−30°N) from the three model simulations. Black, blue and red solid lines denote SPCAM, NNCAM and CAM5, respectively.

[Figure]

**Figure 13**. ... ...quency spectrum ... ( S– ) daily pre... ...a...a...omal... ...r (a, b) SPCAM, (... ...N ...A...d (... ...CAM... simul... ...rs f...b...real winter...

**Figure 14.** ...tude-time ...on of ...correlation ...icient ...th... 20-100...ay b...d-pass-filtered pre...tion ano...l ...eraged...r 10°S−10... ...st reg...ly averag... p...itation (shaded) an... ...vin...in... (conto...ed... ...qu...al easte... ...e...100°E, 10°S−10°N... line...panel d... s...rd pro...sp...

**Figure S1 – S4**

[Figure]

**Figure S1.** Wave spectra of a) a stable NN parameterization and b) an unstable parameterization. The light blue background indicates where the phase speed is above 5m/s and the growth rate is positive. The stability diagrams are obtained by coupling linear responses of NN parametrizations to simplified 2D dynamics with a chosen base state, which is normal convection background in the long-term prognostic for a) and an initial state for unreal gravity wave in Move S1 for b).

[Figure]

**Figure S3.** Latitude-pressure cross-section zonal and annual mean for a) temperature and b) specific humidity in NNCAM simulated from 2004 to 2008 with their differences with the SPCAM simulation from 1999 to 2003.

[Figure]

**Figure S4.** Global distribution of temporal mean precipitation predicted by NNCAM from January 1st 2004 to December 31st 2008 for a) annual, b) boreal summer (JJA), and c) boreal winter (DJF).

---

## Author Comment (AC5)

**Response to reviewer 2**

In this manuscript the authors use neural networks to emulate the grid-box mean output of a superparametrization scheme, which predicts the sub-grid tendencies for moist physics and radiative heating. After a period of offline training the authors develop a new coupling approach for online testing. In online testing they find some evidence of improvements over the existing CAM5, i.e. a closer fit to the SPCAM approach.

There are several interesting ideas in this manuscript and some impressive technical developments in the coupling framework. However, I do not currently feel the manuscript is near acceptance for publication. My main issue is that the online testing analysis is not consistent, and does not persuade the reader that NNCAM is an improvement over the existing CAM5 model. Given that NNCAM is slower than the normal CAM5 parametrizations I think it is important to show that NNCAM provides an improvement. If this is not possible, then instead the authors could focus more effort on establishing whether any offline metrics provide a better indicator of online stability. Below I will detail my comments further. I hope the authors will take these on board, as I think this manuscript could make for an interesting and useful paper.

**Reply:** Thanks for the careful review. We have tried our best to answer all comments below and made proper revisions to the manuscript. The reviewer's comments are in italic and our response is in normal and blue font. Please note that the Figures 1-14 are in the revised manuscript, the Figures S1-S4 are in the supplementary, the Figures R1-R5 in the last part of this response, and the Movies S1 and S2 are in the attached files. All figures mentioned in this response are listed in the last part.

• The section on online performance analysis is a weak point. I think it is important to standardise the measurement periods used by the CAM5, SPCAM and NNCAM. Showing CAM5 and NNCAM as deviations away from the "truth" of SPCAM would ease the process of comparison. In many of the figures it is unclear that NNCAM is an improvement on CAM5, which begs the question of the purpose of the networks. I also think that showing plots of global metrics against time would help identify drift in the models. Given that this paper has a climate motivation, examining this behaviour seems crucial. I also think there is insufficient analysis of the effects of emulating radiative heating. It would be interesting to see some global maps of average 2m temperature to see the effects of the surface fluxes and near-surface heating rates.

**Reply:** Thanks for the comment. First, we have standardized the measurement period used by SPCAM, NNCAM and CAM5, which is from 1999-01-01 to 2003-12-31 after a year of spin up. We admit that our NNCAM still contains biases in the simulated climate mean states in climate analysis but better in climate variabilities such as extreme rainfall events and MJO. The time evolution of total energy (Figure 5) confirms no significant climate drift in NNCAM (the blue line).

Specifically for the mean climate states, NNCAM has a warmer tropopause, a colder air temperature at the polar upper levels and a warmer mid and low troposphere above polar regions than SPCAM. It is also wetter for the entire troposphere (Figure S2). On the contrary, CAM5's climatology is less deviated. NNCAM has a much warmer 2m temperature in high latitudes during boreal winter, while CAM5 only shows a slightly colder temperature in those regions (Figure R3). For global precipitation distribution, NNCAM deviates less than CAM5 in global averages in boreal summer (Figure 11a) with similar patterns and smaller RMSE (Figure 10c and 10a) but performs worse in winter accompanied by a significant underestimation along the equator (Figure 11b). Therefore, although NNCAM is still considered significant progress from the other unstable or biased NN-parameterization coupled simulations, it is admitted an experimental model. We correct the statement and don't claim it is comprehensively better than CAM5 in the revised manuscript.

Even with inevitable biases in the simulated climatology, the NN-Parameterization can still inherit the variability predicted by the resolved convection process in SP. NNCAM is significantly closer to SPCAM in simulating extreme precipitation and MJO than CAM5, with much higher heavy precipitation probability (Figure 12) and closer MJO spectrum (Figure 13) and propagating speed patterns (Figure 14). Specifically, we use the coefficient of determination ( $R^2$ ) to measure the distance of spectrum and lagged coefficient between SPCAM and NNCAM or CAM5. Higher  $R^2$  means more similarity. With an  $R^2$  of 0.511 rather than that of 0.397 for CAM5, NNCAM indeed performs better in boreal winter MJO precipitation spectra. For the lagged coefficient with 5-year data, NNCAM performs even better in longitude evolution especially for the 200 hPa zonal wind U.

• The authors highlight that online stability is not a given for coupling of parametrization schemes. This is a really interesting and important point for this field of study. However the authors' proposed solution is trial and error, suggesting that short term stability is a good predictor of long term stability. I would like to see more detailed analysis of whether there are good offline measures that can guide online stability. The authors suggest that improving R2 scores are not fully correlated with stability. Can one find a different metric that is better correlated with stability, analysing the results the authors have already conducted? I would be interested to see if mean-squared-error, mean bias or some measure of worst error were better predictors. If I understand the work correctly, you train four networks in your SPCAM. When you test stability are you swapping these four networks individually, or swapping all four together? This might shed light on which components were more important for stability. I think studying these points could provide great insight into the problem.

**Reply:** Thanks for the comment. We have added a series of sensitivity test in Section 3 to determine stable NN parameterizations. Although short term stability still matters the long term stability, we no longer use them as the only metric to select NNs. In the sensitivity test, we have adjusted the machine learning metrics for evaluating NNs, to

the mean squared error of moist static energy changing rate (dh) to evaluate heating and moistening rate together. This new metrics shows insightfull relationships with the max prognostic steps in Figure 4: when the offline MSE of ResDNN decreases to a certain level (e.g.,  $290W^2/m^4$ ), the ResDNN parameterization may run stably for long periods. All less accurate NN parameterizations are unstable. After getting enough offline accuracy, we use the 2D gravity wave analysis framework (Brenowitz et al., 2020) to test stability of the NN parameterizations. The unstable ones will simulate a breaking point and with an unstable wave amplified mode (Figure S1b), while the stable ones can attenuate the perturbation energy all the time and show a stable mode (Figure S1a) in the wave spectra. This framework explains why unstable NN parameterizations crash the model but is not an a priori way to design NN models. We still have to use the trialand-error to filter out unstable ones and then select the best NN parameterization that can reduplicate the total energy time evolution of SPCAM with the least deviation.

As for swapping neural networks, we do not change the neural network for the 8 radiation fluxes because they are highly accurate and well trained with a collaborate  $R^2$  above 0.98. Different from the easily and accurately trained radiation fluxes, the tendencies of temperature and moisture are rather difficult to train and, if not trained well or with the right NN architecture, can seriously affect the prognostic performance and stability. So, we swap the neural networks for dqv and dT together but not individually. However, we find the NN for moistening rate dqv is most difficult to train and possibly more important for stability.

• If I understand the training/validation/testing split correctly, these are random subsets in space and time from the 1998/9 dataset. If so, I do not think this is a safe method for ensuring no overfitting, as this does not take into account spatial/temporal correlations. I think the total dataset should be split by time only, with temporal gaps between training, validation and testing to ensure independence. This might explain why NN with better R2 values provide less stable answers, if there is overfitting on the dataset

**Reply:** Thanks for the comment. We have changed the way to divide the training and offline validation dataset to ensure independence. The training dataset used by all considered NNs is 40% temporally random sampled from the 2-year SPCAM simulation from 1997-01-01 to 1998-12-31. Notably, the random sampling is only done in the time dimension but not in latitude and longitude, including all samples globally of the selecting timestep. To avoid any mix or temporal connection between the training set and offline validation set, we random sample 40% timesteps from the SPCAM simulation in the year 2000 as the offline validation set.

There is very little discussion about the benefits and downsides to superparametrization. It is my understanding that there is very limited (if any) evidence that superparametrization actually improves model climate versus typical parametrization schemes. I think it is worth stating this, or if the authors disagree, provide citations. **Reply:** Great question. We also think researchers should be frank with the pros and cons of SPCAM before using it as a target model. Khairoutdinov et al. (2005) shows that SPCAM produced quite "reasonable" geographical distribution of precipitation, precipitable water and cloud fraction, but has a notable precipitation bias in the Western Pacific. On the other hand, the SP substantially improves convection variability in multiple ways, including diurnal variation, probability distribution of precipitation intensity, and intraseasonal variability such as MJO.

Are the only parametrizations within the CAM model those in the superparametrization? e.g. there is no parametrization for sub-grid orographic gravity waves.

**Reply:** Thank you for the question. The orographic gravity waves drag and vertical diffusion are computed after calling the land surface model and before the next round calculation of the dynamic core.

I suggest re-ordering manuscript to explain coupling before explaining results. The results section makes reference to coupled testing without explaining how this is achieved.

**Reply:** Thank you for the comment. We actually did explain coupling before explaining results in the original manuscript. Anyway, in the revised manscript, we have added the explanation of the coupling strategy in Section 2.2.3 and 2.3, and the description of the results is in Section 4 and 5.

L135: Where does the variability originate in the CRMs? Are they initialised with different perturbations of the larger-scale conditions? If there is stochasticity in the system? It would be good to state this if true.

**Reply:** Thank you for the question. Yes, they are kind of initialization with different perturbations of the large-scale conditions. Specifically, at the beginning of each simulation, the SP/CRM fields in each CAM grid column are initialized by the soundings with small amplitude noise added to SP temperature fields near the surface. No noise is added at later times (Khairoutdinov et al., 2005).

*L166: "as output the NN-Parameterization". I think this should be "as outputs from the NN-Parameterization".*

**Reply:** Thanks. We applied this change in the revised manuscript as "Also, it is important to include direct and diffuse downwelling solar radiation fluxes as output variables to force the coupled land surface model.".

L167: "is critical to improve the performance of the NNCAM". I could find no further discussion of this. It sounds like a very interesting point. Please expand.

**Reply:** Thanks for the comment. The 4 solar radiation fluxes down to surface represents the received solar energy by the land surface model. If they are not included, the land surface will not be heated up by the sun, weaking the land-sea breeze and monsoon circulations. We have expanded in the second paragraph of Section 2.2.1 in the revised manuscript.

L190: Are you training to maximise R2? If not, what is your function to minimise /maximise?

**Reply:** Yes, we want to minimize the mean squared error for each variable not the  $R^2$ . However,  $R^2$  is later used to measure the degree of fit between the NN predictions and the reference values generated by SPCAM.

L195: Have you tested this theory of mutual interference? I would have thought that training two different models to predict the TOA and surface fluxes would introduce physical inconsistencies. These are not separate pieces of physics.

**Reply:** The work of Crawshaw et al. (2020) and Zhang & Yang. (2021) proved the necessity of separating forecast targets. At the beginning of the experiment, we did test using a DNN to try to predict all variables, and the DNN could hardly converge. After we separated the predicted targets dqv, dT and radiation variables according to Crawshaw et al. (2020) and Zhang & Yang . (2021), the network began to converge and obtained satisfactory results. We use one NN to train all radiation fluxes in the revised manuscript.

L214: "a well-fit is necessary". This was unclear and could be better written.

**Reply:** Thanks for the comment. We have added relative analysis in the new section 3.1. "*a well-fit is necessary*" is replaced with a throught analysis in sensitivity tests. First, we train NNs over a threshold of accuracy, which makes stable for long-term prognostic simulations possible. Then, we have to use trial-and-error to filter out unstable NNs and select the best for the most accuate long-term simulations.

*L229: Is the "best performance" network based upon the best performance in offline or online testing?*

**Reply:** Thanks for the question. *"best performance"* means best ResDNN set in online testing.

L242: The online coupler sounds like an interesting solution of value to the wider scientific community. Are the authors planning to share this as a stand-alone piece of software?

**Reply:** Thanks for the comment. You can access it through our open resource lib (https://doi.org/10.5281/zenodo.5596273). We plan to continuously improve and optimize the online coupler in the future.

L278: I do not understand "reaches half the speed of CAM5". Are the authors comparing to the speed of CAM5 with the normal parametrization schemes? By half the speed to they mean it will take twice the time to simulate the same period?

**Reply:** Thanks for the question. *"reaches half the speed of CAM5"* means that the total simulation time of NNCAM is double of that of CAM5. When using 192 CPU Cores of our commondity cluseter computer, the SYPD of Coupler-based NNCAM (with the support of 1 GPU) is 10.0, and the SYPD of Fortran-based NNCAM using Intel Math Kernel Library but witout GPU is only 1.5.

L279: Have the authors profiled how much time is spent communicating data versus doing ML inference? This would be very interesting to see.

**Reply:** Thanks for the question. To answer your question, we conducted the run of NNCAM for time breakdown. Indeed, the communication through coupler and computation of neurl networks takes almost equal time, and there is still a lot of room for performance optimization. For your concerns, we implemented a ResDNN in Fortran and tested the performance, please see the lines 295-298 in the revised manuscript.

L280: If I have understood correctly the authors carry out the online testing on the same time period that the NN was trained on. Has any effort been made to ensure independence between the training and testing data?

**Reply:** Thanks for the comment. We have changed the way to subsets the training and offline validation dataset to ensure independence. The training dataset used by all considered NNs is 80% temporally random sampled from the 2-year SPCAM simulation from 1997-01-01 to 1998-12-31. Notably, the random sampling is only done in the time dimension but not in latitude and longitude, which means once a timestep is selected, all global samples belonging to that step go "on board". To avoid any mix or temporal connection between the training set and offline validation set, we random sample 40% timesteps from the SPCAM simulation in the year 2000 as the offline validation set.

L305: "tunned" -> "tuned"

**Reply:** Thanks. We have applied this change.

L320: The authors run for 10 years but only analyse 4 years of data. So their only expectation of the final 5 years is for the model to not crash. I do not think this is an appropriately strict assessment for their NN models. I think examining model drift is

exactly the important test of a NN. If not, what is the purpose of the model that the authors are building?

**Reply:** Thanks for the question. After running NNCAM as start up, we reorganized all the results. NNCAM does simulate more biased climate states than CAM5 but has no obvious climate drift. We have shown the global distribution of temporal averaged precipitation for the last years in Figure S4. The averages are not drifted and the patterns are similar to those for the first 5 years.

L325: It seems a very strange choice to not use the same periods for each of the models being tested. I understand that there are computational costs to be accounted for, why not assess each model for the 1998-2001 period?

**Reply:** Thanks for pointing out this. To avoid any confusion, we choose the 5-year period from 1999-01-01 to 2003-12-31 for prognostic simulations of SPCAM, NNCAM and CAM5.

L600 Table 2. "Number of samples trained per iteration". Are the authors referring to batch size here? "Number of rounds to traverse the data set". Sorry, this is unclear to me. Is this stating that the training dataset contains 50 batches of 1024?

**Reply:** Thanks for the comment. *"Number of rounds to traverse the data set"* means epochs, The description of our training process was not clear enough, and we apologize for that. We have reorganized the training process of NNs, please refer to the revised manuscript in lines 205-216.

*L620: "Note: Spatial averaging of MSE is performed before calculating R2." This is unclear. Could the authors please explain further.*

**Reply:** Thank you for the comment. We just want to emphasize that the mean square errors from samples that globally are and weighted equally to calculate the total mean square error, and that the variance is also calculated across all samples. The we derive  $R^2$  via  $R^2 = 1 - mse/var$ .

Figure 7: It would be very interesting to also plot the R2 values for the CAM5 parametrization as a model for the SP.

**Reply:** Thank you for the suggestion. We have added the  $R^2$  for CAM5 parameterization as a baseline in the new Figure 7. In offline validation, it is clear that NN parameterization is closer to the SP much better than the traditional CAM5 moist parameterizations.

Figure 8: There appear to be negative R2 values in portions of the globe. This is a worryingly low skill for the model.

**Reply:** Thanks. Our NN parameterization is trained with the loss function of mean squared error, which is not sensitive to incorrect predictions of small values. In Figure R1b, the local variance/std is close to zero for those low skill regions. The MSE in those regions is also low but is still high compared with its variance. Therefore, when calculating  $R^2$  as 1-mse/var, many of those low std regions will have  $R^2$  close to zero. (please check Figure R1)

Figure 9: I think this figure could strongly benefit from a companion figure where the differences from the SPCAM run are shown for both CAM5 and NNCAM. Otherwise is it challenging to decipher if NNCAM lies closer to the SPCAM mean state than CAM5. I also think it would be very interesting to compare all of these runs to the ERA5 state of the atmosphere for those years. This would go towards answering the question of whether SP is an improvement over CAM5.

**Reply:** Thank you for the suggestion. We have added the differences between SPCAM and NNCAM or CAM5 in the new *Figure S2*. NNCAM actually simulate more biased climate states than CAM5 compared with SPCAM. We also compare the mean states of temperature and humidity from SPCAM with ERA-Interim. It turns out that SPCAM simulate a colder and wetter climate. Its improvements over CAM5 is limited in terms of climate states (Figure R4).

Figure 10: As with figure 9, I think showing the differences would add significant information.

**Reply:** Thank you for the suggestion. We have added the differences in the new Figure 11 of the revised manuscript.

Figure 11: My interpretation of this plot is that NNCAM is a worse model of SPCAM than CAM5. Do the authors agree, and if so, why do they think this is true?

**Reply:** Thank you for the question. We have clarified the pros and cons of NNCAM in your first major comment. NNCAM indeed carries some biases in mean states and we admitted that in the revised manuscript. The winter precipitation biases of NNCAM is most significant, we believe the spatial difference plots add more information than the zonal averaged plots. Therefore, we have replaced the old Figure 11 with the differences plot. For now, the biggest error is the underestimated rainfall along the equator in boreal winter. From the 2m temperature in Figure R4, the high latitude regions in both southern and northern (especially the northern) hemisphere are too warm in NNCAM. Therefore, anomalous northwest surface wind stress is found on the north of the equator in the western Pacific, making the ITCZ shift north in DJF (Figure R5). Also an easterly intrusion is found in the location between the ITCZ and SPCZ.

Figure 14: As with figure 11. It is not clear that NNCAM has succeeded at this task.

**Reply:** Thank you for the comment. In the revised new Figure 11, we plotted it with precipitation and U200 from the new 5-year simulations for SPCAM, NNCAM, and CAM5. SPCAM and NNCAM show eastern propagation signals over Indian Ocean and Maritime Continent while CAM5 shows the opposite. The  $R^2$  for precipitation and U200 are below zero in NNCAM, but they are higher than those in CAM5 where the  $R^2$  for U200 is as low as -0.74.

**Reference:**

- Crawshaw, M.: Multi-task learning with deep neural networks: A survey, arXiv preprint arXiv:2009.09796, 2020.
- Khairoutdinov, M., Randall, D., and DeMott, C.: Simulations of the Atmospheric General Circulation Using a Cloud-Resolving Model as a Superparameterization of Physical Processes, Journal of the Atmospheric Sciences, 62, 2136-2154, 10.1175/jas3453.1, 2005.
- Zhang, Y. and Yang, Q.: A Survey on Multi-Task Learning, IEEE Transactions on Knowledge and Data Engineering, 1-1, 10.1109/TKDE.2021.3070203, 2021.

---

## Referee Report (RR1)

**1    Summary**

Overall, the authors have addressed my comments. After reading the manuscript I still have a major comment on the manuscript regarding how the authors have reported their results. Overall, I think that the manuscript deserves publication but I would like to get a clarification from the author regarding why there are inconsistencies in the results between the different versions of the manuscript.

**2    Major Comments**

I might be misinterpreting what the authors have been writing, but I think that the authors might be using some inconsistent reporting when they present their data. Mostly I am worried since during the evolution of the paper between iterations, there are some inconsistencies in the text and figures, and at least to my understanding the authors did not clarify these differences. I suggest that the authors would clarify these inconsistencies.

My main concern is rooted in how the authors report and write about their methodology of trial and error and in the large differences between the results presented in the first version of the manuscript and the recent version of the manuscript.

- Results in Figures 12 and 14. The results presented in figure 12 shows overall that NNCAM is performing better than CAM in terms of latitudinal distribution of precipitation. However, in the first version of the manuscript a similar figure was shown (figure 11 in the first version) and in the first version NNCAM was performing substantially worse than CAM. How come this result was changed? Did the authors change something that caused this difference, and if yes where was it reported? I know that now the analysis is performed over more years, but I doubt that this can explain the large differences between these results. (A similar comment on figure 14 that is now different compared to figure 13 in the first version of the manuscript).

- Results from online tests. In the first version of the manuscript the authors wrote that: "prepared 50 groups of ResMLPs with similar R2 as candidate models using different train samples and epochs. Secondly, we conducted comprehensive prognostic tests on these candidate neural networks and obtained the feasible NN-Parameterization schemes that can support NNCAMs stable simulation for multiple years."

  From that statement it is clear that the authors have run online simulations with at least 50 NNs and some of them were stable. Since this is an important part of the authors' work (where they also argue that a fully connected DNN are not stable and therefore they use a different architecture) in the first revision I requested that they give more details about how many networks were stable and how many were not (for each type of NN that they tried). In the response to my first review the authors wrote that they add:

"Figure 4 shows the offline validation versus the number of prognostic steps that our NNCAM can run. First, the DNN parameterizations are less accurate than the ResDNN ones in terms of offline validation accuracy. As a results, all the DNN parameterizations cannot run stably longer than half a year in prognostic tests. For the ResDNNs (blue dots and black inverted triangles), the less well-trained ones with high MSE crash within half a year simulation. However, when the offline MSE of ResDNN decreases to a certain level the ResDNN parameterization may run stably for long periods. In Figure 4, we observed 4 ResDNNs can run stably."

In the figure that the authors added they show 3 DNN and 21 ResDNNs. In the initial version of their manuscript the author wrote that they have prepared and tested 50 groups of networks so I am not sure why they report only on part of their results.

Furthermore, since I thought that the comparison between 21 runs (out of the 21 runs of ResDNN only 4 of the runs were stable) and 3 runs of DNN cannot support the claim that DNNs aren't stable is not established since only very few networks DNN were examined. In a response for my comment the authors included in their latest response:

"We tested all 37 NN sets (27 ResDNN sets and 10 DNN sets) in the sensitivity tests. As shown in Figure 4, there are 10 ResDNN sets that can sustain simulations of longer than 10 years. Figure 4 is intended to show the relationship between the MSEh and the stability, not to prove that the ResDNN is better than the DNN in terms of stability."

What I found strange is that they have added 3 additional simulations with the ResDNN case, and found all additional simulations to be stable (I can count only 24 ResDNN and not 27; BTW, the addition of DNNs makes the argument much better). I find this result of adding 3 additional ResDNN simulations and finding them to be all stable is not very likely given that they previously reported to find that only 4 out of 21 simulations were stable but maybe I am missing something here, so it would be great if the authors could clarify

I would encourage the authors to include all the results they have obtained and reported in the first version of the manuscript (e.g., could the authors provide the results of the 50 simulations they reported in the first version of the manuscript? Is there a reason that these results are not included?)

**3   minor comments**

- lines 100-102: The authors write that the Yuval and O'gorman 2020 used random forest to predict fluxes to ensure physical constraints, but this is not correct. As far as I understand the usage of random forest allows to ensure linear physical constraints because RF just averages over samples.

- line 168: the authors use the terms qrs and qrl but they never define these.

---

## Editor Decision (ED1)

**1   Summary**

The authors have addressed some of my comments and added some sections to the manuscript. The authors did add some analysis that substantially improve the manuscript. After reading the manuscript I still have several major comments on the manuscript, and I think the manuscript is not ready for publication yet. In case the authors intend to resubmit, please clearly highlight in the response all the important changes that were made in the manuscript since I felt such changes were not highlighted enough in their previous response.

Overall, I think that the manuscript deserves publication but still needs major revisions before it can be published.

**2   Major Comments**

- I still think the authors overstate their achievement. The authors highlight many times that their NNCAM is more accurate than CAM. I think that in a few aspects it is true, but in many aspects it is not. The authors should highlight also the parts that NNCAM fails. e.g., the climatology (which is a crucial thing in simulations of climate) is not accurate (e.g., RMSE in temperature is  3 degrees with certain regions getting to 10 degrees in the zonal mean!  which is much larger than CAM RMSE). Please clearly mention the large biasses in climatology (e.g., in the abstract).  For example, I do not understand why "most importantly, NN parameterization successfully reproduces the climate variability in a superparameterized GCM" - is it more important than achieving the correct climatology?  Furthermore, the large errors in the climatology (e.g., of temperature) should be shown in the main text and not in the SI since it is a key problem in the simulations and should be highlighted. In addition, in the previous version the authors showed (Figure 11 in previous version) results for precipitation such that it was easy to see that SPCAM is less accurate than CAM in many precipitation features, please keep the figure in the manuscript.

- The authors write that a DNN cannot run stably. However, I do not think that this claim is backed up by scientific evidence. The authors ran 3 different DNN and found them unstable (though with lower accuracy they could run longer than the RESNETs). The authors tested 10s (over 50 if I understand correctly) of ResNets and found only a few of them stable. If the authors want to argue that DNN cannot be run stably, they should follow the same methodology as with the ResNets and train 10s of DNNs to show that they do not run stably. I am also not convinced yet that a that the key for stable runs is accurate NNs since some NNs are more accurate than your stable networks and they are still unstable. If the authors want to argue this they should provide better evidence. Otherwise they should clearly write that this is a hypothesis that they have not yet provided clear evidence for.

- The authors write that they do not couple the NN radiation in the sensitivity tests (lines 272-275).  I am surprise by this statement as it did not exist in the previous

version of the manuscript and was not mentioned in their response to my review. I am not sure I understand correctly what it means. Does the radiation scheme run in the prognostic simulations? If not, this is very strange to me because the text discusses several times (even in the implementation section) how they implemented NN radiation. Furthermore, the reason that the authors give why they do not use the NN radiation in the prognostic simulations is that it is very accurate. I do not think that this is a convincing reason (especially since if the NN radiation is very good, I think it is important to show that it works also in a prognostic test). Also, the argument that the NN radiation is very accurate is based on a zonally averaged result so please include the $R^2$ result before using a zonal average. Overall, the authors should show online results when their NNs change SP (including radiation). If they decide not to use their NN radiation in prognostic tests please provide a convincing explanation why (does simulations have climate drift like in the previous version of the manuscript when radiation is used? do online results similar to what is presented). I note that It is possible that I misunderstood what the authors did with radiation.

- The authors ran several stable NNs, and from what I understand they choose the "best" (line 303-305: "we still have to use the trial-and-error to filter out unstable ones and then select the best ResDNN pair for moistening and heating rate that can reduplicate the total energy time evolution of SPCAM with the least deviation"). I feel that this is a bit of cherry picking and I am not sure what to think about this process. Maybe this should be highlighted in the abstract that for no clear reason there are large differences in the results of stable NNs and that you have trained multiple networks and present only the "best" results.

**3  Comments**

- The authors wrote that in SPCAM it is not possible to separate different processes. Why? As far as I understand, the authors could keep track on the different processes that run in the SP (which is SAM model) and model each process separately.

- The authors write that "In our study, the NN parameterizations are tendency-based trained with realistic configuration SPCAM simulation without any physical-constrain, where stability is indeed a problem to face". I disagree with this statement. There are many constraints in the parameterization. Their SP is SAM based model. Each process (e.g., subgrid convection, microphysics etc.) has certain constraints and physical relationships within each process.

- The authors should discuss negative precipitation in detail in the manuscript. They write in thier response that 27 percent of the time NN give negative precip but they do not mention in it in the manuscript.

- In the answer to my review, there are several times the authors respond in a manner that is not related to what I have asked. (e.g., I wrote "Is there some citation that can

backup the statement that MLP can generalize better than other types of networks? If not please remove statement.")

- I insist that the authors will give the results for $R^2$ before zonal averaging. It will help to understand how accurate the networks are.

- The authors write in their response: "Our NN parameterization is trained with the loss function of mean squared error, which is not sensitive to incorrect predictions of small values. In Figure R1b, the local variance/std is close to zero for those low skill regions. The MSE in those regions is also low but is still high compared with its variance. Therefore, when calculating R2 as 1-mse/var, many of those low std regions will have R2 close to zero." However, the authors have low skill in the tropics where STD is large.

- There are still unclear citations for me. For example the author cite Moores et al. 2020 (and no such reference exist in their bibliography).

- The authors compare the offline results of their NN to SP and of a conventional parameterization to SP. They should highlight that this is not exactly a fair comparison. Their NN was tuned to emulate the SP, and CAM parameterizations where tuned to get a better online results (so the SP is not a ground truth for CAM).

- there are still many sentences in the manuscript that do not have context - e.g., "Brenowitz et al. (2020) proposed methods to interpret and stabilize ML parameterization of convection. In their work, a wave spectra analysis tool was introduced to explain why ML coupled GCMs blew up."

- - It is unclear to me how variables were normalized - did each output was normalized separately at each level? If yes - didn't it lead to problems in regions with very little subgrid values (e.g., there should be hardly any moisture and moisture tendencies in the stratosphere - how did you deal with that)

- The paragraph from line 185-196 has some statements I do agree with, but more importantly than that -I do not see why it is necessary to be included.

- The authors write "After numerous experiments" - can they please provide in the SI the hyperparameters that they used in their search?

- The author write that their NN predicts the temperature tendency (line 213). However, temperature is not a prognostic variable in SAM or in CAM so I do not see how it makes sense to modify the temperature with the NN and not the prognostic variables. Could the authors write what are the prognostic variables they are actually changing in the simulations.

- In line 217 there is a reference to figure 1 but I think the text should not refer to this figure

- In their response - the authors write "random forest is less likely to perform as accurately as neural networks and cannot be implemented in GPUs (Yuval et al., 2021)" however, I think that this statement is not correct and RFs were already used with GPUs

- The moist static energy should include a term $gz$ ($g$ is the acceleration due to gravity and z is the height), and currently the moist static energy is not written correctly.

- line 313: dump->damp

- line 379: The authors refer to figure S2 (line 380) and say it shows RMSE - but I do not think it shows this. Please add RMSE to the figure. F

I thank the authors for their detailed reply and significant changes to the manuscript. I now have a better understanding of where ResDNN exceeds the performance of CAM and where it does not yet do so. I believe that the article is close to acceptance but would like to see a few more improvements.

From author replies:

"As for swapping neural networks, we do not change the neural network for the 8 radiation fluxes because they are highly accurate and well trained    with a collaborate R2 above 0.98."

So for each of the dots in figure 4, the same network is used for the 8 fluxes? This information should please be included in the manuscript (or highlighted if it is already included).

"the tendencies of temperature and moisture are rather difficult to train and, if not trained well or with the right NN architecture, can seriously affect the prognostic performance and stability. So, we swap the neural networks for dqv and dT together but not individually"

Sorry. I do not understand this important point. To my understanding all the configurations producing dots in figure 4 result from the same loss function and architecture. But each time the temperature NN is trained separately from the moisture NN. In this case why could they not be interchanged? Would it not be fairly simple to take the most accurate stable configuration, the most accurate unstable configuration, and exchange networks and see the impact. I think this is a reasonable request in order to elucidate where the destabilisation is originating.

L19: It is worth stating that these biases are larger than those found in CAM5.

L176: I do not agree with the authors' assessment here. To me the calculation of moisture and temperature tendencies resulting from moist processes does not count at multi-task learning but as multi-output regression (see figure 1 of Zhang). The authors may state that they chose to split the learning of these two outputs, but I do not believe the work of Crawshaw or Zhang and Yang gives any evidence that the task will be easier by doing so. If the authors still hold that Crawshaw and Zhang & Yang provide clear evidence that this task will be harder because of the multiple outputs could they please reference where in these papers this is discussed. Both papers are surveys, without any strong overall argument that MTL is flawed or more difficult. I would request that the authors change this section, as it could negative influence future research. In my mind it is future research to establish if splitting the task between two models (a) results in lower offline scores (b) produces more stable coupled results.

Section 2.2.2. I found it hard to establish how many neural networks are built and what they are each learning. Please could this be made more clear in the text. In the original version of the manuscript there was evidence that 4 NN were built (e.g. the now removed figure 2). But I find no explanation of the change. Did the authors change their approach, if so, why?

L456: successes -> succeeds

Figure 3: Several of the numbers mentioned in the caption do not appear in the figure.

Figure 5. Is there a reason you do not include CAM in this figure?

Figure 9. This caption could now be tidied as all panels represent the same period.

Figure 10, as with 9.

Figure 11: What bias amount does the white colour represent in panel (d)? Are the colours correct in panel (c)? There seems to be a global bias of between 2 and 4mm, based on the colours, yet the number reported in the top is 0.148. Personally, I would recommend using a neutral colour, e.g. white, for errors less than some threshold, e.g. 0.5mm. Otherwise the eye is drawn to places where the bias changes from positive to negative even when such a change is miniscule. But the authors are free to ignore this suggestion if they disagree.

---

## Author Response (AR2)

**Response to reviewer 1**

**1 Summary**

*The authors have addressed some of my comments and added some sections to the manuscript. The authors did add some analysis that substantially improve the manuscript. After reading the manuscript I still have several major comments on the manuscript, and I think the manuscript is not ready for publication yet. In case the authors intend to resubmit, please clearly highlight in the response all the important changes that were made in the manuscript since I felt such changes were not highlighted enough in their previous response.*

*Overall, I think that the manuscript deserves publication but still needs major revisions before it can be published.*

Thank you for carefully reviewing our manuscript. We have tried our best to respond to all of the comments below and have made proper revisions to the manuscript. The reviewers' comments are in italics and our responses are in normal blue font. For clarification, we have made changes to the figures, and there are 15 figures in the revised manuscript, and 5 figures and 2 movies in the supplementary materials. We have revised Figures 4 and 5, added Figure S2 (a plot of mean state differences) in the last manuscript as Figure 10 now. Figure 11 shows the global distribution of the mean precipitation, and the zonally averaged precipitation is shown in Figure 12. Figures 12–14 in the previous manuscript are now Figures 13–15. We also added a plot of the cross-NN precipitation STD (Figure S1), and the original plot of the precipitation differences is now Figure S5. For the convection and combined radiation heating rate, we no longer use the phrase temperature tendency. Now, we use the tendency of the dry static energy.

**2 Major Comments**

- *I still think the authors overstate their achievement. The authors highlight many times that their NNCAM is more accurate than CAM. I think that in a few aspects it is true, but in many aspects it is not. The authors should highlight also the parts that NNCAM fails. e.g., the climatology (which is a crucial thing in simulations of climate) is not accurate (e.g., RMSE in temperature is 3 degrees with certain regions getting to 10 degrees in the zonal mean! which is much larger than CAM RMSE). Please clearly mention the large biasses in climatology (e.g., in the abstract). For example, I do not understand why "most importantly, NN parameterization successfully reproduces the climate variability in a superparameterized GCM" - is it more important than achieving the correct climatology? Furthermore, the large errors in the climatology (e.g., of temperature) should be shown in the main text and not in the SI since it is a key*

*problem in the simulations and should be highlighted. In the previous version the authors showed (Figure 11 in previous version) results for precipitation such that it was easy to see that SPCAM is less accurate than CAM in many precipitation features, please keep the figure in the manuscript.*

**Reply:** Thank you for your comments. For the parts where NNCAM fails, we have clearly highlighted the climate biases in the abstract as follows: *"However, there are still substantial biases in the mean states, including the temperature field in the tropopause and at high latitudes and the precipitation over tropical oceanic regions, which are larger than those in CAM5."* We have moved the plots of the temperature differences from Figure S2 to Figure 10, have added clear statements about temperature biases into Section 5.1.1 and precipitation biases into Section 5.1.2. The original Figure 11 is now Figure 12, and the TRMM precipitation has been added to supplement SPCAM and CAM5. The plot of the precipitation difference is now Figure S5. These two figures show that NNCAM produces larger precipitation simulation biases over the equatorial tropical ocean in boreal winter. However, both SPCAM and NNCAM produce better rainfall simulations than CAM5 over the land, which is related to the SPCAM's better-simulated convection variability.

As for the target model SPCAM, which uses the 2-D SAM as the SP, SPCAM does not significantly improve the climate mean states (Khairoutdinov et al., 2005; Kooperman et al., 2016). We have made such statements in lines 62-63 in the introduction and in lines 406-407 in Section 5. However, SPCAM improves the model's climate variability, as well as the tropical precipitation over the land, which is shown by Figures 12–15 and is consistent with the findings of other studies (see the SPCAM part in the introduction). The NNCAM also retains these advantages and succeeds in these aspects.

We have also added lines 407–410: *"What is remarkable about NNCAM is not its performance in simulating the mean climate, but its ability to achieve a stable multi-year prognostic simulation under a real-world global land-ocean distribution. The advantages and problems of this study will provide important references for future research on NN-based stable long-term model integrations."*

- *The authors write that a DNN cannot run stably. However, I do not think that this claim is backed up by scientific evidence. The authors ran 3 different DNN and found them unstable (though with lower accuracy they could run longer than the RESNETs). The authors tested 10s (over 50 if I understand correctly) of ResNets and found only a few of them stable. If the authors want to argue that DNN cannot be run stably, they should follow the same methodology as with the ResNets and train 10s of DNNs to show that they do not run stably. I am also not convinced yet that a that the key for stable runs is accurate NNs since some NNs are more accurate than your stable networks and they are still unstable. If the authors want to argue this they should provide better evidence. Otherwise they should clearly write that this is a hypothesis that they have not yet provided clear evidence for.*

**Reply:** Thank you for the comments. We tested all 37 NN sets (27 ResDNN sets and 10 DNN sets) in the sensitivity tests. As shown in Figure 4, there are 10 ResDNN sets that can sustain simulations of longer than 10 years. Figure 4 is intended to show the relationship between the $MSE_h$ and the stability, not to prove that the ResDNN is better than the DNN in terms of stability.

The stability of the NN-Parameterization still requires further study. We have also made changes to the abstract by only stating that "*We explore the relationship between the accuracy and stability by validating multiple Deep Neural Network (DNN) and ResDNN sets in prognostic runs.*" We have added a statement in lines 313–314: *" We speculate that the more accurate ResDNN sets have a higher probability of becoming stable NN-Parameterizations since all of the stable NN-Parameterizations are ResDNNs."*

- *The authors write that they do not couple the NN radiation in the sensitivity tests (lines 272-275). I am surprise by this statement as it did not exist in the previous version of the manuscript and was not mentioned in their response to my review. I am not sure I understand correctly what it means. Does the radiation scheme run in the prognostic simulations? If not, this is very strange to me because the text discusses several times (even in the implementation section) how they implemented NN radiation. Furthermore, the reason that the authors give why they do not use the NN radiation in the prognostic simulations is that it is very accurate. I do not think that this is a convincing reason (especially since if the NN radiation is very good, I think it is important to show that it works also in a prognostic test). Also, the argument that the NN radiation is very accurate is based on a zonally averaged result so please include the R2 result before using a zonal average. Overall, the authors should show online results when their NNs change SP (including radiation). If they decide not to use their NN radiation in prognostic tests please provide a convincing explanation why (does simulations have climate drift like in the previous version of the manuscript when radiation is used? do online results similar to what is presented). I note that It is possible that I misunderstood what the authors did with radiation.*

**Reply:** The reviewer may have misunderstood what we did with the radiation. This is probably due to the confusing text in the previous version of the manuscript. We used three neural networks for (1) the moistening rate, (2) the combined heating rate of the convection and radiation, and (3) the radiation fluxes at the surface and the TOA (for more details please see lines 192–198). Each of them has the same input, the same network structure and the same hyperparameters, but different outputs. Therefore, the radiative heating is part of the NN-Parameterization; and it is only not treated separately from the heating due to the moist physics. The radiation fluxes at the surface and the TOA are also predicted using the NN and are updated to the model. In addition, regarding our $R^2$ calculations, we directly applied the equation $R^2 = 1 - \frac{mse}{var}$ for all

of the considered fields without any preprocessing, except in Figure 7. Specifically, Figure 2 uses a *"raw"* $R^2$ for both time and space. Figure 7 uses the $R^2$ for the zonally averaged fields, and Figure 8 uses the $R^2$ for the time sequence at each location. The proper information has been added to the captions of the corresponding figures.

- *The authors ran several stable NNs, and from what I understand they choose the "best" (line 303-305: "we still have to use the trial-and-error to filter out unstable ones and then select the best ResDNN pair for moistening and heating rate that can reduplicate the total energy time evolution of SPCAM with the least deviation"). I feel that this is a bit of cherry picking and I am not sure what to think about this process. Maybe this should be highlighted in the abstract that for no clear reason there are large differences in the results of stable NNs and that you have trained multiple networks and present only the "best" results.*

**Reply:** Thank you for pointing this out. As for the concern of cherry picking and the large differences in the results of the stable NNs, we have highlighted this in the abstract as follows: *"We explore the relationship between the accuracy and stability by validating multiple Deep Neural Network (DNN) and ResDNN sets in prognostic runs. In addition, there are significant differences in the prognostic results of the stable ResDNN sets. Therefore, trial-and-error is used to acquire the optimal ResDNN set for both high-skill and long-term stability, which we name the NN-Parameterization."* We have also added similar statements in lines 324–327: *"Apart from global averages, the prognostic results of the 10 stable ResDNN sets vary from each other in terms of the global distribution. Figure S1 shows the precipitation spread across all of the stable NN sets for the prognostic simulation from 1999 to 2003. The obvious standard deviation centers coincide with the heavy tropical precipitation areas."*

**3 Comments**

• *The authors wrote that in SPCAM it is not possible to separate different processes. Why? As far as I understand, the authors could keep track on the different processes that run in the SP (which is SAM model) and model each process separately.*

**Reply:** Thank you for pointing this out. We agree with the reviewer that one can separate the different processes such as advection, diffusion, microphysics, and ice sedimentation in the SP. However, in SPCAM under the multiscale model framework, the SP is a 2-D SAM and the host model is CAM5, and the accumulated tendencies across all of the processes are used to exchange information between the SAM and the host model. Therefore, to emulate the SAM in the simplest way, we directly used the accumulated tendencies without tracking the different processes. However, this is an interesting idea. We plan to study it in our future work.

• *The authors write that "In our study, the NN parameterizations are tendency-based trained with realistic configuration SPCAM simulation without any physical-*

*constrain, where stability is indeed a problem to face". I disagree with this statement. There are many constraints in the parameterization. Their SP is SAM based model. Each process (e.g., subgrid convection, microphysics etc.) has certain constraints and physical relationships within each process.*

**Reply:** Thank you for pointing this out. We think it is a misunderstanding because of an unclear sentence. We agree with the reviewer on this issue. What we intended to say is that our NN-Parameterization does not have any imposed physical constrains.

*• The authors should discuss negative precipitation in detail in the manuscript. They write in their response that 27 percent of the time NN give negative precip but they do not mention in it in the manuscript.*

**Reply:** Thank you for the comment. We have added the descriptions in lines 165–170. *"In this study, we used the vertical integration of the NN predicted moisture tendency as an approximation of the surface precipitation, which has also been used in previous studies (e.g., O'Gorman et al., 2018; and Han et al., 2020). In the offline validation test, we observed negative precipitation events (27% occurrence in 1-year of results). Nonetheless, 93% of the negative precipitation events had a magnitude of less than 1 mm/day. In the online prognostic runs, reasonable rainfall results (more details will be provided in Section 5) were achieved using this approximation scheme."*

*• In the answer to my review, there are several times the authors respond in a manner that is not related to what I have asked. (e.g., I wrote "Is there some citation that can backup the statement that MLP can generalize better than other types of networks? If not please remove statement.")*

**Reply:** Thank you for the comment. We had already removed the claim that the *"MLP can generalize better"* in the last revision of the manuscript. We also briefly introduced why the DNN and the ResDNN were used in this study in lines 203–210.

*• I insist that the authors will give the results for $R^2$ before zonal averaging. It will help to understand how accurate the networks are.*

**Reply:** Thank you for the comment about the $R^2$ calculations. Except for in Figure 7, we did not apply zonal averaging in any of the $R^2$ calculation (Figures 2 and 8). For Figure 7, we calculated $R^2$ of zonally averaged values because we compared this figure to those in Mooers et al. (2021), and thus, the same method needed to be used. We believe this is sufficient to show the accuracy of the moistening and heating. Since the reviewer expected to calculate $R^2$ before zonal averaging, we calculated the $R^2$ for every single position in the 3-D space over its own time series and then zonally averaged them into a latitude-pressure cross section in Figure R1.

*• The authors write in their response: "Our NN parameterization is trained with the loss function of mean squared error, which is not sensitive to incorrect predictions of*

*small values. In Figure R1b, the local variance/std is close to zero for those low skill regions. The MSE in those regions is also low but is still high compared with its variance. Therefore, when calculating $R^2$ as 1-mse/var, many of those low std regions will have $R^2$ close to zero."* However, the authors have low skill in the tropics where STD is large.

**Reply:** Thank you for pointing this out. We agree with the reviewer that the NN-Parameterization also produces low-skill regions where the STD is large. The explanation in the last response is only for some locations near the subtropical southeast ocean areas and therefore does not apply to the regions near the equator. Therefore, we have removed this argument from the manuscript. Instead, we have added the following text in lines 390–393: *"Still, our NN-Parameterization produced low accuracy predictions along the equator over the oceans where the convection is complex and vigorous and in subtropical ocean areas where the convection is weak and concentrated at low levels. This indicates that the NN-Parameterization is still inadequate in rems of its emulation skill when simulating various types of deep and shallow convection in the tropics."*

• *There are still unclear citations for me. For example the author cite Moores et al. 2020 (and no such reference exist in their bibliography).*

**Reply:** Thank you for the comment. We are sorry for the typo in this citation. We actually cited Moores et al. (2021), and this has been corrected in the latest version of the manuscript.

• *The authors compare the offline results of their NN to SP and of a conventional parameterization to SP. They should highlight that this is not exactly a fair comparison. Their NN was tuned to emulate the SP, and CAM parameterizations where tuned to get a better online results (so the SP is not a ground truth for CAM).*

**Reply:** We thank the reviewer for their insightful comment. The second reviewer asked for this comparison. We have added the sentence *"It should be noted that the NN-Parameterization was tuned to emulate the SP, and CAM's Parameterization was tuned to obtain close results compared to the observations."* in lines 354–355 before comparing NN-Parameterization with the conventional CAM5 parameterizations through offline tests.

• *There are still many sentences in the manuscript that do not have context - e.g., "Brenowitz et al. (2020) proposed methods to interpret and stabilize ML parameterization of convection. In their work, a wave spectra analysis tool was introduced to explain why ML coupled GCMs blew up."*

**Reply:** Thank you for this comment. We have checked and corrected these sentences in the latest version of the manuscript by changing it to *"To determine why some methods can achieve stable prognostic simulations and others cannot, Brenowitz et al.*

*(2020) proposed methods for interpreting and stabilizing ML parameterization for convection. In their study, a wave spectra analysis tool was introduced to explain why the ML coupled GCMs blew up."* (see lines 93–95)

• *It is unclear to me how variables were normalized - did each output was normalized separately at each level? If yes - didn't it lead to problems in regions with very little subgrid values (e.g., there should be hardly any moisture and moisture tendencies in the stratosphere - how did you deal with that).*

**Reply:** Thank you for this comment. All of the variables with vertical levels are normalized as a whole using the maximum value following the method of Han et al. (2020), rather than being normalized separately at each level. We have added this statement in line 175.

• *The paragraph from line 185-196 has some statements I do agree with, but more importantly than that - I do not see why it is necessary to be included.*

**Reply:** Thank you for the comment. We have simplified this paragraph, and now briefly and directly explain why a DNN and ResDNN were used in this study (see lines 203–210).

• *The authors write "After numerous experiments" - can they please provide in the SI the hyperparameters that they used in their search?*

**Reply:** Thank you for the comment. We mainly tried to use the weight decay, learning rate, width and depth of the neural network, batch size, dropout, activation function, and other hyperparameter combinations. We have sorted this out and have added it to Table S1 in the SI.

• *The author write that their NN predicts the temperature tendency (line 213). However, temperature is not a prognostic variable in SAM or in CAM so I do not see how it makes sense to modify the temperature with the NN and not the prognostic variables. Could the authors write what are the prognostic variables they are actually changing in the simulations.*

**Reply:** Thank you for the comment. In fact, it should be the tendency of the dry static energy not the tendency of the temperature. We have changed all of the temperature tendency to the tendency of the dry static energy to represent the combined convective and radiative heating in the latest version of the manuscript. According to the appendix in Khairoutdinov and Randall (2003), the SAM predicts the changes in the liquid ice/water energy $[h_L = C_p T + gz - L_c(q_c + q_r) - L_s(q_i + q_s + q_g)]$ and the total nonprecipitating water $q_T$ (water vapor $q_v$ + cloud water $q_c$ + cloud ice $q_i$). The host GCM (CAM5) uses the tendencies of the dry static energy $ds$ and water vapor $dq_v$, and it updates them to the states of the temperature, water vapor, and layer heights via the *physics_update* subroutine. There is a process in the 2-D

SAM codes that derives ds from the changes in $h_L$ and $dq_v$ and from the changes in $q_T$.

• *In line 217 there is a reference to figure 1 but I think the text should not refer to this figure.*

**Reply:** Thank you for the comment. We have removed the reference in Figure 1. Please see lines 230-231.

• *In their response - the authors write "random forest is less likely to perform as accurately as neural networks and cannot be implemented in GPUs (Yuval et al., 2021)" however, I think that this statement is not correct and RFs were already used with GPUs.*

**Reply:** Thank you for the comment. Although Yuval et al. (2021) stated that *"random forest is less likely to perform as accurately as neural networks and cannot be implemented in GPUs"*, we agree that the random forest has been implemented and used on GPUs.

• *The moist static energy should include a term gz (g is the acceleration due to gravity and z is the height), and currently the moist static energy is not written correctly.*

**Reply:** Thank you for the insightful comment. Based on the answer *"NN predicts the temperature tendency,"* we have changed $dT$ to $ds$. Therefore, we have corrected the moist static energy as follows: $h = C_p T + gz + L_v q_v = s + L_v q_v$.

• *Line 313: dump->damp*

**Reply:** Thank you for pointing this out. We have corrected this in the latest version of the manuscript.

• *Line 379: The authors refer to figure S2 (line 380) and say it shows RMSE - but I donot think it shows this. Please add RMSE to the figure.*

**Reply:** Thank you for pointing this out. Actually, the parentheses in Figure S2 are merely for the phrase *"larger differences than CAM5"*. This is indeed an unclear sentence. So we have moved Figure S2 showing the state differences to Figure 10, and we revised the sentence as follows: *"However, the multi-year mean temperature and moisture fields produced by NNCAM are more biased than those produced by CAM5, which is reflected by the larger root mean square errors (RMSEs) (Figure 9) and larger differences compared to those of CAM5 (Figure 10)."*

**Reference:**

Brenowitz, N. D., Beucler, T., Pritchard, M., and Bretherton, C. S.: Interpreting and Stabilizing Machine-Learning Parametrizations of Convection, Journal of the Atmospheric Sciences, 77, 4357-4375, 10.1175/jas-d-20-0082.1, 2020.

Han, Y., Zhang, G. J., Huang, X., and Wang, Y.: A Moist Physics Parameterization Based on Deep Learning, Journal of Advances in Modeling Earth Systems, 12, e2020MS002076, 10.1029/2020ms002076, 2020.

Khairoutdinov, M. F. and Randall, D. A.: Cloud Resolving Modeling of the ARM Summer 1997 IOP: Model Formulation, Results, Uncertainties, and Sensitivities, Journal of the Atmospheric Sciences, 60, 607-625, 2003.

Khairoutdinov, M., Randall, D., and DeMott, C.: Simulations of the Atmospheric General Circulation Using a Cloud-Resolving Model as a Superparameterization of Physical Processes, Journal of the Atmospheric Sciences, 62, 2136-2154, 2005.

Kooperman, G. J., Pritchard, M. S., Burt, M. A., Branson, M. D., and Randall, D. A.: Robust effects of cloud superparameterization on simulated daily rainfall intensity statistics across multiple versions of the Community Earth System Model, Journal of Advances in Modeling Earth Systems, 8, 140-165, 10.1002/2015ms000574, 2016.

Mooers, G., Pritchard, M., Beucler, T., Ott, J., Yacalis, G., Baldi, P., and Gentine, P.: Assessing the Potential of Deep Learning for Emulating Cloud Superparameterization in Climate Models With Real-Geography Boundary Conditions, Journal of Advances in Modeling Earth Systems, 13, e2020MS002385, https://doi.org/10.1029/2020MS002385, 2021.

Rasp, S., Pritchard, M. S., and Gentine, P.: Deep learning to represent subgrid processes in climate models, Proceedings of the National Academy of Sciences, 115, 9684-9689, 10.1073/pnas.1810286115, 2018.

Yuval, J., O'Gorman, P. A., and Hill, C. N.: Use of Neural Networks for Stable, Accurate and Physically Consistent Parameterization of Subgrid Atmospheric Processes With Good Performance at Reduced Precision, Geophysical Research Letters, 48, e2020GL091363, https://doi.org/10.1029/2020GL091363, 2021.

**Response to reviewer 2**

*I thank the authors for their detailed reply and significant changes to the manuscript. I now have a better understanding of where ResDNN exceeds the performance of CAM and where it does not yet do so. I believe that the article is close to acceptance but would like to see a few more improvements.*

**Reply:** Thank you for carefully reviewing our manuscript. We have tried our best to respond to all of the comments below and have made proper revisions to the manuscript. The reviewers' comments are in italics and our responses are in normal blue font. For clarification, we have made changes to the figures, and there are 15 figures in the revised manuscript, and 5 figures and 2 movies in the supplementary materials. We have revised Figures 4 and 5, added Figure S2 (a plot of mean state differences) in the last manuscript as Figure 10 now. Figure 11 shows the global distribution of the mean precipitation, and the zonally averaged precipitation is shown in Figure 12. Figures 12–14 in the previous manuscript are now Figures 13–15. We also added a plot of the cross-NN precipitation STD (Figure S1), and the original plot of the precipitation differences is now Figure S5. For the convection and combined radiation heating rate, we no longer use the phrase temperature tendency. Now, we use the tendency of the dry static energy.

*From author replies:*

*"As for swapping neural networks, we do not change the neural network for the 8 radiation fluxes because they are highly accurate and well trained with a collaborate R2 above 0.98."*

*So for each of the dots in figure 4, the same network is used for the 8 fluxes? This information should please be included in the manuscript (or highlighted if it is already included).*

**Reply:** Thanks for the comment. We use the same ResDNN for predicting 8 radiation fluxes in all sensitivity tests. The sensitivity test results in Figure 4 are for the different neural networks predicting the moisture tendency $dq_v$ and the dry static energy tendency $ds$. We have highlighted it in lines 298-299: *"First, we selected the best ResDNN for the radiation fluxes at the surface and the TOA that was shared in every NN set since their offline validation was exceptionally accurate with $R^2 > 0.98$ over 50 training epochs (Figure 2b)."*

*"the tendencies of temperature and moisture are rather difficult to train and, if not trained well or with the right NN architecture, can seriously affect the prognostic performance and stability. So, we swap the neural networks for dqv and dT together but not individually"*

*Sorry. I do not understand this important point. To my understanding all the configurations producing dots in figure 4 result from the same loss function and architecture. But each time the temperature NN is trained separately from the moisture NN. In this case why could they not be interchanged? Would it not be fairly simple to take the most accurate stable configuration, the most accurate unstable configuration, and exchange networks and see the impact. I think this is a reasonable request in order to elucidate where the destabilisation is originating.*

**Reply:** Thanks for the comment. In this research, the ResDNN/DNN predicting $dq_v$ and $ds$ have the same hyperparameter configuration. In theory, it is possible to combine the neural networks that predict $dq_v$ and $ds$ under different training configurations. It is possible to influence the NN-Parameterization stability by combining the moisture NN with the dry-static energy NN for different epochs. However, the permutations and combinations will bring too many test cases (50x50). We do not have enough computing resources to perform stability tests and climate state evaluations for all these NN combinations. Even with the current method, stability can be guaranteed. As a result, we are not going to address this in this work.

*L19: It is worth stating that these biases are larger than those found in CAM5.*

**Reply:** Thank you for the question. We have made such statements in the abstract: *"However, there are still substantial biases with the hybrid ML-physical GCM in the mean states, including the temperature field in the tropopause and at high latitudes and the precipitation over tropical oceanic regions, which are larger than those in CAM5."* and also in Section 5.1.

*L176: I do not agree with the authors' assessment here. To me the calculation of moisture and temperature tendencies resulting from moist processes does not count at multi-task learning but as multi-output regression (see figure 1 of Zhang). The authors may state that they chose to split the learning of these two outputs, but I do not believe the work of Crawshaw or Zhang and Yang gives any evidence that the task will be easier by doing so. If the authors still hold that Crawshaw and Zhang & Yang provide clear evidence that this task will be harder because of the multiple outputs could they please reference where in these papers this is discussed. Both papers are surveys, without any strong overall argument that MTL is flawed or more difficult. I would request that the authors change this section, as it could negative influence future research. In my mind it is future research to establish if splitting the task between two models (a) results in lower offline scores (b) produces more stable coupled results.*

**Reply:** Thanks for the comment. We had already revised *"multi-target"* to *"A ResDNN set"* in the last revised manuscript. We have also written a more precise paragraph in this revision in lines 184-202. Especially, we have changed the two original references to Yu et al. (2020) for the mutual interference in gradient descending in the revised manuscript. We believe that the separated training can avoid the mutual interference in

the gradient descending. Moreover, we agree with the reviewer that more future research should be done here.

*Section 2.2.2. I found it hard to establish how many neural networks are built and what they are each learning. Please could this be made more clear in the text. In the original version of the manuscript there was evidence that 4 NN were built (e.g. the now removed figure 2). But I find no explanation of the change. Did the authors change their approach, if so, why?*

**Reply:** Thanks for your comment. We did not change our approach in the revised manuscript. Figure 2 in the original manuscript is outdated and misleading. So we removed it and described our method verbally in Section 2.2.2. The ResDNN set contains three neural networks, which were trained to predict $dq_v$, $ds$ and eight radiation fluxes at the surface and the TOA, respectively. They all have the same hyperparameter configuration.

*L456: successes -> succeeds?*

**Reply:** Thanks for pointing this out. We have corrected it in the latest revised manuscript. For details, please see line 502.

*Figure 3: Several of the numbers mentioned in the caption do not appear in the figure.*

**Reply:** Thanks for pointing this out. There are some compatibility issues when compiling the pdf file. Several icons are missing in Figure 3. We have corrected them in the revised manuscript.

*Figure 5. Is there a reason you do not include CAM in this figure?*

**Reply:** Thanks for pointing this out. We have included CAM in the revised Figure 5.

*Figure 9. This caption could now be tidied as all panels represent the same period.*

**Reply:** Thanks for pointing this out. We have revised the caption of Figure 9 and presented it in the revised manuscript.

*Figure 10, as with 9.*

**Reply:** Thanks for pointing this out. We have revised the caption. Please see the new Figure 11 in the revised manuscript.

*Figure 11: What bias amount does the white colour represent in panel (d)? Are the colours correct in panel (c)? There seems to be a global bias of between 2 and 4mm, based on the colours, yet the number reported in the top is 0.148. Personally, I would*

*recommend using a neutral colour, e.g. white, for errors less than some threshold, e.g. 0.5mm. Otherwise the eye is drawn to places where the bias changes from positive to negative even when such a change is miniscule. But the authors are free to ignore this suggestion if they disagree.*

**Reply:** Thanks for pointing this out. It seems that we ran into some bugs in NCL with the automatic contour levels. Here we use the explicit levels with an interval of 1mm/day to plot the precipitation differences and present them in Figure S5 in the SI. According to Figure S5, NNCAM produces larger precipitation biases than CAM5 in the boreal winter and does not improve that much in the boreal summer even though NNCAM's global mean difference and root mean square error are smaller. We explicitly make such statements in lines 438-441: *"However, on a difference plot (Figure S5), NNCAM moderately underestimates the precipitation along the equator, in the Indian monsoon region, and over the Maritime Continent in the summer (Figure S5a). In the boreal winter, NNCAM simulates a weak SPCZ that is excessively separated from the ITCZ, with both precipitation centers shifted away from each other."*

**Reference:**

Yu, Tianhe, et al. "Gradient surgery for multi-task learning." Advances in Neural Information Processing Systems, 33 (2020): 5824-5836.

**Figure R1**

[Figure]

**Figure R1.** Latitude-pressure cross sections of the zonally averaged coefficient of determination ($R^2$) for (a) heating and (b) moistening predicted using the NN-Parameterization. Both are evaluated at a 30 min time step interval. Note: the areas where $R^2$ is greater than 0.7 are contoured in pink, and the areas where $R^2$ is greater than 0.9 are contoured in orange.

**Table S1**

**Table S1.** The hyperparameter search space. Note that the "Number of Hidden Layers" are used in the fully connected DNNs, "Residual blocks" are used in the ResDNNs, and each Residual block has two fully connected layers.

| Hyperparameter Type | Hyperparameter Space |
| --- | --- |
| Number of Hidden Layers/Residual blocks | 3, 4, 5, 6, 7, 8, 9 |
| Width of Hidden Layers | 64, 128, 256, 512, 1024 |
| Learning rate | 0.001, 0.002, 0.0001, 0.005 |
| Dropout rate | 0, 0.2, 0.3, 0.5 |
| Batch size | 128, 256, 512, 1024 |
| Weight decay | 0, 1e-5, 5e-4 |
| Activation function | ReLU, Leaky-ReLU, Sigmoid |
| Loss function | Mean Squared Error |
| optimizers | Adam, SGD |

[Figure]

**Figure S1.** Spatial distribution of the precipitation STD across all 10 stable NN-Parameterizations for the prognostic simulation from 1999 to 2003.

[Figure]

**Figure S2.** Wave spectra of (a) a stable NN-Parameterization and (b) an unstable parameterization. The light blue background indicates where the phase speed is greater than 5 m/s and the growth rate is positive. The stability diagrams were obtained by coupling the linear responses of the NN-parametrizations with the simplified 2-D dynamics with a chosen base state, which is the normal convection background in the long-term prognostication in (a) and the initial state for the unreal gravity wave in Move S1 in (b).

[Figure]

**Figure S3.** Latitude-pressure cross-section of the zonal and annual mean (a) temperature and (b) specific humidity for NNCAM simulated from 2004 to 2008, with their differences from the SPCAM simulation from 1999 to 2003.

[Figure]

**Figure S4.** Global distribution of the temporal mean precipitation predicted using NNCAM from January 1, 2004, to December 31, 2008, for the (a) annual, (b) boreal summer (JJA), and (c) boreal winter (DJF).

[Figure]

**Figure S5.** Global distributions of the precipitation differences between (a & b) NNCAM and SPCAM and (c & d) CAM5 and SPCAM averaged over the boreal summer (left panels) and winter (right panels).

---

## Author Response (AR3)

**Response to reviewer 1**

**1 Summary**

*Overall, the authors have addressed my comments. After reading the manuscript I still have a major comment on the manuscript regarding how the authors have reported their results. Overall, I think that the manuscript deserves publication but I would like to get a clarification from the author regarding why there are inconsistencies in the results between the different versions of the manuscript.*

We appreciate you very much for the suggestions that improved our work. We performed sensitivity tests and re-ran prognostic simulations for better consistency based on the reviewers' suggestions. As you mentioned, there are inconsistencies in the results between the different versions of the manuscript. We try to make a clarification as follows.

**2 Major Comments**

*I might be misinterpreting what the authors have been writing, but I think that the authors might be using some inconsistent reporting when they present their data. Mostly I am worried since during the evolution of the paper between iterations, there are some inconsistencies in the text and figures, and at least to my understanding the authors did not clarify these differences. I suggest that the authors would clarify these inconsistencies. My main concern is rooted in how the authors report and write about their methodology of trial and error and in the large differences between the results presented in the first version of the manuscript and the recent version of the manuscript.*

*• Results in Figures 12 and 14. The results presented in figure 12 shows overall that NNCAM is performing better than CAM in terms of latitudinal distribution of precipitation. However, in the first version of the manuscript a similar figure was shown (figure 11 in the first version) and in the first version NNCAM was performing substantially worse than CAM. How come this result was changed? Did the authors change something that caused this difference, and if yes where was it reported? I know that now the analysis is performed over more years, but I doubt that this can explain the large differences between these results. (A similar comment on figure 14 that is now different compared to figure 13 in the first version of the manuscript).*

**Reply**: Thanks for this comment. In all versions of manuscripts, the prognostic validation uses the same NNCAM, and only the model start date is different.

In the 1st version of the manuscript, NNCAM started with the SPCAM checkpoint on July 1, 1998, and SPCAM and CAM started on January 1, 1998. We chose the next four years, from January 1, 1999 to December 31, 2002 for evaluation. And this SPCAM checkpoint was made by running SPCAM from 1998-01-01 to 1998-07-01.

As requested by the Reviewer 1, in the 2nd and 3rd versions of the manuscripts, the start time of NNCAM, CAM, and SPCAM was all January 1, 1998. All models were run for 6 years, with the first year for spin up and the next 5 years (January 1, 1999, to December 31, 2003) for evaluation and comparison.

By applying a different way of the startup, the simulations results show some differences in the land-surface precipitation (the right column in Figure 12) and the MJO spectra signals (Figure 14), which is an interesting point and should be studied in the future.

*Results from online tests. In the first version of the manuscript the authors wrote that: "prepared 50 groups of ResMLPs with similar R2 as candidate models using different train samples and epochs. Secondly, we conducted comprehensive prognostic tests on these candidate neural networks and obtained the feasible NN-Parameterization schemes that can support NNCAMs stable simulation for multiple years." From that statement it is clear that the authors have run online simulations with at least 50 NNs and some of them were stable. Since this is an important part of the authors' work (where they also argue that a fully connected DNN are not stable and therefore they use a different architecture) in the first revision I requested that they give more details about how many networks were stable and how many were not (for each type of NN that they tried). In the response to my first review the authors wrote that they add:*

*"Figure 4 shows the offline validation versus the number of prognostic steps that our NNCAM can run. First, the DNN parameterizations are less accurate than the ResDNN ones in terms of offline validation accuracy. As a results, all the DNN parameterizations cannot run stably longer than half a year in prognostic tests. For the ResDNNs (blue dots and black inverted triangles), the less well-trained ones with high MSE crash within half a year simulation. However, when the offline MSE of ResDNN decreases to a certain level the ResDNN parameterization may run stably for long periods. In Figure 4, we observed 4 ResDNNs can run stably."*

*In the figure that the authors added they show 3 DNN and 21 ResDNNs. In the initial version of their manuscript the author wrote that they have prepared and tested 50 groups of networks so I am not sure why they report only on part of their results.*

*Furthermore, since I thought that the comparison between 21 runs (out of the 21 runs of ResDNN only 4 of the runs were stable) and 3 runs of DNN cannot support the claim that DNNs aren't stable is not established since only very few networks DNN were examined. In a response for my comment the authors included in their latest response:*

*"We tested all 37 NN sets (27 ResDNN sets and 10 DNN sets) in the sensitivity tests. As shown in Figure 4, there are 10 ResDNN sets that can sustain simulations of longer than 10 years. Figure 4 is intended to show the relationship between the MSEh and the stability, not to prove that the ResDNN is better than the DNN in terms of stability."*

*What I found strange is that they have added 3 additional simulations with the ResDNN case, and found all additional simulations to be stable (I can count only 24 ResDNN and not 27; BTW, the addition of DNNs makes the argument much better). I find this*

*result of adding 3 additional ResDNN simulations and finding them to be all stable is not very likely given that they previously reported to find that only 4 out of 21 simulations were stable but maybe I am missing something here, so it would be great if the authors could clarify.*

*I would encourage the authors to include all the results they have obtained and reported in the first version of the manuscript (e.g., could the authors provide the results of the 50 simulations they reported in the first version of the manuscript? Is there a reason that these results are not included?)*

**Reply:** Thanks for the question. It should be noted that all trainsets used in the 1st, 2nd, and 3rd versions of the manuscripts are split and sampled by time on the same original data which are stored open-soured on Zenodo as netCDF files (https://doi.org/10.5281/zenodo.5625616). Furthermore, we use the same set of ResDNNs for prognostic validation in all versions of the manuscripts. This set of ResDNNs has been open-sourced at Zenodo (https://doi.org/10.5281/zenodo.5596273).

To make our sensitivity tests more convincing in later versions of the manuscripts, we reorganized and described the trainset according to the suggestions by Reviewer 1. Especially, the testset was changed to the SPCAM's simulation in 2000 which is well separated from the trainset. To ensure the rigor of the sensitivity test, we did not use the first 50 NN sets because the conditions (datasets for training and evaluation) to build these NN sets are not guaranteed to be the same as the later 37 NN sets in sensitivity tests. In the 2nd revision of the manuscripts, we just arbitrarily selected 21 NN sets (18 ResDNN sets and 3 DNN sets) from the 37 NN sets and performed prognostic runs according to the settings in section 3.1. As requested by Reviewer 1, in the 3rd version of the manuscript, we included all the 37 NN sets by performing prognostic runs over the rest NN sets (9 ResDNN sets and 7 DNN sets) for sensitivity tests.

Among the 27 ResDNN sets, 10 ResDNN sets are stable, shown in the Figure 4 with close $MSE_h$ in x-axis and large prognostic steps marked by infinite line, leading to several (the results of four sets of ResDNNs) points overlapping. We replotted Figure 4 and showed three overlapping points above the infinity line.

[Figure]

Figure 4. The mean square error of the offline moist static energy vs. the prognostic steps. The black inverted triangles (the three black inverted triangles above the infinity line to avoid overlapping) denote stable NN coupled prognostic simulations that last for more than 10 years. The blue dots denote unstable simulations, and the blue triangles denote unstable DNNs. The dots with colored outlines are shown in Figure 5 for the time evolution of the globally averaged energy.

**3 minor comments**

• *lines 100-102: The authors write that the Yuval and O'gorman 2020 used random forest to predict fluxes to ensure physical constraints, but this is not correct. As far as I understand the usage of random forest allows to ensure linear physical constraints because RF just averages over samples.*

**Reply:** Thanks for pointing this out. We have changed them in the latest revised manuscript in line 89-91 as: *"Yuval and O'Gorman (2020) used the random forest algorithm to develop an ML parameterization based on training data from a high-resolution idealized 3-D model with a setup on the equatorial beta plane. They used two independent random forests to separately emulate different processes. Later, Yuval et al. (2021) ensured the physical constrains by using an NN parameterization with a special structure to predict the subgrid fluxes instead of tendencies"*

• *line 168: the authors use the terms qrs and qrl but they never define these.*

**Reply:** Thanks for pointing this out. We have removed the terms *qrs* and *qrl* in line 153.

**Response to reviewer 2**

**1 Summary**

*I thank the authors for the revised manuscript and replies. I think the article is much clearer with the advantages and disadvantages of the neural network models. I have a few minor points below but otherwise I would be happy for the manuscript to be accepted.*

*In my opinion, the tool that the authors develop to do coupling in parallel simulations is worthy of more detail and publicity. I would suggest that the authors package their tool up with clear documentation. If this is well done, and the tool has the advantages they suggest over existing coupling methods, then researchers in the field will be keen to adopt the tool.*

Thank you very much for your constructive comments. We have uploaded the coupler NN-GCM in https://doi.org/10.5281/zenodo.5596273 together with the NNCAM model. Proper instructions were added to the root directory of open-sourced codes and described in the manuscript. We are making further development on it and would like to share the tool with the community. We reply to your comments as follows.

**2 minor comments**

*R1 is an interesting figure. I think it should be included in the supplementary material and a reference that it is there be added to Figure 7. I understand that the authors wanted figure 7 to compare with previous articles but I think R1 is a valuable addition too.*

**Reply:** Thanks for the comment. We have added the Figure R1 in the supplementary as the Figure S6 combined with a reference information in the figure captions.

*L295: "10 DNN sets and dozens" - please give the exact number here, imprecision is not helpful.*

**Reply:** Thanks. We have made clear that there are 27 ResDNN sets in the latest manuscript in line 295.

*L304: g & Cp no longer features in these equations.*

**Reply:** Thanks for the comment. *"Cp"* is no longer used in the equation and we have removed it in line 304.